# MONARCH MIXER: A Simple Sub-Quadratic GEMM-Based Architecture

**Daniel Y. Fu**[1], **Simran Arora**[*,1], **Jessica Grogan**[*,2], **Isys Johnson**[*,2], **Sabri Eyuboglu**[*,1],
**Armin W. Thomas**[*,3], **Benjamin Spector**[1], **Michael Poli**[1], **Atri Rudra**[2], **Christopher Ré**[1]

[*]Equal Contribution. [1]Department of Computer Science, Stanford University.
[2]Department of Computer Science and Engineering, University at Buffalo, SUNY.
[3]Department of Psychology, Stanford University.

danfu@cs.stanford.edu, simarora@stanford.edu,
{jrgrogan,isysjohn}@buffalo.edu, {eyuboglu,athms,bfs,poli}@stanford.edu,
atri@buffalo.edu, chrismre@cs.stanford.edu

## Abstract

Machine learning models are increasingly being scaled in both sequence length and model dimension to reach longer contexts and better performance. However, existing architectures such as Transformers scale quadratically along both these axes. We ask: are there performant architectures that can scale *sub-quadratically* along sequence length and model dimension? We introduce MONARCH MIXER (M2), a new architecture that uses the same sub-quadratic primitive along both sequence length and model dimension: Monarch matrices, a simple class of expressive structured matrices that captures many linear transforms, achieves high hardware efficiency on GPUs, and scales sub-quadratically. As a proof of concept, we explore the performance of M2 in three domains: non-causal BERT-style language modeling, ViT-style image classification, and causal GPT-style language modeling. For non-causal BERT-style modeling, M2 matches BERT-base and BERT-large in downstream GLUE quality with up to 27% fewer parameters, and achieves up to $9.1\times$ higher throughput at sequence length 4K. On ImageNet, M2 outperforms ViT-b by 1% in accuracy, with only half the parameters. Causal GPT-style models introduce a technical challenge: enforcing causality via masking introduces a quadratic bottleneck. To alleviate this bottleneck, we develop a novel theoretical view of Monarch matrices based on multivariate polynomial evaluation and interpolation, which lets us parameterize M2 to be causal while remaining sub-quadratic. Using this parameterization, M2 matches GPT-style Transformers at 360M parameters in pretraining perplexity on The PILE—showing for the first time that it may be possible to match Transformer quality without attention or MLPs.[1]

## 1 Introduction

Machine learning models in natural language processing and computer vision are being stretched to longer sequences and higher-dimensional representations to enable longer context and higher quality, respectively [6, 10, 62, 84]. However, existing architectures exhibit time and space complexities that grow quadratically in sequence length and/or model dimension—which limits context length and makes scaling expensive. For example, attention and MLP in Transformers scale quadratically in sequence length and model dimension [15]. In this paper, we explore a natural question: *can we find a performant architecture that is sub-quadratic in both sequence length and model dimension?*

---

[1]Code is available at https://github.com/HazyResearch/m2.

37th Conference on Neural Information Processing Systems (NeurIPS 2023).

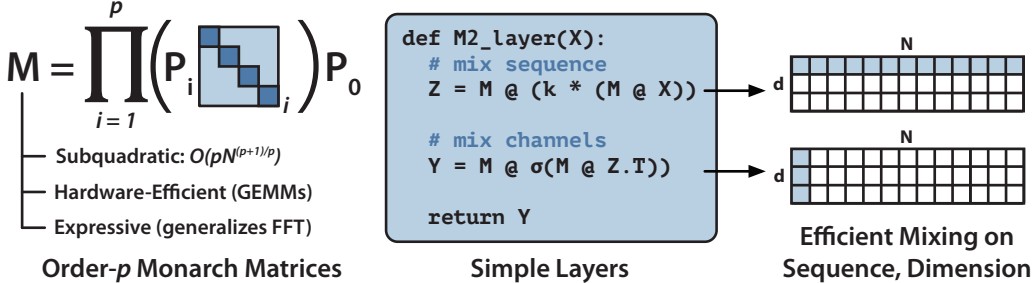

Figure 1: Monarch matrices are a simple, expressive, and hardware-efficient class of sub-quadratic structured matrices. MONARCH MIXER (M2) uses Monarch matrices to mix inputs first along the sequence dimension and then along the model dimension. See the Appendix for PyTorch implementation of an M2 layer.

In our exploration, we seek a sub-quadratic primitive for both the sequence length and model dimension. Our framing takes inspiration from work such as MLP-mixer [73] and ConvMixer [74], which observed that many machine learning models operate by repeatedly *mixing* information along the sequence and model dimension axes, and used a single operator for both axes. Finding mixing operators that are expressive, sub-quadratic, and hardware-efficient is challenging. For example, the MLPs in MLP-mixer and convolutions in ConvMixer are expressive, but they both scale quadratically in their input dimension [73, 74]. Several recent studies have proposed sub-quadratic sequence mixing with long convolutions or state space models [27, 64, 77]—both computed using the FFT—but these models have poor FLOP utilization (3-5% [28]) and maintain quadratic scaling in model dimension. Meanwhile, there has been promising work in sparsifying dense MLP layers without losing quality, but some of the models can actually be *slower* than their dense counterparts, due to low hardware utilization [7, 8, 14, 26, 35].

We turn to an expressive class of sub-quadratic structured matrices called *Monarch matrices* [14] (Figure 1 left) to propose MONARCH MIXER (M2). Monarch matrices are a family of structured matrices that generalize the fast Fourier transform (FFT) and have been shown to capture a wide class of linear transforms including Hadamard transforms, Toeplitz matrices [32], AFDF matrices [57], and convolutions. They are parameterized as the products of block-diagonal matrices, called *monarch factors*, interleaved with permutation. Their compute scales sub-quadratically: setting the number of factors to $p$ results in computational complexity of $O(pN^{(p+1)/p})$ in input length $N$, allowing the complexity to interpolate between $O(N \log N)$ at $p = \log N$ and $O(N^{3/2})$ at $p = 2$.[2]

M2 uses Monarch matrices to mix information along the sequence and model dimension axes. It is both simple to implement and hardware-efficient: the block-diagonal Monarch factors can be computed efficiently on modern hardware using GEMMs (generalized matrix multiply algorithms). Our proof-of-concept implementation of an M2 layer, written in less than 40 lines of pure PyTorch (including imports), relies only on matrix multiplication, transpose, reshape, and elementwise products (see pseudocode in Figure 1 middle) and achieves 25.6% FLOP utilization[3] for inputs of size 64K on an A100 GPU. On newer architectures such as the RTX 4090, a simple CUDA implementation achieves 41.4% FLOP utilization at the same size.

**Non-Causal Settings** As a first proof of concept of M2, we evaluate how it compares to Transformers in terms of speed and quality in non-causal settings such as BERT-style masked language modeling [21] and ImageNet classification. We introduce M2-BERT, which replaces the attention blocks in BERT with bidirectional gated convolutions implemented using Monarch matrices and replaces the dense matrices in the MLP with Monarch matrices. M2-BERT reduces parameter count but maintains quality—matching BERT-base and BERT-large in downstream GLUE quality with 27% and 24% fewer parameters, respectively. Sub-quadratic scaling in sequence length enables high throughput at longer sequences—up to 9.1× higher throughput at sequence length 4K than

---

[2]Monarch matrices were originally [14] parameterized with $p = 2$, but the general $p$ case is a natural extension.

[3]For context, the most optimized attention implementations achieve 25% FLOP utilization, while unoptimized implementations of attention can have as low as 10% FLOP utilization [15].

HuggingFace BERT, and 3.1× higher throughput at sequence length 8K than BERT optimized with FlashAttention [15].

For image classification, we adapt HyenaViT-b [64], an attention-free vision transformer based on gated convolutions. We replace the convolution operation with M2 primitives and replace the MLP layers with an M2 block as well. These changes reduce the parameter count compared to a ViT-b [22] model with the same model width and depth by a factor of 2. Surprisingly, despite this parameter reduction, we find that M2 slightly outperforms ViT-b and HyenaViT-b baselines, achieving 1% higher accuracy on ImageNet [18].

**Causal Settings** Causal settings such as GPT-style [65] auto-regressive language modeling present a technical challenge: masking out the upper triangular elements in an attention matrix (or equivalent structure) introduces a quadratic bottleneck. To alleviate this quadratic bottleneck with Monarch matrices, we develop new theory to characterize which parameterizations of Monarch matrices maintain causality. To do so, we take a view of $p$-order Monarch matrix multiplication as $p$-variate polynomial evaluation and interpolation (e.g., $p = 2$ factors corresponds to bivariate polynomials, Figure 2 left). Using this view, we show that the M2 convolution shown in Figure 1 (middle) can be viewed as manipulation of modular polynomial multiplication. This result allows us to develop conditions (Theorem 3) under which M2 is causal. We can use this causal parameterization to outperform GPT-style language models on causal language modeling by 0.2 PPL points on the PILE at model size 360M–without using either attention or MLP blocks.

**Summary** Overall, our results present a potential path to building machine learning models with sub-quadratic primitives. We hope our work can serve as a starting point to explore models that are more efficient in both sequence length and model dimension.

## 2 Preliminaries

In this section, we provide some background on the key components behind the cost of operations on GPUs, and then discuss the scaling characteristics of some common primitives used to mix information across the sequence dimension and model dimension in modern machine learning models.

**GPU Accelerator Cost Model** We provide a brief discussion of relevant factors affecting runtime performance of deep learning operations on GPUs. Depending on the balance of computation and memory accesses, operations can be classified as either compute-bound or memory-bound [44]. In compute-bound operations, the time accessing GPU memory is relatively small compared to the time spent doing arithmetic operations. Typical examples are matrix multiply with large inner dimension, and short convolution kernels with a large number of channels.

The speed of these operations is determined by the FLOP/s available on compute units, and the number of FLOPs necessary to complete the operation. In our paper, we exploit fast matrix multiply units such as tensor cores. On the A100, tensor cores can achieve 312 TFLOP/s in half-precision matrix multiply operations, while non-matrix multiply operations are limited to 19 TFLOP/s [59]. This trend began with tensor cores in the V100 [58], and is continuing into the next-generation H100 chips [60].

In memory-bound operations, the time taken by the operation is determined by the number of memory accesses, while time spent in computation is much smaller. Examples include most elementwise operations (e.g., activation, dropout) and reductions (e.g., sum, softmax, batch norm, layer norm).

The runtime of memory-bound operations is determined by the memory bandwidth of different layers of the *memory hierarchy*. GPU memory is large but relatively slow—up to 80 GB on A100, but with bandwidth of 2 TB/s [59]. Higher levels of the memory hierarchy such as caches are much smaller (20 MB) but an order of magnitude faster (19 TB/s).

Table 1: FLOP cost and utilization of various mixer layers, input dimension 64K on an RTX 4090.

| Layer | FLOP Cost | Util |
|---|---|---|
| MLP | $N^2$ | 95.5% |
| FlashAttn | $N^2$ | 24.0% |
| FFT | $N \log N$ | 3.0% |
| M2 Conv | $N^{3/2}$ | 41.4% |

**Common Mixer Primitives**   To help contextualize our work, we provide scaling and hardware utilization characteristics for a few common operations that are used to mix information in machine learning models, summarized in Table 1.

Transformers [75] use attention to mix information across the sequence dimension, and MLP blocks to mix information across the model dimension. Both of these blocks scale quadratically in input length. MLP layers are compute-bound, so they have high FLOP utilization out of the box. Attention blocks are memory-bound, so even the most optimized implementations such as FLASHATTENTION [15] have relatively lower FLOP utilization.

Recent work has made progress towards attention-free models by replacing attention layers with long convolution layers, interleaved with elementwise gating [27, 28, 36, 54, 64, 67–69]. These layers are computed using FFT operations using the FFT convolution theorem: $y = \mathbf{K} * \mathbf{X} = FFT^{-1}(FFT(\mathbf{X}) * FFT(\mathbf{K}))$. While the FFT scales asymptotically well in $O(N \log N)$, it is often memory-bound and thus has low FLOP utilization. In our work, we aim to construct a mixer that has both sub-quadratic scaling and high FLOP utilization.

# 3   MONARCH MIXER

In this section, we recall Monarch matrices, introduce how M2 uses Monarch matrices to mix along the sequence and model dimensions, and benchmark a M2 convolution in terms of hardware utilization.

## 3.1   Monarch Matrices

Monarch matrices [14] are a sub-quadratic class of structured matrices that are hardware-efficient and expressive. They can represent many linear transforms, including convolutions, Toeplitz-like transforms, low-displacement rank transforms, and orthogonal polynomials. Directly implementing these different structured transforms on GPUs as dense matrices can be inefficient. In contrast, their Monarch decompositions can be computed by interleaving matrix multiplications with tensor permutations.

A Monarch matrix $\mathbf{M} \in \mathbb{R}^{N \times N}$ of order $p$ is defined by the following:

$$\mathbf{M} = \left( \prod_{i=1}^{p} \mathbf{P}_i \mathbf{B}_i \right) \mathbf{P}_0, \tag{1}$$

where each $\mathbf{P}_i$ is related to the 'base $\sqrt[p]{N}$' variant of the bit-reversal permutation, and $\mathbf{B}_i$ is a block-diagonal matrix with block size $b$. Setting $b = \sqrt[p]{N}$ achieves *sub-quadratic* compute cost. For example, for $p = 2, b = \sqrt{N}$, Monarch matrices require $O(N^{3/2})$ compute in sequence length $N$.

In this paper, we use Monarch matrices to construct architectures that are sub-quadratic in both sequence length $N$ and model dimension $d$. We will often parameterize order-2 Monarch matrices, written as $\mathbf{M} = \mathbf{PLPRP}$, where $\mathbf{L}$ and $\mathbf{R}$ are block-diagonal matrices (for "left" and "right"), and $\mathbf{P} = \mathbf{P}_2 = \mathbf{P}_1 = \mathbf{P}_0$ is a permutation that reshapes the input to 2D, transposes it, and flattens it to 1D. A common case is to set $\mathbf{L} = \mathbf{R} = (\mathbf{I}_{\sqrt{N}} \otimes \mathbf{F}_{\sqrt{N}})$, where $\mathbf{F}_{\sqrt{N}}$ is a $\sqrt{N}$ DFT matrix, and $\otimes$ is the Kronecker product.

## 3.2   MONARCH MIXER Architecture

We describe how MONARCH MIXER uses Monarch matrices and elementwise operations to construct sub-quadratic architectures (Figure 1 middle). We take a mixer view of model architectures, where each layer is a sequence of mixing operations across the sequence and the model dimension axes. Each layer takes as input a sequence of embeddings $\mathbf{X} \in \mathbb{R}^{N \times d}$, and outputs a sequence $\mathbf{Y} \in \mathbb{R}^{N \times d}$, where $N$ is the sequence length, and $d$ is the model dimension. For simplicity, we show the order-2 case here, though we can use higher-order blocks to scale to longer sequences and larger model dimensions.

Let $\mathbf{M}_1, \mathbf{M}_2 \in \mathbb{R}^{N \times N}$ and $\mathbf{M}_3, \mathbf{M}_4 \in \mathbb{R}^{d \times d}$ be order-2 Monarch matrices, let $\mathbf{K}_1 \in \mathbb{R}^{N \times d}$, let $\sigma$ be an optional point-wise non-linearity (*e.g.* ReLU), and let $\odot$ be elementwise multiplication. M2

Table 2: FLOP cost and utilization of M2 compared to dense MLP at different input sizes $N$, with block size $\sqrt{N}$, on an A100 and RTX 4090.

| $N$ | 4K | 16K | 64K | 256K |
|---|---|---|---|---|
| Dense Matmul TFLOP Cost | 0.025 | 0.412 | 6.60 | 106.0 |
| M2 TFLOP Cost | 0.002 | 0.013 | 0.103 | 0.824 |
| Dense FLOP Utilization (A100) | 63.0% | 78.0% | 80.0% | OOM |
| M2 FLOP Utilization (A100) | 4.78% | 12.7% | 25.6% | 42.8% |
| Wall-Clock Speedup (A100) | 1.2× | 5.1× | 20.6× | >55.0× |
| Dense FLOP Utilization (4090) | 74.6% | 96.7% | 98.0% | OOM |
| M2 FLOP Utilization (4090) | 11.1% | 32.1% | 41.4% | 53.7% |
| Wall-Clock Speedup (4090) | 2.2× | 10.5× | 27.0× | >69.1× |

uses Monarch matrices to construct expressive architectures. For example, a convolutional block with a sparse MLP can be expressed as follows:

1. Mix along **sequence** axis:
$$\tilde{\mathbf{X}} = \mathbf{M}_2(\mathbf{K}_1 \odot \mathbf{M}_1 \mathbf{X}) \tag{2}$$

2. Mix along **embedding** axis:
$$\mathbf{Y}^\top = \mathbf{M}_4 \sigma(\mathbf{M}_3 \tilde{\mathbf{X}}^\top) \tag{3}$$

When $\mathbf{M}_1$ is set to the DFT and $\mathbf{M}_2$ is set to the inverse DFT, Equation 2 exactly corresponds to a convolution with kernel $\mathbf{K}_1$ parameterized in frequency space. Equation 3 corresponds to an MLP with the dense matrices replaced by Monarch matrices. More expressive layers are also easily expressible; for example, replacing Equation 2 with $\mathbf{V} \odot \mathbf{M}_2(\mathbf{K}_1 \odot \mathbf{M}_1(\mathbf{Q} \odot \mathbf{K}))$, where $\mathbf{Q}, \mathbf{K}, \mathbf{V}$ are linear projections of $\mathbf{X}$, reproduces a gated convolution block, as in [27, 28, 64].

The basic M2 layer is simple to implement; pseudocode is shown in Figure 1 (middle), and the Appendix gives an efficient implementation of M2 in under 40 lines of pure PyTorch (including imports). The convolution case with Monarch matrices fixed to DFT and inverse DFT matrices also admits implementations based on FFT algorithms [11].

### 3.3 Architecture Benchmarks

We benchmark the efficiency of the $\mathbf{M}(\mathbf{K} \odot \mathbf{M}\mathbf{X})$ convolution operator (Equation 2) implemented in a simple CUDA kernel (calling standard cuBLAS sub-routines [61]), as the dimension $N$ increases. Equation 3 scales similarly, as dimension $d$ increases. We keep the block size $b$ fixed to $\sqrt{N}$.

Table 2 shows the FLOP cost and utilization of a M2 operator as a function of the input size on an A100 as well as on an RTX 4090. On the A100, the operator is more dominated by the data movement costs of the permutation operations (see the Appendix for a roofline analysis). For longer inputs, the sub-quadratic scaling allows MONARCH MIXER to outperform dense matrix multiplication. On the RTX 4090, which has a larger and faster L2 cache than the A100, we can manually optimize an implementation to amortize data movement costs.

## 4 Theoretical Analysis: M2 as Polynomial Multiplication

In this section, we develop theory to make the M2 layer causal in the input $\mathbf{X}$—e.g., ensure that an output $Y_i$ of the M2 should only depend on $X_1, ..., X_i$. Our approach involves interpreting Monarch matrix multiplication as multivariate polynomial evaluation and interpolation. We then show that an M2 convolution is equivalent to modular polynomial manipulation in a univariate basis.

The challenge is controlling the degrees of the resulting univariate polynomials, to prevent "underflow" under modular multiplication (see Figure 2 for an overview). Our key result is deriving sufficient conditions on the degrees of the bivariate polynomials defining the Monarch factors to prevent such underflow. We focus on the bivariate case (order $p = 2$) in the body, and give the general multivariate case in the Appendix. We present proof sketches in the main body, and leave proofs and additional results for the Appendix.

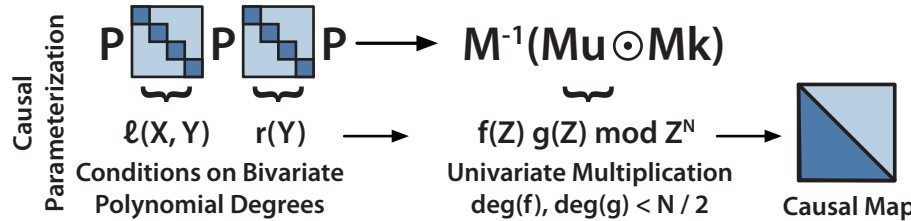

Figure 2: Monarch multiplication can be interpreted as polynomial evaluation and interpolation. We derive sufficient conditions on the polynomial formulation of Monarch matrices for M2 to be causal.

**Monarch Multiplication as Polynomial Evaluation**    First, we show that order-2 Monarch matrix-vector multiplication $\mathbf{M} \cdot \mathbf{u}$ is equivalent to bivariate polynomial evaluation.

Fix a Monarch matrix $\mathbf{M} \in \mathbb{R}^{N \times N} = \mathbf{PLPRP}$, for two block-diagonal matrices $\mathbf{L}$ and $\mathbf{R}$ with blocks of size $b = \sqrt{N}$. We can interpret Monarch matrices as bivariate polynomial evaluation by setting $A = \{\omega_0, \ldots, \omega_{b-1}\}$ as a set of evaluation points (e.g., the $b$th roots of unity), and letting $\{\ell_0(X, Y), \ldots, \ell_{b-1}(X, Y)\}, \{r_0(Y), \ldots, r_{N-1}(Y)\}$ be sets of basis polynomials with individual degrees of $X, Y$ being $< \sqrt{N}$. The values of $\{\ell_0(X, Y), \ldots, \ell_{b-1}(X, Y)\}$ evaluated on $A^2$ determine the entries of $\mathbf{L}$, and the values of $\{r_0(Y), \ldots, r_{N-1}(Y)\}$ evaluated on $A$ determine the entries of $\mathbf{R}$. We give the mapping from $\ell, r$, and $A$ to $\mathbf{L}$ and $\mathbf{R}$ in the Appendix.

Then, matrix-vector multiplication between $\mathbf{M}$ and a vector $\mathbf{u}$ is equivalent to polynomial evaluation of the basis functions $\ell, r$ on the evaluation points $A^2$:

**Theorem 1.** *Let $m(j) = j \mod \sqrt{N}$. For any vector $\mathbf{u} \in \mathbb{R}^N$, $\mathbf{Mu}$ is a bivariate polynomial $u(X, Y)$ evaluated at $A^2$, with $u(X, Y) = \sum_{j=0}^{N-1} u_j f_j(X, Y)$, where $f_j(X, Y) = \ell_{m(j)}(X, Y) r_j(Y)$.*

**Monarch Inverse as Polynomial Interpolation**    Next, we exploit the fact that Monarch inverse multiplication $\mathbf{M}^{-1} \cdot \mathbf{u}$ is equivalent to polynomial interpolation in the basis polynomials of $\mathbf{M}$.

**Theorem 2.** *Let $\mathbf{M}_0, \mathbf{M}_1, \mathbf{M}_2$ be Monarch matrices, and let $A$ be the set of $\sqrt{N}$ roots of unity. Then, the operation*

$$\mathbf{f} = \mathbf{M}_0^{-1} \left( (\mathbf{M}_1 \mathbf{k}) \odot (\mathbf{M}_2 \mathbf{u}) \right). \tag{4}$$

*is equivalent to representing the polynomial*

$$h(X, Y) = k(X, Y) u(X, Y) \mod (X^{\sqrt{N}} - 1, Y^{\sqrt{N}} - 1)$$

*in terms of the basis polynomials $\ell, r$ corresponding to $\mathbf{M}_0$, and where $k(X, Y)$ and $u(X, Y)$ are the polynomials corresponding to $\mathbf{M}_1 \mathbf{k}$ and $\mathbf{M}_2 \mathbf{u}$, respectively.*

The above follows from Theorem 1 and the fact that Monarch matrix-vector multiplication with an inverse Monarch matrix is equivalent to polynomial interpolation in a given basis. The mod part comes from the fact that $A$ is the set of roots of the polynomial $Z^{\sqrt{N}} - 1$.

**Causal Monarch Maps**    Now, we give a class of Monarch matrices from which we can build a causal map. First, we define a polynomial with *minimum* degree $j$:

**Definition 1.** *A polynomial of minimum degree $j$ (and maximum degree $N - 1$) is defined as $\bar{q}_j(Z) = \sum_{a=j}^{N-1} \bar{q}_j[a] Z^a$.*

To ensure causality, we first convert the bivariate polynomial basis into a univariate basis, and then we expand the degree of the univariate polynomial. The resulting univariate polynomial multiplication is naturally causal (exploiting similar properties as the causal FFT convolution).

We use the Kronecker substitution ($X \leftarrow Z, Y \leftarrow Z^{\sqrt{N}}$) to convert the bivariate polynomial basis into a univariate basis:

$$q_j(Z) = \ell_{m(j)}(Z) r_j \left( Z^{\sqrt{N}} \right), \tag{5}$$

where $m(j)$ is defined as in Theorem 1.

Then, the following class of Monarch matrices (with the conversion to univariate polynomial basis as above) forms a causal map:

**Theorem 3.** *Let* $\mathbf{u}, \mathbf{k} \in \mathbb{R}^n$, *where* $n < N/2$. *Let* $m(j)$ *be as in* Theorem 1, *and* $k(j) = \left\lfloor j/\sqrt{N} \right\rfloor$. *Then define the basis polynomials* $\ell_{m(j)}$ *to have minimum degree* $m(j)$, *basis polynomials* $r_j$ *to have minimum degree* $k(j)$, *and all polynomials* $q_j(Z)$ *to have maximum degree* $< N/2$ *for all* $j < N/2$ *and for* $N/2 \leq j < N$ *have maximum degree* $N - 1$. *Let* $\mathbf{M}_N$ *be defined by such basis polynomials via* (5) *where the evaluation points are now the* $N$th *roots of unity. Then, we have that*

$$\mathbf{u} \mapsto \left(\mathbf{M}_N^{-1}(\mathbf{M}_N(\mathbf{k}, \mathbf{0}_{N-n}) \odot \mathbf{M}_N(\mathbf{u}, \mathbf{0}_{N-n}))\right)[0 : n - 1] \tag{6}$$

*gives a causal map in* $\mathbf{u}$.

Theorem 3 gives a causal map that can be computed entirely using Monarch matrices – enforcing causality with sub-quadratic scaling. The main technical ingredient in proving the above result is that the product $q_j(Z)q_{j'}(Z)$ can be written as a linear combination of $q_a(Z)$ for $j + j' \leq a < N$ (this uses the above specified properties on the minimum and maximum degrees of $q_j(Z)$). This in turn implies that the term $k_{j'}u_j q_j(Z)q_{j'}(Z)$ only contributes to the coefficients of "higher order" basis polynomials $q_a(Z)$ for $a \geq j + j'$ in the product $k(Z)u(Z)$, which is needed for causality. Figure 2 gives an example of restricted polynomials generating a causal map.

# 5 Experiments

We compare MONARCH MIXER to Transformers on three tasks where Transformers have been dominant: BERT-style non-causal masked language modeling, ViT-style image classification, and GPT-style causal language modeling. In each, we show that we can match Transformers in quality using neither attention nor MLPs. We additionally evaluate wall-clock speedups against strong Transformer baselines in the BERT setting. Additional experiments on speech and alternative architectures are given in Appendix B, and experimental details are given in Appendix C.

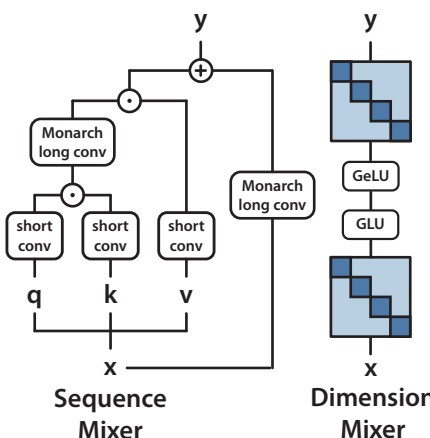

Figure 3: M2-BERT uses Monarch matrices to create a bidirectional gated long convolution in the sequence mixer, and uses Monarch matrices to replace the linear layers in the dimension mixer.

## 5.1 Non-Causal Language Modeling

We introduce M2-BERT, an M2-based architecture for non-causal language modeling. M2-BERT acts as a drop-in replacement for BERT-style language models [21], which are a workhorse application of the Transformer architecture [1, 39, 40, 45, 48, 49, 52, 56, 85, 89]. We train M2-BERT using masked language modeling over C4 [66] with the `bert-base-uncased` tokenizer.

M2-BERT starts with a Transformer backbone and replaces the attention and MLPs with M2 layers, shown in Figure 3. In the sequence mixer, we replace attention with bidirectional gated convolutions with a residual convolution (Figure 3 left). To recover convolutions, we set the Monarch matrices to DFT and inverse DFT matrices. Following [27, 64], we also add short depthwise convolutions after the projections. In the dimension mixer, we replace the two dense matrices in MLPs with learned block-diagonal matrices (Monarch matrix of order 1, $b = 4$). We pretrain two M2-BERT-base models, at 80M and 110M, and two M2-BERT-large models, at 260M and 341M. These are equivalent to BERT-base and BERT-large, respectively.

**Downstream GLUE Scores** First, we evaluate M2-BERT models on downstream fine-tuning compared to BERT-base and BERT-large from [20]. We take the pretrained models and fine-tune them on BERT, following the procedure in [38]. Table 3 shows performance for BERT-base equivalent models, and Table 4 shows performance for BERT-large equivalent models. M2-BERT-base can

Table 3: Average GLUE Score for M2-BERT-base compared to BERT-base [20], along with change in parameters and GLUE score.

| Model | GLUE Score | Δ Params | Δ GLUE Score |
|---|---|---|---|
| BERT-base (110M) | 79.6 | -0% | +0.0 |
| M2-BERT-base (80M) | 79.9 | -27% | +0.3 |
| M2-BERT-base (110M) | **80.9** | -0% | +1.3 |

Table 4: Average GLUE Score for M2-BERT-large compared to BERT-large [20], along with change in parameters and GLUE score.

| Model | GLUE Score | Δ Params | Δ GLUE Score |
|---|---|---|---|
| BERT-large (340M) | 82.1 | -0% | +0.0 |
| M2-BERT-large (260M) | 82.2 | -24% | +0.1 |
| M2-BERT-large (341M) | **82.8** | +0.2% | +0.7 |

match BERT-base in GLUE quality with 27% fewer parameters—or outperform BERT-base in quality by 1.3 points when parameter matched. M2-BERT-large matches BERT-large with 24% fewer parameters, and outperforms by 0.7 points when parameter matched.

**GPU Throughput by Sequence Length**   Next, we evaluate throughput of M2-BERT models by sequence length, compared to HuggingFace implementations of BERT, as well as optimized implementations of BERT running FlashAttention [15]. Table 5 shows forward throughput for BERT-base equivalent models, and the appendix shows throughput for BERT-large (where the performance trends are similar). Inference times are reported in tokens/ms on an A100-40GB GPU. M2-BERT-base achieves higher throughput than even highly-optimized BERT models, and up to $9.1\times$ faster throughput than a standard HuggingFace implementation at sequence length 4K.

**CPU Inference Latency**   Finally, we report CPU inference latency for M2-BERT-base (80M) compared to BERT-base, running direct PyTorch implementations for both. In short sequences, the impacts of data locality still dominate the FLOP reduction, and operations such as filter generation (which are not present in BERT) pay a higher cost. Starting at sequences 1K and longer, M2-BERT-base starts to have speedup over BERT-base, up to $6.5\times$ at sequence length 8K. We believe further optimization and applying IO-aware principles can further improve CPU performance.

## 5.2   Image Classification

To validate that our methods generalize to images as well as language for non-causal modeling, we next evaluate M2 on image classification. We compare M2 to ViT-style models and recent work, HyenaViT-b [64], which uses gated long convolutions to replace the attention layers in ViT-b. In our work, M2-ViT builds off HyenaViT-b and replaces the long convolutions with the M2 operator in Equation 2 (again setting the Monarch matrices to the DFT and inverse DFT). We replace the MLP blocks in HyenaViT-b with block-diagonal matrices, similarly to M2-BERT. Appendix B additionally compares M2 to the Swin-family of architectures [50, 51].

Table 7 shows the performance of MONARCH MIXER against ViT-b, HyenaViT-b, and ViT-b-Monarch (which replaces the MLP blocks of standard ViT-b with Monarch matrices) on ImageNet-1k. MONARCH MIXER outperforms the other models with only half the parameters of the original ViT-s model. Surprisingly, MONARCH MIXER also outperforms ResNet-152, with fewer parameters—even though the latter was explicitly designed for ImageNet performance.

## 5.3   Causal Language Modeling

GPT-style causal language modeling is a critical application for Transformers [6, 31, 43]. We introduce M2-GPT, a M2-based architecture for causal language modeling. For the sequence mixer, M2-GPT combines the convolutional filter from Hyena [64], the state-of-the-art attention-free

Table 5: Throughput in tokens/ms by context length for M2-BERT-base (80M) compared to BERT-base.

| Model | 512 | 1024 | 2048 | 4096 | 8192 |
|---|---|---|---|---|---|
| HF BERT-base (110M) | 206.1 | 130.8 | 71.3 | 39.0 | OOM |
| FlashAttention BERT-base (110M) | 367.4 | 350.1 | 257.2 | 179.1 | 102.4 |
| M2-BERT-base (80M) | **386.3** | **380.7** | **378.9** | **353.9** | **320.1** |
| M2 Speedup over HF BERT-base (110M) | 1.9× | 2.9× | 5.2× | 9.1× | – |

Table 6: CPU inference latency in milliseconds with a batch size of 1 at varied input sequence lengths. Measurements averaged over 10 examples on a 48 vCPU, 96 GB RAM instance from the GCP n2-standard-48 series, which runs Intel Cascade Lake processors. This is based on the protocol in [29].

| Model | 512 | 1024 | 2048 | 4096 | 8192 |
|---|---|---|---|---|---|
| BERT-base (110M) | **182** | 389 | 918 | 2660 | 11820 |
| M2-BERT-base (80M) | 289 | **361** | **651** | **948** | **1820** |
| Speedup | 0.6× | 1.1× | 1.4× | 2.8× | 6.5× |

language model, with parameter sharing across multiple heads from H3 [27]. We use the causal parameterization of Equation 2 to replace the FFT in these architectures, and we remove the MLP layers entirely. The resulting architecture is entirely attention- and MLP-free.

We pretrain M2-GPT on the PILE, a standard dataset for causal language modeling. Following prior work [28, 64], we train models at two model sizes, with varying amounts of training data—decaying the learning rate appropriately for each experiment. Table 8 shows the results. Even though our model is attention- and MLP-free, it outperforms both Transformers and Hyena in perplexity on pretraining. These results suggest that radically different architectures than Transformers may be performant on causal language modeling.

## 6 Related Work

**Long Convolutions** Recent work proposes to use long convolution layers as a replacement for the Transformer attention layers in sequence modeling [28, 64, 67–69]. Many of these models rely on the FFT convolution theorem to compute the long convolutions. We build on the insights in many of these architectures in constructing our M2 architectures, and additionally replaces the FFT operations with Monarch matrices.

Our work is also related to a rich literature in convolutions in other bases, such as Chebyshev bases [82] or orthogonal polynomial bases [34]. These approaches have analogues in our multivariate analysis; replacing the basis polynomials of the Monarch matrices in MONARCH MIXER may be able to approximate some of these operations. An interesting question for future work would be to study how well our techniques and concerns about causality and hardware utilization translate to these alternative convolution bases.

**Optimization of deep learning primitives** There is a rich history of the optimization of deep learning primitives, as accelerating their performance can yield substantial savings in compute and cost for large models. There are many approaches to speed up these operations, but they usually either reduce data movement or compute.

Reducing data movement: In many applications, the major bottleneck is the storage and movement of large amounts of memory. One popular approach to reducing data movement is checkpointing, wherein one stores fewer intermediate results and recomputes the others on-the-fly where they are needed, trading additional compute for memory [46, 78]. Another approach is kernel fusion, wherein algorithms initially described as sequential steps can often be fused in ways that improve their properties. For example, it is generally faster to implement a dot-product through a multiply-

Table 7: Accuracy on ImageNet-1k. ResNet-152 provided for reference.

| Model | Top-1% | Top-5% | Description |
|---|---|---|---|
| ResNet-152 (60M) | 78.6 | 94.3 | ConvNet, MLP |
| ViT-b (87M) | 78.5 | 93.6 | Attention, MLP |
| ViT-b + Monarch (33M) | 78.9 | 94.2 | Attention, MLP-Free |
| HyenaViT-b (88M) | 78.5 | 93.6 | Attention-Free, MLP |
| M2-ViT-b (45M) | **79.5** | **94.5** | Attention-Free, MLP-Free |

Table 8: Perplexity on the PILE when trained for different numbers of tokens.

| Model | 5B | 10B | 15B | Description |
|---|---|---|---|---|
| Transformer (125M) | 13.3 | 11.9 | 11.2 | Attention, MLP |
| Hyena (155M) | 13.1 | 11.8 | 11.1 | Attention-Free, MLP |
| M2-GPT (145M) | **12.9** | **11.6** | **10.9** | Attention-Free, MLP-Free |
| Transformer (355M) | 11.4 | 9.8 | 9.1 | Attention, MLP |
| Hyena (360M) | 11.3 | 9.8 | 9.2 | Attention-Free, MLP |
| M2-GPT (360M) | **11.0** | **9.6** | **9.0** | Attention-Free, MLP-Free |

accumulate rather than first multiplying and then accumulating. Recently, libraries such as PyTorch 2.0 [63] have added kernel fusion capabilities, although the very best performance usually still arises from handwritten kernels. Third, in order to better exploit memory locality, it is often fastest to load small blocks of memory, do intensive computation on them, and then write the results a tile at a time [83]. Finally, many algorithms also have hand-optimizations that can remove unnecessary computation or memory accesses [55].

Efficient algorithms usually make use of a combination of these techniques. For example, FlashAttention [15] uses all four to dramatically decrease both the latency and memory consumption of multi-head attention. Though we have made a modest effort to implement MONARCH MIXER efficiently, we think it likely that MONARCH MIXER could be further optimized by these techniques.

Reducing flops: A first target for optimization is the multi-layer perceptron (MLP), owing to its ubiquity. A variety of structured sparse factorizations exist, many of which we draw on in this work [7, 11, 14, 16, 17, 19, 26, 91]. Attention is also a popular target for optimization. Recently, a plethora of sub-quadratic approximations of attention have emerged, that aim to approximate attention to reduce its quadratic complexity. Some methods rely on sparsification, relying on the fact that the attention matrix is extremely sparse at long sequence lengths [2, 23, 24, 42, 53]. Others use low-rank approximations of the attention matrix [13, 79, 91] or kernel methods instead [9, 41]. A subset use a combination of these techniques, such as [8, 72]. Finally, a third category of methods [27, 64] aim to replace attention entirely, relying on state-space models [33].

## 7  Discussion and Conclusion

We explore MONARCH MIXER (M2), a new architecture that is sub-quadratic in both sequence length and model dimension and is hardware-efficient on modern accelerators. We motivate M2 from both theoretical and systems performance perspectives and conduct a preliminary proof-of-concept investigation into performance on masked language modeling, image classification, and causal language modeling.

While our initial results are promising, our work is only a first step in this direction. The M2 layer can likely be further optimized with systems optimization techniques such as kernel fusion. Our work has also not been optimized for inference like more well-established models such as Transformers, or even more recent models such as state space models. It also remains to be seen whether M2 layers can have as widespread applicability as Transformers. We hope that these can be fruitful directions for future work.

A discussion of broader impacts can be found in the Appendix.

## Acknowledgments

We gratefully acknowledge the support of DARPA under Nos. FA86501827865 (SDH) and FA86501827882 (ASED); NIH under No. U54EB020405 (Mobilize), NSF under Nos. CCF1763315 (Beyond Sparsity), CCF1563078 (Volume to Velocity), and 1937301 (RTML); ONR under No. N000141712266 (Unifying Weak Supervision); the Moore Foundation, NXP, Xilinx, LETI-CEA, Intel, IBM, Microsoft, NEC, Toshiba, TSMC, ARM, Hitachi, BASF, Accenture, Ericsson, Qualcomm, Analog Devices, the Okawa Foundation, American Family Insurance, Google Cloud, Microsoft Azure, Swiss Re, Brown Institute for Media Innovation, Department of Defense (DoD) through the National Defense Science and Engineering Graduate Fellowship (NDSEG) Program, Fannie and John Hertz Foundation, National Science Foundation Graduate Research Fellowship Program, Texas Instruments Stanford Graduate Fellowship in Science and Engineering, and members of the Stanford DAWN project: Teradata, Facebook, Google, Ant Financial, NEC, VMWare, and Infosys. The U.S. Government is authorized to reproduce and distribute reprints for Governmental purposes notwithstanding any copyright notation thereon. Any opinions, findings, and conclusions or recommendations expressed in this material are those of the authors and do not necessarily reflect the views, policies, or endorsements, either expressed or implied, of DARPA, NIH, ONR, or the U.S. Government. JG and AR's work is supported by NSF grant# CCF-2247014. IJ's work is supported by an NSF Graduate Fellowship.

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

## Author Contributions

| | |
|---|---|
| **D.Y.F.** | Conceptualized the research; coordinated collaborations; developed M2 architectures; led experimental and implementation efforts; assisted in development of theoretical results; coordinated writing. |
| **S.A.** | Worked on developing M2 architectures; worked on implementing and conducting BERT experiments; worked on implementing and performing CPU experiments; assisted in writing and framing the work. |
| **J.G.** | Led development of theory and causal algorithms; wrote Appendix D. |
| **I.J.** | Led development of theory and causal algorithms; wrote Appendix D. |
| **S.E.** | Worked on writing and framing the work; assisted in development of M2 architectures; assisted in the optimized M2 implementation; conducted mixer benchmarks; assisted with BERT experiments; conducted Swin ImageNet experiments. |
| **A.W.T.** | Conducted ViT experiments; assisted in writing. |
| **B.S.** | Assisted in optimized M2 implementation; conducted mixer benchmarks; assisted in writing. |
| **M.P.** | Assisted in development of M2-GPT architecture. |
| **A.R.** | Supervised theory development; developed proofs; reviewed manuscript. |
| **C.R.** | Supervised research; reviewed manuscript. |

*Simran Arora, Jessica Grogan, Isys Johnson, Sabri Eyuboglu, and Armin Thomas contributed equally to this work.*

## Appendix

Appendix A discusses broader impacts of our work. Appendix B presents additional experiments. Appendix C gives details for the experiments, including model architectures and hyperparameters. Appendix D gives missing details and proofs for the theoretical analysis, as well as generalizations to broader results. Appendix E gives a PyTorch code listing of an M2 layer.

## A  Broader Impacts

Our work seeks to understand the fundamental capabilities and limitations of newly-emerging model architectures. As the amount of data and model size grows, we also seek to understand how to make training these models more efficient, both in terms of the amount of training context and the model size. This potentially connects to energy savings during model development and deployment, as well as making machine learning models accessible to a larger population of people.

However, as with any machine learning models, developing new techniques may impact a wide range of applications, each with potential benefits and harms. Making language model training cheaper and longer context may make it cheaper to spread disinformation. Similarly, improving the efficiency of model training may not reduce the overall environmental footprint of training, since the same resources may be used to train more models, or train the same models for longer. While our work makes partial progress on the fronts of efficiency and understanding, it does not explicitly address the issues of fairness and bias in language models. In addition, our work demonstrates a proof-of-concept; it has error modes, and we recognize the inherent risks of training and using machine learning models, including language models. Detailed discussions of these risks are in [3, 5, 80].

## B  Additional Experiments

### B.1  Per-Task GLUE Numbers

We report full GLUE numbers for M2-BERT-base and M2-BERT-large in Table 9.

### B.2  Additional Throughput Results

We report the throughput of M2-BERT-base (80M) compared to BERT models of the same size (BERT-base with fewer parameters), as well as the throughput of M2-BERT-large (260M) compared to BERT-large.

Table 9: Fine-tuning performance on GLUE [76]. We report the standard metrics – F1 scores for QQP and MRPC, Matthew's correlation for CoLA, Spearman's correlation for STS-B, and accuracy for the remaining tasks, following the procedure from [38].

| Model | MNLI (m / mm) | RTE | QNLI | QQP | SST2 | STS-B | CoLA | MRPC | **Average** |
|---|---|---|---|---|---|---|---|---|---|
| M2-BERT-base (80M) | 78.4 / 78.6 | 68.5 | 84.6 | 86.7 | 92.0 | 86.3 | 53.0 | 89.8 | 79.9 |
| M2-BERT-base (110M) | 79.6 / 80.5 | 69.3 | 86.0 | 87.0 | 92.3 | 86.9 | 56.0 | 89.2 | 80.9 |
| M2-BERT-large (260M) | 81.7 / 81.9 | 72.8 | 84.7 | 87.8 | 93.3 | 88.0 | 59.2 | 90.0 | 82.2 |
| M2-BERT-large (341M) | 82.2 / 82.3 | 75.0 | 87.0 | 87.7 | 92.4 | 88.3 | 59.6 | 90.1 | 82.8 |

Table 10: Throughput in tokens/ms by context length for M2-BERT-base (80M) compared to 80M BERT models.

| **Model** | **512** | **1024** | **2048** | **4096** | **8192** |
|---|---|---|---|---|---|
| HF BERT (79M) | 248.4 | 157.3 | 86.0 | 46.8 | OOM |
| FlashAttention BERT (79M) | **433.3** | **425.1** | 335.2 | 217.4 | 122.6 |
| M2-BERT-base (80M) | 386.3 | 380.7 | **378.9** | **353.9** | **320.1** |
| M2 Speedup over HF BERT (80M) | 1.6× | 2.4× | 4.4× | 7.5× | – |

Table 10 compares the performance of M2-BERT-base (80M) to BERT models parameter-matched to 80M parameters. M2 is slower than FlashAttention for sequence lengths 512 and 1K, but outperforms FlashAttention starting at sequence length 2K. We believe further optimization of the M2 kernel can close the gap to FlashAttention for short sequences.

Table 11 compares M2-BERT-large (260M) to BERT-large. Trends are mostly similar to comparisons against BERT-base; M2 nearly matches FlashAttention at sequence length 512, and outperforms it for sequence length 1K and longer. We also see up to 4.3× speedup over HuggingFace BERT-large at sequence length 2K.

## B.3 ImageNet Comparison against Swin

Table 12 reports the results of replacing attention and MLP in Swin-V2 using M2 as a drop-in replacement. Surprisingly, Swin-M2 outperforms Swin-MLP-B, is competitive with Swin-V1-B, and comes within 1 point of Swin-V2-B, even without any hyperparameter tuning or architecture adjustment from the ViT formula. We expect that performance may improve further with hyperparameter tuning specific to M2.

## B.4 Speech Applications

Table 13 presents the performance of M2 on Speech Commands-10, a speech classification task over raw 1-second clips sampled at 16 kHz. M2 is competitive with state-of-the-art architectures on this task.

## B.5 CIFAR10

Table 14 shows the performance of MONARCH MIXER on CIFAR10. The trends are largely the same as on ImageNet.

## B.6 Learnable Monarch Matrices in Sequence Mixer

In most of our models, we have used fixed Monarch matrices for the sequence mixer, and learnable Monarch matrices for the dimension mixer. Table 15 presents an experiment evaluating using learnable Monarch matrices for the sequence mixer on the sequential CIFAR task. We use a non-gated convolutional architecture based off long convolutions, as presented in [28]. Learning the Monarch matrices in the sequence mixer yields 1.5 points of lift.

Table 11: Throughput in tokens/ms by context length for M2-BERT-large (260M) compared to BERT-large.

| Model | 512 | 1024 | 2048 | 4096 | 8192 |
|---|---|---|---|---|---|
| HF BERT-large (340M) | 75.4 | 47.1 | 25.2 | OOM | OOM |
| FlashAttention BERT-large (340M) | **125.0** | 111.9 | 91.6 | 54.5 | OOM |
| M2-BERT-large (260M) | 122.5 | **118.6** | **109.4** | **94.5** | **75.0** |
| M2 Speedup over HF BERT-large (340M) | 1.6× | 2.5× | 4.3× | - | - |

Table 12: ImageNet accuracy of Swin models.

| Model | ImageNet (acc@1) | ImageNet (acc@5) |
|---|---|---|
| Swin-MLP-B | 81.3 | 95.3 |
| Swin-V1-B | 83.5 | 96.5 |
| Swin-V2-B | **84.2** | **96.9** |
| M2-Swin-B | 83.5 | 96.7 |

## B.7 Roofline Analysis

Figure 4 shows a Roofline analysis of a simple PyTorch implementation of a single M2 operator $\mathbf{M}^{-1}(\mathbf{Mu} \odot \mathbf{Mk}$ on an A100 GPU, with 4K input length. The operation is more dominated by the data movement operations, which helps explain why performance is higher on newer architectures like RTX 4090 (which have faster and larger L2 cache).

## B.8 Associative Recall

In Table 16, we present a simple experiment demonstrating the causal parameterization of M2 on associative recall, a synthetic language designed to test in-context learning. The model demonstrates in-context learning abilities in sequences up to 128K tokens, but Transformers do not scale past 8K.

## B.9 BERT Experiments with Alternative Architecture

Here, we report results using an older version of the M2-BERT architecture, that uses non-gated convolutions and is trained on English Wikipedia [25] and English Bookcorpus [92]. For clarity, we refer to this model as M1-BERT.

We found that M1-BERT could match Transformers on MLM quality, but underperformed on down-stream fine-tuning. We attribute this gap in performance to sub-optimal training hyperparameters (optimized for throughput using NVIDIA MLPerf hyperparameters) as well as a sub-optimal architecture. We report results here for completeness, but refer to the gated convolution architecture in the main body as the proper M2-BERT model.

These models followed the reference implementations and hyperparameters from Hugging Face Transformers examples [81] and Nvidia Deep Learning examples (`https://github.com/NVIDIA/DeepLearningExamples`). In particular, we use the LAMB optimizer with a learning rate of $5e - 3$. For each sequence length, we use as large a minibatch size as possible that fits on the GPU (A100-80GB in Table 17 and V100 in Table 18). We set the gradient accumulation to reach a global batch size of $65, 536$ sequences. To investigate the effect of sequence length, each model is trained for a fixed sequence length in a single phase of training (in contrast to some training protocols, which train the model in multiple phases, each at different sequence lengths).

**Time to a Fixed Pretraining Quality on 8xA100** We compare time to a fixed pretraining quality, training M1-BERT-base on English Wikipedia [25] and English Bookcorpus [92]. We compare against BERT-base trained with FLASHATTENTION [15], as well as the Monarch-BERT-base implementation from the original Monarch paper [14]. We measure wall-clock time for M1-BERT and the base Transformer to reach 50% in masked language modeling accuracy on 8xA100 Nvidia GPUs with 80GB memory each. Table 17 summarizes results. In short sequence lengths, M1-BERT is

Table 13: Accuracy on Speech-Commands 10. An "x" means that the model did not fit in memory.

| M2 | S4 | WaveGan-D | Transformer | Performer | CKConv |
|---|---|---|---|---|---|
| **97.9** | 97.5 | 96.3 | x | 30.8 | 71.7 |

Table 14: Accuracy on CIFAR-10.

| Model | Top-1% | Description |
|---|---|---|
| ViT (1.2M) | 78.6 | Attention + MLP |
| ViT + Monarch (607K) | 79.0 | Attention, MLP-Free |
| HyenaViT (1.3M) | 80.6 | Attention-Free + MLP |
| HyenaViT-M2 (741K) | **80.8** | Attention-Free + MLP Free |

comparable to FLASHATTENTION, even without using a heavily-optimized fused kernel. In longer sequence lengths, the FLOP savings make M1-BERT more efficient—up to $2.4\times$ faster than BERT with FLASHATTENTION at sequence length 4096.

**BERT in Half a Day** Inspired by recent work focusing on training under limited resource constraints [30], we measure how far we can get when training on a single V100 GPU in 12 hours. In Table 18, we report the masked language modeling accuracy achieved by the same set of models and sequence lengths (except for the FLASHATTENTION baseline, which is not supported on V100). We observe M1-BERT both achieves higher accuracy within the time limit and can be trained at longer sequence lengths than the baseline architectures.

**Downstream Fine-Tuning** We evaluate the quality of M1-BERT-base models on the GLUE benchmark [76]. Table 19 shows fine-tuning performance on the GLUE tasks, using the same hyperparameters and 5 epochs for all tasks and both models. M1-BERT-base is competitive with Transformers trained using MLPerf hyperparameters on Bookcorpus and Wikitext, but underperforms fully-trained transformers and M2-BERT-base.

## C  Experiment Details

### C.1  Model Architectures

In this section, we describe the exact model architectures we used for each task, including the design of the block (residuals and gating). We additionally release our code for reproducibility,

**BERT Language Modeling** The M2-BERT architectures use a standard BERT backbone, but replace the attention with bidirectional gated convolutions and replace the linear layers in the MLPs with block-diagonal matrices. All the M2-BERT architectures use an expansion factor of four. M2-BERT-base (80M) has a model width of 768 and 12 layers; M2-BERT-base (110M) has a model width of 960 and 12 layers; M2-BERT-large (260M) has a model width of 1536 and 12 layers; and M2-BERT-large (341M) has a model width of 1792 and 12 layers. We train all these models on C4 for 70,000 steps, with sequence length 128, and global batch size 4096 sequences. For all the models, we use decoupled AdamW with learning rate 8e-4 and decoupled weight decay 1e-5. We use linear learning rate decay with a warmup of 6% of the steps, and we use MLM masking percentage of 30%.

For GLUE fine-tuning, we do a small search of learning rate, weight decay, and number of epochs. Following [38], we fine-tune RTE, MRPC, and STS-B from the MNLI checkpoint. We fine-tune all tasks with sequence length 128. For some tasks, we also pool the embeddings of all the non-padding tokens instead of using the CLS token.

The final hyperparameters for M2-BERT-base (80M) are decoupled AdamW with learning rate 5e-5 and weight decay 5e-6 for 3 epochs for MNLI; AdamW with learning rate 5e-5 and weight decay 0.01 for 6 epochs for RTE; AdamW with learning rate 3e-5 and weight decay 0.01 for 10 epochs on QQP; AdamW with learning rate 5e-5 and weight decay 1e-5 for 10 epochs with average pooling for QNLI; decoupled AdamW with learning rate 3e-5 and weight decay 3ed-6 for 3 epochs for SST-2; AdamW

Table 15: Accuracy on sequential CIFAR for fixed vs. learnable Monarch in the sequence mixer.

| Model | sCIFAR Accuracy |
|---|---|
| M2, Fixed Monarch | 91.0 |
| M2, Learnable Monarch | **92.5** |

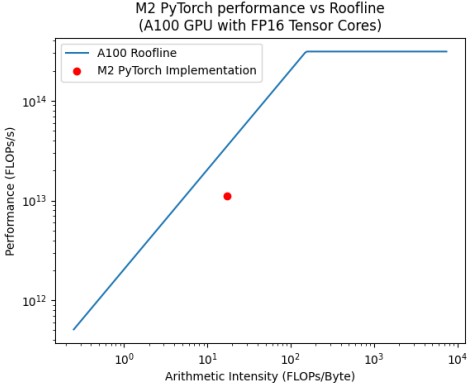

Figure 4: Roofline plot of a PyTorch implementation of a single M2 operator $\mathbf{M}^{-1}(\mathbf{Mu} \odot \mathbf{Mk})$.

with learning rate 7e-5 and weight decay 0.01 for 10 epochs for STS-B; AdamW with learning rate 5e-5 and weight decay 0.01 for 10 epochs for MRPC; and decoupled AdamW with learning rate 5e-5 and weight decay 5e-6 for 10 epochs for COLA.

For M2-BERT-base (110M), the hyperparameters are decoupled AdamW with learning rate 5e-5 and weight decay 5e-6 for 3 epochs for MNLI; decoupled AdamW with learning rate 1e-5 and weight decay 1e-6 for 3 epochs for RTE; decoupled AdamW with learning rate 3e-5 and weight decay 3e-6 for 5 epochs on QQP; decoupled AdamW with learning rate 5e-5 and weight decay 1e-5 for 10 epochs with average pooling for QNLI; decoupled AdamW with learning rate 3e-5 and weight decay 3ed-6 for 3 epochs for SST-2; decoupled AdamW with learning rate 8e-5 and weight decay 3e-6 for 10 epochs for STS-B; decoupled AdamW with learning rate 8e-5 and weight decay 8e-5 for 10 epochs for MRPC; and AdamW with learning rate 8e-5 and weight decay 5e-6 for 10 epochs for COLA.

For M2-BERT-large (260M), the hyperparameters are decoupled AdamW with learning rate 5e-5 and weight decay 5e-6 for 3 epochs for MNLI; decoupled AdamW with learning rate 1e-5 and weight decay 1e-6 for 3 epochs for RTE; decoupled AdamW with learning rate 3e-5 and weight decay 3e-6 for 5 epochs on QQP; decoupled AdamW with learning rate 5e-5 and weight decay 1e-5 for 10 epochs for QNLI; decoupled AdamW with learning rate 3e-5 and weight decay 3ed-6 for 3 epochs for SST-2; decoupled AdamW with learning rate 7e-5 and weight decay 3e-6 for 10 epochs for STS-B; decoupled AdamW with learning rate 8e-5 and weight decay 8e-6 for 10 epochs for MRPC; and AdamW with learning rate 5e-5 and weight decay 5e-6 for 10 epochs for COLA.

For M2-BERT-large (341M), the hyperparameters are decoupled AdamW with learning rate 5e-5 and weight decay 5e-6 for 3 epochs for MNLI; AdamW with learning rate 5e-5 and weight decay 1e-6 for 2 epochs for RTE; decoupled AdamW with learning rate 3e-5 and weight decay 3e-6 for 5 epochs on QQP; decoupled AdamW with learning rate 5e-5 and weight decay 1e-6 for 10 epochs for QNLI; decoupled AdamW with learning rate 3e-5 and weight decay 3ed-6 for 3 epochs for SST-2; decoupled AdamW with learning rate 8e-5 and weight decay 3e-5 for 8 epochs for STS-B; decoupled AdamW with learning rate 8e-5 and weight decay 8e-6 for 10 epochs for MRPC; and decoupled AdamW with learning rate 5e-5 and weight decay 1e-6 for 10 epochs for COLA.

**ViT** We use a standard ViT model architecture as base [22]. In line with recent improvements to the ViT architecture [4, 70, 86], we use sinusoidal position embeddings and global average-pooling (GAP) instead of a class token.

Table 16: In-context learning performance on associative recall at various sequence lengths, vocab size 20. ✗ indicates the Transformer did not finish in a week.

| Model | 0.5K | 2K | 8K | 32K | 128K |
|---|---|---|---|---|---|
| Transformer | 100.0 | 100.0 | 100.0 | ✗ | ✗ |
| MONARCH MIXER | 98.7 | 99.4 | 99.4 | 99.4 | 99.4 |

Table 17: Time in hours to reach 50% masked language modeling validation accuracy on 8xA100 with different sequence lengths.

| Model | 512 | 1024 | 2048 | 4096 | Architecture Details |
|---|---|---|---|---|---|
| BERT-base-FLASHATTENTION (110M) | 2.7 | 3.8 | 5.7 | 13.2 | Attention, MLP |
| BERT-base-HuggingFace (110M) | 3.3 | 5.6 | 13.1 | 26.7 | Attention, MLP |
| BERT-Monarch-base (80M) | 3.1 | 4.7 | 10.3 | 22.1 | Attention, MLP-free |
| M1-BERT-base (55M) | 2.5 | 3.5 | 4.0 | 5.5 | Attention-Free, MLP-free |
| Speedup | $1.1\times$ | $1.1\times$ | $1.3\times$ | $2.4\times$ | |

We adapt the ViT architecture by replacing its MLP and/or attention components with Monarch Matrices (similar to our adaptation of BERT):

We replace the MLP with randomly initialized Monarch Matrices of the same dimension as the dense matrices of the MLP and learn those matrices during training, setting the number of blocks in the block-diagonal matrices to 4.

We replace attention with the recently introduced Hyena operator [64]. The Hyena operator represents a recurrence of two efficient sub-quadratic primitives, an implicit long convolution and multiplicative element-wise gating of the projected input. Hyena operators apply the FFT algorithm to achieve fast long convolutions in sub-quadratic time. We further adapt the Hyena operator by replacing its long convolutions with the M2 operator and setting the Monarch Matrices to the DFT and inverse DFT.

**ViT for ImageNet-1k**   In line with other work [4, 14, 64, 70], we use a ViT-base architecture with 12 layers, a hidden size of 768, 12 attention heads per layer, an intermediate size of the MLP projection of $3,072$, and a patch size of $16 \times 16$ pixels. For optimization, we follow the training procedure of T2T-ViT [86], including augmentations such as RandAugment [12] (magnitude $=$ $9$, magnitude-std $= 0.5$, layers $= 2$), Mixup [88] ($\alpha = 0.8$), CutMix [87] ($\alpha = 1.0$), Random erasing [90] (rate $= 0.25$), and AugMix [37]. See Table 20 for all other training settings.

**ViT for CIFAR-10**   We use a ViT architecture with 6 layers, a hidden size of 128, 8 attention heads per layer, an intermediate size of the MLP projection of $512$, and a patch size of $4 \times 4$ pixels. We further tune weight decay ($0$ or $0.1$), stochastic depth rate ($0$ or $0.1$), and base learning rate ($1e-4$ or $3e-4$ or $1e-3$) and report the test performance for the model variant that achieved the highest accuracy in a separate held-out validation dataset (randomly selected $10\%$ of training data). We also apply an early stopping rule such that training is stopped if the model's validation loss does not improve for 10 training epochs. See Table 20 for all other training settings.

**GPT Causal Language Modeling**   Similarly to our ViT approach, we also replace attention with the Hyena operator, using the same architecture as in [64] as a starting point. The Hyena architecture has two convolutions, which can be computed using the FFT convolution theorem. In our architecture, we additionally replace these FFT operations with causal Monarch matrices.

In addition, we re-use the heads extension from the H3 architecture [27]. The heads extension groups the model dimension into heads, ties together the long convolution parameters in each head, and then computes the outer product between different input projections. An algorithmic listing adapted from the H3 paper [27] is provided in Listing 1, with updates to replace the SSM layers with Hyena convolutions. We use a head dimension of 16. Setting the head dimension to be 1 and replacing the Monarch matrices with FFT is equivalent to the Hyena layer.

Table 18: Masked language modeling validation accuracy achieved on a single V100 in 12 hours with different sequence lengths. ✗ indicates the model does not fit on device with a batch size of 1.

| Model | 512 | 1024 | 2048 | 4096 | 8192 | Architecture Details |
|---|---|---|---|---|---|---|
| BERT-base (110M) | 11.5 | 7.8 | 6.8 | ✗ | ✗ | Attention, MLP |
| BERT-Monarch-base | 6.9 | 8.5 | 6.8 | ✗ | ✗ | Attention, MLP-Free |
| M1-BERT-base | 20.2 | 20.2 | 20.1 | 17.1 | 12.9 | Attention-Free, MLP-Free |

Table 19: Fine-tuning performance on the GLUE benchmark [76], after pretraining on Wikipedia and Bookcorpus. We report the standard metrics – F1 scores for QQP and MRPC, Matthew's correlation for CoLA, Spearman's correlation for STS-B, and accuracy for the remaining tasks [21].

| Model | MNLI (m / mm) | RTE | QNLI | QQP | SST2 | STS-B | CoLA | MRPC | Architecture Details |
|---|---|---|---|---|---|---|---|---|---|
| BERT no pretrain | 34.1 / 34.1 | 47.3 | 50.0 | 68.6 | 79.9 | 17.8 | 0.0 | **77.9** | Attention, MLP |
| BERT-base | **74.5 / 74.7** | **55.6** | 69.3 | **81.8** | 83.9 | 19.8 | 12.1 | 74.2 | Attention, MLP |
| M1-BERT-base | 69.9 / 70.5 | 53.1 | **73.2** | 81.4 | **85.2** | **68.1** | **33.6** | 75.4 | Attention-free, MLP-free |

---

**Algorithm 1** M2 Hyena Layer with Heads

**Input:** Input sequence $u \in \mathbb{R}^{N \times d}$ from the previous layer, weight matrices $\mathbf{W}_{X1}, \mathbf{W}_{X2}, \mathbf{W}_V, \mathbf{W}_O \in \mathbb{R}^{d \times d}$, causal Monarch matrix $\mathbf{M}$, short convolution kernels $\mathbf{K}_1, \mathbf{K}_2, \mathbf{K}_3$, a Hyena convolution kernel $\mathbf{K}_{\text{long}}$, head dimension $d_h$.

**Output:** Output sequence $y \in \mathbb{R}^{N \times d}$

Compute $\mathbf{X}_1 = u\mathbf{W}_{X1}, \mathbf{X}_2 = u\mathbf{W}_{X2}, \mathbf{V} = u\mathbf{W}_V \in \mathbb{R}^{N \times d}$.

Pass $\mathbf{X}_1, \mathbf{X}_2, \mathbf{V}$ each through the short convolution using the causal Monarch matrices: $\overline{\mathbf{X}}_1, \overline{\mathbf{X}}_2, \overline{\mathbf{V}} = \mathbf{M}^{-1}(\mathbf{M}\mathbf{X}_1 \odot \mathbf{M}\mathbf{K}_1), \mathbf{M}^{-1}(\mathbf{M}\mathbf{X}_2 \odot \mathbf{M}\mathbf{K}_2), \mathbf{M}^{-1}(\mathbf{M}\mathbf{V} \odot \mathbf{M}\mathbf{K}_3)$.

Split $\overline{\mathbf{X}}_1, \overline{\mathbf{X}}_2, \overline{\mathbf{V}}$ into $H$ "heads" $(\overline{\mathbf{X}}_1^{(h)}, \overline{\mathbf{X}}_2^{(h)}, \overline{\mathbf{V}}^{(h)}$ for $h = 1, \ldots, H)$, each a sequence of $N$ vectors of size $d_h = d/H$.

**for** $1 \leq h \leq H$ **do**

Take the batched outer product $\overline{\mathbf{X}}_2^{(h)}(\overline{\mathbf{V}}^{(h)})^\top \in \mathbb{R}^{N \times d_h \times d_h}$ (batched in the $N$-dimension) and pass it through the long convolution using the causal Monarch: $\mathbf{XV}^{(h)} = \mathbf{M}^{-1}(\mathbf{M}\overline{\mathbf{X}}_2^{(h)}(\overline{\mathbf{V}}^{(h)})^\top \odot \mathbf{M}\mathbf{K}_{\text{long}}) \in \mathbb{R}^{N \times d_h \times d_h}$.

Batch-multiply by $\overline{\mathbf{X}}_1$: $\mathbf{O}^{(h)} = [\overline{\mathbf{X}}_{11}^{(h)}\mathbf{XV}_1^{(h)}, \ldots, \overline{\mathbf{X}}_{1N}^{(h)}\mathbf{XV}_N^{(h)}] \in \mathbb{R}^{N \times d_h}$ (batched in the $N$-dimension).

Concatenate the output $\mathbf{O}^{(h)}$ of each head, and multiply by the output projection matrix $\mathbf{W}_O \in \mathbb{R}^{d \times d}$.

---

Finally, we remove the MLP layers entirely (equivalent to replacing the layer with an identity), and make the model wider to compensate (the depths match the equivalent Hyena models). The small model has a model width of 1160 with 18 layers and uses a learning rate of 0.0006, and the medium model has model width of 1344 with 40 layers and uses a learning rate of 0.0008. All other hyperparameters match the Hyena models [64].

# D  Missing details from Section 4

This section contains all the missing details (including proofs) from Section 4.

In Appendix D.1, we review some definitions and results on multi-variate polynomials and set some notation needed for this section. In Appendix D.2, we explicitly connect Monarch matrices for $p = 2$ and bivariate polynomial evaluation. Specifically, we prove Theorem 1 and Theorem 2. Then in Appendix D.3 we show how to instantiate the bivariate basis polynomials so that we get a causal map. This includes converting the bivariate polynomials to univariate polynomials (with evaluations over the $N$th roots of unity) and this proves Theorem 3. We then show how this causal map can be implemented only using GEMMs (and $O(N^{3/2})$ FLOPs) in Appendix D.4.

Next, we note that while our evaluations points are over complex numbers, our input and output to the Monarch convolution layers are over reals. Hence, it is natural to wonder if we can implement the entire layer just with operations over real numbers. One potential advantage of this is that we theoretically only have to keep $N$ real numbers for intermediate results (instead of $2N$ reals

Table 20: ViT training settings.

|  | ImageNet-1k | CIFAR-10 |
|---|---|---|
| Optimizer | AdamW | |
| Optimizer momentum | $\beta_1, \beta_2 = 0.9, 0.999$ | |
| Learning rate schedule | Cosine decay w/ linear warmup | |
| Dropout rate | 0 | |
| Label smoothing | 0.1 | |
| Image size | 224 x 224 | 32 x 32 |
| Base learning rate | 1e-3 | {1e-4, 3e-4, 1e-3} |
| Batch size | 1024 | 512 |
| Training epochs | 300 | up to 500 |
| Warmup epochs | 10 | 5 |
| Stochastic depth rate | 0.1 | {0, 0.1} |
| Weight decay | 0.05 | {0, 0.1} |

numbers when we keep track of vectors in $\mathbb{C}^N$). This can reduce the data movement costs. Further, multiplication of two complex numbers requires six operations over real numbers (four multiplication and two addition). Thus, moving to an implementation that only uses real numbers could potentially lead to wall clock time speedup. We propose one such scheme in Appendix D.5 that proves a version of Theorem 3 just over reals by moving to the Chebyshev basis (instead of the standard monomial basis). This creates new technical challenges, which we also address.

Finally, we generalize our results to arbitrary $p \geq 2$ in Appendix D.6. We would like to point out that to get a causal map (in Theorem 17) we need to 'embed' input vectors of size $n$ into vectors of size $N = 2^p \cdot n + O\left(n^{1-1/p}\right)$. For $p = 2$, we avoided the blowup of $2^2 = 4$ with a blowup of 2 instead (via Theorem 3). Whether this is possible to do (i.e. have a blowup of 2 instead of $2^p$) for $p > 2$ is an interesting direction for future work. Further, the matrices that lead to causal map can be represented with $O\left(pN^{2/p}\right)$ parameters while the matrices in Theorem 3 use more parameters. Extending the causal map for $p > 2$ that uses $O\left(N^{1+\frac{1}{p}}\right)$ parameters is an exciting direction for future work.

## D.1 Background and Notation

We collect known facts and definitions about multi-variate polynomials in Appendix D.1.1 and recall some notation from [14] in Appendix D.1.2. These will be needed throughout this appendix section.

### D.1.1 Multi-variate Polynomials

**Basic Definitions** Let $p \geq 1$ be an integer. We recollect some definitions on $p$-variate polynomials (over $\mathbb{R}$) in variables $X_0, \ldots, X_{p-1}$. When $p \in \{1, 2\}$, we will use variables in $\{X, Y, Z\}$ for notational simplicity.

We will use $\mathbf{X}$ to denote the vector of variables $(X_0, \ldots, X_{p-1})$. Further for $\mathbf{j} \in \mathbb{Z}_{\geq 0}^p$, we use the notation

$$\mathbf{X}^{\mathbf{j}} = \prod_{a=0}^{p-1} X_a^{j_a}.$$

$\mathbf{X}^{\mathbf{j}}$ is a (standard basis) *monomial*, where $\mathbf{j} = (j_0, \ldots, j_{p-1})$.

A generic $p$-variate polynomial is defined as (with standard monomial representation)

$$q(\mathbf{X}) = \sum_{\mathbf{j} \in \mathbb{Z}_{\geq 0}^p} q_{\mathbf{j}} \cdot \mathbf{X}^{\mathbf{j}},$$

where the *coefficient* $q_{\mathbf{j}} \in \mathbb{R}$.

We will need the following notion of degrees:

**Definition 2** (Degree). *Let $0 \leq a < p$. The degree of $X_a$ in $\mathbf{X^j}$ (with $\mathbf{j} = (j_0, \ldots, j_{p-1})$) is $j_a$. The degree of $X_a$ of $q(\mathbf{X})$, denoted by $\deg_{X_a}(q)$ is the maximum degree of $X_a$ over all monomials $\mathbf{X^j}$ with $q_{\mathbf{j}} \neq 0$.*

Note that for $p = 1$ the above coincides with the usual notion of degree of a univariate polynomial $q(Z)$, in which case we just use $\deg(q(Z))$ to denote $\deg_Z(q(Z))$.

We will need the notion of taking $\mod$ of a $p$-variate polynomial with $p$-tuple of polynomials. The notion of $\mod$ is well defined for a univariate polynomial (which we will assume as a given below) but in general for arbitrary $p$-variate polynomials $q(\mathbf{X})$ and $q'(\mathbf{X})$, the operation $q(\mathbf{X}) \mod q'(\mathbf{X})$ is not well defined. However, we will only need the following restricted operation:

**Definition 3.** *Let $p \geq 1$. Fix a $p$-tuple of polynomials $R_0(X_0), \ldots, R_{p-1}(X_{p-1})$. Then for any $\mathbf{j} \in \mathbb{Z}_{\geq 0}^p$, we define*

$$\mathbf{X^j} \mod (R_0(X_0), \ldots, R_{p-1}(X_{p-1})) = \prod_{a=0}^{p-1} \left( X^{j_a} \mod (R_a(X_a)) \right).$$

*For a general polynomial $p(\mathbf{X})$,*

$$p(\mathbf{X}) \mod (R_0(X_0), \ldots, R_{p-1}(X_{p-1}))$$

*is defined by extending the definition for $\mathbf{X^j}$ by linearity.*

**Polynomial Evaluation**   Given a $p$-variate polynomial $q(\mathbf{X})$ and an point $\mathbf{a} \in \mathbb{R}^p$, the *evaluation of* $q$ at $\mathbf{a}$ denoted by $q(\mathbf{a})$ is evaluation of $q$ as a function at $\mathbf{a}$.

Given subsets $S_a \subseteq \mathbb{C}$, we define $q(\mathbf{X})$ evaluated at $\times_{a=0}^{p-1} S_a$ as the vector of values $q(\mathbf{a})$ overall $\mathbf{a} \in \times_{a=0}^{p-1} S_a$.

In this paper, we will in many cases evaluate polynomials at the appropriate roots of unity. Specifically for an integer $N$, we will define

$$\omega_N = e^{2\pi\iota/N}$$

and note that the $N$th roots of unity is the set $\{\omega_N^i \mid 0 \leq i < N\}$.

**Polynomial Interpolation**   We now recall univariate and bivariate polynomial interpolation results (proved via the Lagrange basis), which we will use in later subsections.

**Theorem 4.** *Let $D \geq 1$ be an integer. Given $y_i$ for $0 \leq i < D$ and $\alpha_i$ for $0 \leq i < D$ there exists a unique univariate polynomial $P(X)$ with $\deg(P) < D$, such that for all $0 \leq i < D$,*

$$P(\alpha_i) = y_i. \tag{7}$$

*Proof.* This proof is based on the Wikipedia entry for Lagrange polynomials [47].

Given a sequence of values $\alpha_i$ for $0 \leq i < D$ s.t. $\alpha_i \neq \alpha_j$, $i \neq j$, the Lagrange basis for polynomials of degree $< D$ for these values is the set of each polynomials $\{p_0(X), p_1(X), \ldots p_{D-1}(X)\}$ each of degree $D - 1$. Each basis polynomial are defined as:

$$p_i(X) = \frac{X - \alpha_0}{\alpha_i - \alpha_0} \cdots \frac{X - \alpha_{i-1}}{\alpha_i - \alpha_{i-1}} \cdot \frac{X - \alpha_{i+1}}{\alpha_i - \alpha_{i+1}} \cdots \frac{X - \alpha_{D-1}}{\alpha_i - \alpha_{D-1}} = \prod_{\substack{0 \leq j < D \\ j \neq i}} \frac{X - \alpha_j}{\alpha_i - \alpha_j}. \tag{8}$$

By definition,

$$p_i(\alpha_j) = \begin{cases} 1 & \text{for } j = i \\ 0 & \text{otherwise} \end{cases}. \tag{9}$$

The Lagrange interpolating polynomial for those nodes through the corresponding values $y_i$ for $0 \leq i < D$ is the linear combination:

$$P(X) = \sum_{i=0}^{D-1} y_i \cdot p_i(X). \tag{10}$$

By (9), for all $0 \le i < D$:

$$P(\alpha_i) = y_i. \tag{11}$$

Finally, the interpolating polynomial is unique. Assume there is another polynomial $M(X)$ of degree $< D$ such that $M(\alpha_i) = y_i$ for all $0 \le i < D$. Then the difference $M(X) - P(X)$ is 0 at $D$ distinct points $\alpha_i$ for $0 \le i < D$. And the only polynomials of degree $< D$ with more than $D - 1$ roots is the 0 polynomial. So, $M(X) = P(X)$.

$\square$

**Theorem 5.** *Let $D_X, D_Y \ge 1$ be integers. Given values $y_{ij}$ for $0 \le i < D_X, 0 \le j < D_Y$ and $D_X$ distinct points $(\alpha_0, \ldots, \alpha_{D_X-1})$, $D_Y$ distinct points $(\beta_0, \ldots, \beta_{D_Y-1})$ there exists a unique bivariate polynomial $P(X, Y)$ with $\deg_X(P) < D_X$, $\deg_Y(P) < D_Y$, such that for all $0 \le i < D_X$, $0 \le j < D_Y$:*

$$P(\alpha_i, \beta_j) = y_{ij}. \tag{12}$$

*Proof.* Define

$$P(X, Y) = \sum_{j=0}^{m} \sum_{i=0}^{n} y_{ij} \cdot p_i(X) \cdot \bar{p}_j(Y), \tag{13}$$

where $p_i$ and $\bar{p}_j$ are Lagrange basis polynomials defined in the proof of Theorem 4 such that for $0 \le i, k < D_X$,

$$p_i(\alpha_k) = \begin{cases} 1 & \text{for } i = k \\ 0 & \text{otherwise} \end{cases} \tag{14}$$

and for $0 \le j, \ell < D_Y$

$$\bar{p}(\beta_\ell) = \begin{cases} 1 & \text{for } k = \ell \\ 0 & \text{otherwise} \end{cases}. \tag{15}$$

From above, we have for all $i, j, k, \ell$

$$p_i(\alpha_k) \cdot \bar{p}_j(\beta_\ell) = \begin{cases} 1 & \text{for } i = k \text{ and } j = \ell \\ 0 & \text{otherwise} \end{cases}. \tag{16}$$

Then, for all $0 \le i < D_X$, $0 \le j < D_Y$:

$$P(\alpha_i, \beta_j) = y_{ij}. \tag{17}$$

By definition of Lagrange basis polynomials, $\deg_X(P) < D_X$ and $\deg_Y(P) < D_Y$.

Finally, the interpolating polynomial is unique. Assume there is another polynomial $M(X, Y)$ with $\deg_X(M) < D_X$ and $\deg_Y(M) < D_Y$ such that $M(\alpha_i, \beta_j) = y_{ij}$ for all $0 \le i < D_X$ and $0 \le j < D_Y$. Then the difference $M(X, Y) - P(X, Y)$ is 0 at $D_X \cdot D_Y$ distinct points, $(\alpha_i, \beta_j)$ for $0 \le i < D_X, 0 \le j < D_Y$. And the only polynomial with $\deg_X < D_X$ and $\deg_Y < D_Y$ that has $D_X \cdot D_Y$ roots is the 0 polynomial. $\square$

### D.1.2 Notation

Here we recall notation we will use from [14].

1. The class of Monarch matrices is defined in appendix C of [14] as $\mathcal{M}^{(b,N)}$ which are $N \times N$ matrices with block size $b$ for any integer $0 \le b \le N$ that divides $N$. When $b = \sqrt{N}$ we drop $b$ from the notation giving $(i_1, i_0)$ and $(j_1, j_0)$. For example, this is used in Proof of Corollary 2.

2. Row index $i$ can be represented as $(i_1, i_0)_b$. Which gives $i = i_1 b + i_0$.

3. Similarly, column index $j$ can be represented as $(j_1, j_0)_b$. Which gives $j = j_1 b + j_0$. Note that when $b = \sqrt{N}$, $j_1 = k(j)$ and $j_0 = m(j)$. We choose to use the $(j_1, j_0)$ notation here since that notation is easier to generalize for $p > 2$.

4. $\overline{\mathbf{L}} \in \mathcal{DB}^{(b,N)}$ is an $N \times N$ matrix with $b \times b$ blocks that are all diagonal matrices.

5.  $\mathbf{R} \in \mathcal{BD}^{(b,N)}$ meaning it's a block diagonal $N \times N$ matrix with block size $b \times b$.

6.  We have a class of permutation matrices defined as $\sigma_{(b,N)}(i) = i_0 \cdot \frac{N}{b} + i_1$. This can be denoted by an $N \times N$ matrix, $P_{(b,N)}$, where the $i^{\text{th}}$ row is $\mathbf{e}_{\sigma(b,N)(i)}$.

7.  We'll use $i$ or pair notation $(i_1, i_0)_b$ to denote the rows, and $j$ or pair notation $(j_1, j_0)_b$ to denote columns. It should be clear from context which one we're using.

For any $0 \leq j_1 < \sqrt{N}$, let $\ell_{j_1}(X, Y)$ be an arbitrary bivariate polynomial with $\deg_X(\ell_{j_1})$, $\deg_Y(\ell_{j_1}) < \sqrt{N}$.

For any $0 \leq j_1, j_0 < \sqrt{N}$, let $r_{j_1,j_0}(Y)$ be an arbitrary univariate polynomial of degree $< \sqrt{N}$.

Let $A = (\alpha_0, \ldots, \alpha_{\sqrt{N}-1})$, $B = (\beta_0, \ldots, \beta_{\sqrt{N}-1})$ each be a sequence of distinct eval points. Note that $A$ and $B$ need not be disjoint.

From the proof of Theorem 3 in the Appendix C of the Monarch paper [14] we get,

$$\mathbf{L} = \mathbf{P}_{(b,N)} \cdot \overline{\mathbf{L}} \cdot \mathbf{P}_{(b,N)}^\top$$
$$= \mathbf{P}_{(b,N)} \cdot \overline{\mathbf{L}} \cdot \mathbf{P}_{(\frac{N}{b}, N)}.$$

Therefore,

$$\overline{\mathbf{L}} = \mathbf{P}_{(b,N)}^\top \cdot \mathbf{L} \cdot \mathbf{P}_{(b,N)}$$
$$= \mathbf{P}_{(\frac{N}{b}, N)} \cdot \mathbf{L} \cdot \mathbf{P}_{(b,N)}.$$

Define $\mathcal{DB}$ and $\mathcal{BD}$ as set of all such $\overline{\mathbf{L}}$ and $\mathbf{R}$ matrices over $\mathbb{R}^{N \times N}$ where if $i_0 \neq k_0$

$$\overline{\mathbf{L}}_{i_1,k_1}[i_0, k_0] \stackrel{\text{def}}{=} \overline{\mathbf{L}}[(i_1, i_0)_{\sqrt{N}}, (k_1, k_0)_{\sqrt{N}}] = 0 \tag{18}$$

and if $k_1 \neq j_1$

$$\mathbf{R}_{k_1,j_1}[k_0, j_0] \stackrel{\text{def}}{=} \mathbf{R}[(k_1, k_0)_{\sqrt{N}}, (j_1, j_0)_{\sqrt{N}}] = 0. \tag{19}$$

Pictorially, $\overline{\mathbf{L}}$ and $\mathbf{R}$ look as follows:

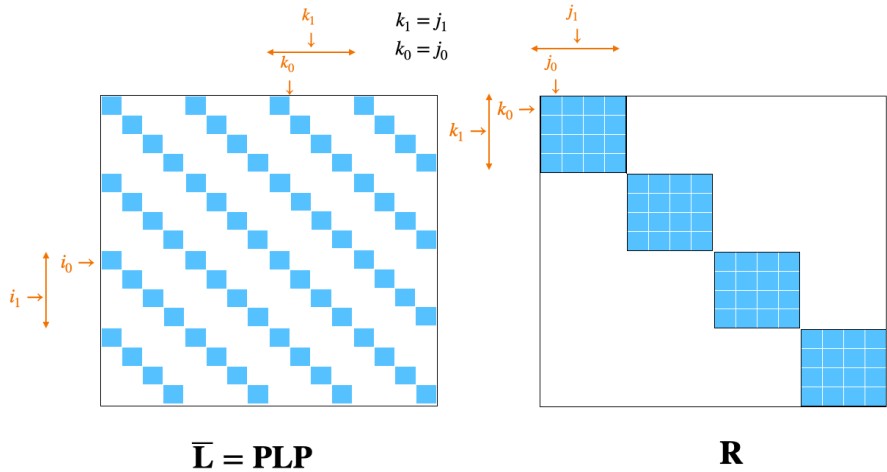

$$\overline{\mathbf{L}} = \mathbf{PLP} \qquad\qquad \mathbf{R}$$

In [14], Monarch matrices with block size $b = \sqrt{N}$, $\mathbf{M}' = \overline{\mathbf{L}} \cdot \mathbf{R}$, and thus for all $0 \leq i_1, i_0, j_1, j_0 < \sqrt{N}$:

$$\mathbf{M}'[(i_1, i_0)_{\sqrt{N}}, (j_1, j_0)_{\sqrt{N}}] = \overline{\mathbf{L}}_{i_1,j_1}[i_0, i_0] \cdot \mathbf{R}_{j_1,j_1}[i_0, j_0]. \tag{20}$$

We note that our definition of Monarch matrix $\mathbf{M}$ in Section 3 is slightly different in that $\mathbf{M} = \mathbf{M}'\mathbf{P}$ with $\mathbf{M}'$ as defined in [14].

## D.2 Monarch Matrices and Bivariate Polynomial Evaluation

Given polynomials $\ell_{j_1}(X, Y)$ for $0 \le j_1 < \sqrt{N}$, polynomials $r_{j_1, j_0}(Y)$ for $0 \le j_1, j_0 < \sqrt{N}$, evaluation points $A = (\alpha_0, ..., \alpha_{\sqrt{N}-1})$ $B = (\beta_0, \cdots, \beta_{\sqrt{N}-1})$ (as in Appendix D.1.2), define the matrices $\overline{\mathbf{L}} \in \mathcal{DB}^{\sqrt{N}, N}$ and $\mathbf{R} \in \mathcal{BD}^{\sqrt{N}, N}$ as:

- For every $0 \le j_1, i_1, i_0 < \sqrt{N}$:

$$\overline{\mathbf{L}}_{i_1, j_1}[i_0, i_0] \leftarrow \ell_{j_1}(\alpha_{i_1}, \beta_{i_0}). \tag{21}$$

- For every $0 \le j_1, j_0, i_0 < \sqrt{N}$:

$$\mathbf{R}_{j_1, j_1}[i_0, j_0] \leftarrow r_{j_1, j_0}(\beta_{i_0}). \tag{22}$$

Note that all entries of $\overline{\mathbf{L}}$ and $\mathbf{R}$ not specified above are set to 0.

Let $f$ be the above function that maps coefficients of $\ell_{j_1}(X, Y)$ (which are coefficients of monomials $X^{i_1} Y^{i_0}$ for all $0 \le i_1, i_0 < \sqrt{N}$ and hence represented by a matrix in $\mathbb{R}^{\sqrt{N} \times \sqrt{N}}$) and coefficients of $r_{j_1, j_0}(Y)$ (which are coefficients of monomials $Y^{i_0}$ for all $0 \le i_0 < \sqrt{N}$ and hence represented by a vector in $\mathbb{R}^{\sqrt{N}}$) for all $0 \le j_1, j_0 < \sqrt{N}$ to pairs of matrices in $\mathcal{DB}^{\sqrt{N}, N} \times \mathcal{BD}^{\sqrt{N}, N}$.

**Theorem 6.** *Let $f$ be as defined above. Then $f$ is a bijection.*

*Proof.* To prove $f$ is bijection we must show $f$ is one-to-one and $f^{-1}$ is one-to-one (and exists).

To show $f$ is one to one means each set of polynomials' coefficients given to $f$, will output a unique set of matrices $(\overline{\mathbf{L}}, \mathbf{R}) \in \mathcal{DB}^{\sqrt{N}, N} \times \mathcal{BD}^{\sqrt{N}, N}$. This follows from (21), (22) and the known fact that polynomial evaluation is a function.

Now, to show $f^{-1}$ exists and is one-to-one, we must show that there is a map from any pair $(\overline{\mathbf{L}}, \mathbf{R}) \in \mathcal{DB}^{\sqrt{N}, N} \times \mathcal{BD}^{\sqrt{N}, N}$ to unique sets of polynomials, $\bar{\ell}, \bar{r}$, with parameters as defined in Appendix D.1.2. Further, we need

$$\overline{\mathbf{L}}_{i_1, j_1}[i_0, i_0] = \bar{\ell}_{j_1}(\alpha_{i_1}, \beta_{i_0}) \tag{23}$$

and

$$\mathbf{R}_{j_1, j_1}[i_0, j_0] = \bar{r}_{j_1, j_0}(\beta_{i_0}). \tag{24}$$

We will use Theorems 5 and 4 to show the existence of $\bar{\ell}_{j_1}$ and $\bar{r}_{j_1, j_0}$ polynomials, giving us the mapping from the matrices to unique polynomials.

We first show the existence of the unique polynomials in (24). Fix $0 \le j_1, j_0 < \sqrt{N}$. Then consider the values $0 \le i_0 < \sqrt{N}$:

$$y_{i_0} \leftarrow \mathbf{R}_{j_1, j_1}[i_0, j_0]. \tag{25}$$

Then by Theorem 4, there exists a unique polynomial of degree $< \sqrt{N}$ (call it $\bar{r}_{j_1, j_0}(Y)$) such that for all $0 \le i_0 < \sqrt{N}$,

$$\bar{r}_{j_1, j_0}(\beta_{i_0}) = y_{i_0},$$

which by (25) shows (24).

Next we show the existence of the unique polynomials in (23). Fix $0 \le j_1 < \sqrt{N}$. Consider the values $0 \le i_1, i_0 < \sqrt{N}$:

$$y_{i_1, i_0} \leftarrow \overline{\mathbf{L}}_{i_1, j_1}[i_0, i_0]. \tag{26}$$

Then by Theorem 5, there exists a unique bi-variate polynomial of $\deg_X < \sqrt{N}$ and $\deg_Y < \sqrt{N}$ (call it $\bar{\ell}_{j_1}(X, Y)$) such that for all $0 \le i_1, i_0 < \sqrt{N}$,

$$\bar{\ell}_{j_1}(\alpha_{i_1}, \beta_{i_0}) = y_{i_1, i_0}, \tag{27}$$

which by (26) shows (23).

Therefore $f$ is a bijection. $\qquad\square$

We can now conclude:

**Corollary 1.** *For every matrix* $\mathbf{M}'$ *as defined in* (20)*, there exists unique polynomials* $\ell_{j_1}(X, Y)$ *and* $r_{j_1, j_0}(Y)$*, such that for all* $0 \leq i_1, i_0, j_1, j_0 < \sqrt{N}$*,*

$$\mathbf{M}'[(i_1, i_0)_{\sqrt{N}}, (j_1, j_0)_{\sqrt{N}}] = \ell_{j_1}(\alpha_{i_1}, \beta_{i_0}) \cdot r_{j_1, j_0}(\beta_{i_0}). \tag{28}$$

*Proof.* Follows from (20), (21), (22) and Theorem 6. $\qquad\square$

### D.2.1    Proof of Theorem 1

We begin with an immediate consequence of Corollary 1:

**Corollary 2.** *Let* $A, B \subset \mathbb{C}$ *such that* $|A| = |B| = \sqrt{N}$*. Then the jth column of* $\mathbf{M}'$ *is the evaluation of the polynomial* $\ell_{j_1}(X, Y) \cdot r_{j_1, j_0}(Y)$ *over* $A \times B$*.*

*Proof.* Observe that for fixed $j_0, j_1$ the right hand side of (28) is $\ell_{j_1}(X, Y) \cdot r_{j_1, j_0}(Y)$ evaluated at all $(\alpha, \beta) \in A \times B$. Thus, $(j_1, j_0)$ column is evaluation of $\ell_{j_1}(X, Y) \cdot r_{j_1, j_0}(Y)$ over points in $A \times B$, as desired.

$\qquad\square$

Next, we state a generalization of Theorem 1 that follows from Corollary 2:

**Corollary 3.** *Let A and B be as in Corollary 2. For any vector* $\boldsymbol{u}$*,* $\mathbf{M}' \cdot \boldsymbol{u}$ *is* $\bar{u}(X, Y)$ *evaluated at* $A \times B$*. Further,*

$$\bar{u}(X, Y) = \sum_{0 \leq j_1, j_0 < \sqrt{N}} u_{j_1, j_0} \cdot \ell_{j_1}(X, Y) \cdot r_{j_1, j_0}(Y), \tag{29}$$

*where* $\ell$ *and* $r$ *are defined by* $\mathbf{M}'$ *as in Corollary 1.*

*Proof.* Follows from Corollary 2 and definition of matrix vector multiplication. $\qquad\square$

In Theorem 1 and the following sections, we consider the polynomial evaluated over the basis polynomials defined by

$$\mathbf{M} = \mathbf{M}'\mathbf{P}$$

.

**Corollary 4.** *Let A and B be as in Corollary 2. For any vector* $\boldsymbol{u}$*, and* $\mathbf{M} \cdot \mathbf{u}$ *is* $u(X, Y)$ *evaluated at* $A \times B$*. Further,*

$$u(X, Y) = \sum_{0 \leq j_1, j_0 < \sqrt{N}} u_{j_0, j_1} \cdot \ell_{j_0}(X, Y) \cdot r_{j_0, j_1}(Y), \tag{30}$$

*where* $\ell$ *and* $r$ *are defined by* $\mathbf{M}'$ *as in Corollary 1.*

*Proof.* Follows from Corollary 3 and definition of $\mathbf{M}$. $\qquad\square$

Specifically, Theorem 1 is a special case of Corollary 4 where $\ell_{j_0}(X, Y) = \ell_{m(j)}(X, Y)$ and $r_j(Y) = r_{j_0, j_1}(Y)$.

### D.2.2    Proof of Theorem 2

By Corollary 4, for all $0 \leq j_1, i_1 < \sqrt{N}$,

$$\ell_{j_0}(X, Y), \ r_{j_0, j_1}(Y), \bar{\ell}_{j_0}(X, Y), \ \bar{r}_{j_0, j_1}(Y)$$

are the basis polynomials corresponding to $\mathbf{M}_1$ and $\mathbf{M}_2$. For the coefficient vector $\mathbf{k} = (k_{j_1,j_0})_{0 \leq j_1,j_0 < \sqrt{N}}$ and similarly for $\mathbf{u} = (u_{j_1,j_0})_{0 \leq j_1,j_0 < \sqrt{N}}$, we can construct two polynomials

$$k(X,Y) = \sum_{0 \leq j_1,j_0 < \sqrt{N}} k_{j_1,j_0} \cdot \ell_{j_0}(X,Y) \cdot r_{j_0,j_1}(Y)$$

$$u(X,Y) = \sum_{0 \leq j_1,j_0 < \sqrt{N}} u_{j_1,j_0} \cdot \bar{\ell}_{j_0}(X,Y) \cdot \bar{r}_{j_0,j_1}(Y)$$

whose evaluation over $(\alpha_{i_1}, \beta_{i_0}) = (\omega^{i_1}, \omega^{i_0})$ where recall as in Appendix D.1.1 $\omega = e^{\frac{2\pi\iota}{\sqrt{N}}}$, by Theorem 1 is equivalent to the products $\mathbf{M}_1 \cdot \mathbf{k}$ and $\mathbf{M}_2 \cdot \mathbf{u}$, respectively. Taking the component-wise product, $y = (\mathbf{M}_1 \cdot \mathbf{k}) \odot (\mathbf{M}_2 \cdot \mathbf{u})$, the entry at $i = (i_1, i_0)$ is given by

$$\mathbf{y}[(i_1, i_0)] = k(\omega^{i_1}, \omega^{i_0}) \cdot u(\omega^{i_1}, \omega^{i_0}).$$

Noting that the element of $A$, i.e. the $\sqrt{N}$-th roots of unity, satisfy $Z^{\sqrt{N}} = 1$ means that the above are evaluations of

$$h(X,Y) = k(X,Y) \cdot u(X,Y) \mod (X^{\sqrt{N}} - 1, Y^{\sqrt{N}} - 1)$$

at $A \times A$. Finally, Theorem 1 and the fact that $\mathbf{M}_0^{-1}$ exists implies $\mathbf{M}_0 \cdot \mathbf{y}$ is polynomial interpolation into basis polynomials corresponding to $\mathbf{M}_0$. (Here we use the well known fact polynomial interpolation is the inverse of polynomial evaluation).

## D.3 Proof of Theorem 3

We review some concepts in Appendix D.3.1. In Appendix D.3.2, we discuss square matrices and causality in terms of operations on univariate polynomials. This allows us to define a general class of operators for causal 1D convolution. In Appendix D.3.3, we give a class of matrices suitable for perform causal Monarch convolution. Specifically, we prove Theorem 3.

### D.3.1 Review

Consider the linear operation on an input vector $\mathbf{u}$:

$$\mathbf{y} = \mathbf{A} \cdot \mathbf{u}.$$

We say that the map is causal to mean the entry $\mathbf{y}[i]$ only depends on $\mathbf{u}[0], \mathbf{u}[1], \ldots \mathbf{u}[i]$. This will be the case when $\mathbf{A}$ is a lower triangular matrix (we index the top left entry of $\mathbf{A}$ as $(0,0)$). When $\mathbf{A}$ is a lower triangular Toeplitz matrix with entries corresponding to some coefficient vector $\mathbf{k}$, this operation is exactly the 1D convolution

$$\mathbf{y} = \mathbf{k} * \mathbf{u} = \left( \mathbf{F}_{2n}^{-1} \cdot \left( (\mathbf{F}_{2n} \cdot \mathbf{k}') \circ (\mathbf{F}_{2n} \cdot \mathbf{u}') \right) \right) [0 : n-1],$$

where $\mathbf{k}' = (\mathbf{k}, \mathbf{0}_n)$, $\mathbf{u}' = (\mathbf{u}, \mathbf{0}_n)$, and $\mathbf{F}_n$ is the $n \times n$ DFT matrix.

**Definition 4.** *For a matrix $\overline{\mathbf{M}} \in \mathbb{R}^{n \times n}$, let us define the map*

$$\mathbf{y} = \mathbf{M}^{-1}(\overline{\mathbf{M}} \cdot \mathbf{k} \odot \overline{\mathbf{M}} \cdot \mathbf{u}) \tag{31}$$

*as* matrix convolution. *When $\overline{\mathbf{M}}$ is a Monarch matrix,* (31) *is called* Monarch convolution.

In this section, we are interested in determining large subclasses of matrices $\overline{\mathbf{M}}$ such that for any coefficient vector $\mathbf{k}$, (31) is causal in $\mathbf{u}$. We provide a class of matrices for which Monarch convolution is causal.

We note that for general Monarch matrix $\mathbf{M}$, (31) is not causal in $\mathbf{u}$. By Theorem 2, we have

$$y(X,Y) = k(X,Y) \cdot u(X,Y) \mod (X^{\sqrt{N}-1}, Y^{\sqrt{N}-1}).$$

This is not causal because the $\mod (X^{\sqrt{N}-1}, Y^{\sqrt{N}-1})$ term condenses higher order terms into lower order terms, hence the $\mathbf{y}[i]$ wouldn't just depend on input information up to value $i$.

### D.3.2 Univariate Matrix Convolutions

We start with a couple of notation assumptions.

**Assumption 1.** *$N$ is a perfect square.*

**Assumption 2.** *We will not use pair notation for this subsection since throughout we have $i = i_1\sqrt{N} + i_0$ and $j = j + 1\sqrt{N} + j_0$.*

In order to discuss square matrices in terms of univariate polynomials, we give univariate analogs of Theorem 1 and Theorem 2 for general univariate basis. With an eye toward towards performing causal convolution, we restrict our analysis to certain classes of univariate polynomials.

We first define matrices whose $j^{th}$ columns are the evaluation of a minimum degree $j$ (and maximum degree $N - 1$) polynomial (recall Definition 1). We generalize Theorem 3 to such matrices.

**Lemma 1.** *For sequence of points $A = \{1, \omega_N, \cdots \omega_N^{N-1}\}$ where $\omega_N$ is the $N^{th}$ root of unity, let $\overline{\mathbf{M}}$ be defined as*

$$\overline{\mathbf{M}}[i, j] = \bar{q}_j(\omega_N^i) \tag{32}$$

*where $\bar{q}_j(Z)$ is defined as in Definition 1. Then for any vector $\boldsymbol{v} \in \mathbb{R}^N$, $\overline{\mathbf{M}} \cdot \boldsymbol{v}$ is equivalent to evaluating the polynomial*

$$v(Z) = \sum_{j=0}^{N-1} v_j \cdot \bar{q}_j(Z) \tag{33}$$

*at $\{1, \omega_N, \cdots \omega_N^{N-1}\}$.*

*Proof.* By our definition of $\overline{\mathbf{M}}$, the column $\overline{\mathbf{M}}[:, j]$ is exactly the evaluation of the polynomial $\bar{q}_j(Z)$ at each point in $A$. The claimed result comes from the definition of matrix vector multiplication and (33). $\square$

Note that $\overline{\mathbf{M}}$ or any $\mathbf{M}$'s in this sub-section are not necessarily Monarch matrices.

Next, we state the following intermediate result:

**Proposition 1.** *Let $A$ be the set of the $N$-th roots of unity. Then for $\mathbf{M}_1, \mathbf{M}_2$ defined as in (32)*

$$\mathbf{y} = (\mathbf{M}_1 \cdot \mathbf{k}) \odot (\mathbf{M}_2 \cdot \mathbf{u})$$

*is the same as evaluating the polynomial*

$$p(Z) := k(Z) \cdot u(Z) \mod (Z^N - 1)$$

*over $A$ where $k(Z)$, $u(Z)$ are of the form (33), corresponding to $\mathbf{M}_1$ and $\mathbf{M}_2$, respectively. In other words, for any $0 \le i < N$,*

$$\mathbf{y}[i] = p\left(\omega_N^i\right).$$

*Proof.* This result follows from Lemma 1 and the definition of the Hadamard product. $\square$

Next, we state a re-interpretation of $\mathbf{M}^{-1}\mathbf{y}$:

**Proposition 2.** *Let $\mathbf{M}$ be a full rank matrix whose columns are the evaluations of the basis polynomials $\bar{q}_j(Z)$ from Definition 1 for $0 \le j < N$, and let $\mathbf{y} \in \mathbb{R}^N$ be an arbitrary vector. If $\mathbf{u} = \mathbf{M}^{-1}\mathbf{y}$, then for all $0 \le i < N$*

$$\mathbf{y}[i] = u(\omega^i)$$

*where $u(Z)$ is the same as in Lemma 1 for $\mathbf{M}$. In other words, $\mathbf{M}^{-1}\mathbf{y}$ is the polynomial interpolaton problem for the polynomial basis $\bar{q}_j(Z)$ for $0 \le j < N$.*

*Proof.* This follows from Lemma 1 and the fact that $\mathbf{M}$ is invertible. $\square$

From Propositions 1 and 2, we get the following generalization of Theorem 2:

**Theorem 7.** *For matrices* $\mathbf{M}_0, \mathbf{M}_1, \mathbf{M}_2$ *as defined above, the operation*

$$\mathbf{f} = \mathbf{M}_0^{-1} \cdot ((\mathbf{M}_1 \cdot \mathbf{k}) \circ (\mathbf{M}_2 \cdot \mathbf{u}))$$

*is equivalent to representing the polynomial*

$$f(Z) = k(Z) \cdot u(Z) \mod (Z^N - 1)$$

*in terms of the basis polynomials*

$$\hat{q}_j(Z) \text{ for } j = 0, \ldots, N - 1$$

*where* $k(Z), u(Z)$ *are defined as in Lemma 1 in terms of the basis polynomials corresponding to* $\mathbf{M}_1$ *and* $\mathbf{M}_2$, *respectively, and* $(\hat{q}_j(Z))_{0 \le j < N}$ *corresponds to* $\mathbf{M}_0$.

*Proof.* Follows from Propositions 1 and 2. $\qquad\qquad\qquad\qquad\qquad\qquad\qquad\qquad\qquad\quad$ $\square$

Now we give the class of matrices from which we can build a causal map. Specifically we prove a generalization of Theorem 3:

**Theorem 8.** *Let* $n \ge 1$, *let* $N = \left\lceil \sqrt{2n} \right\rceil^2$. *Then define the basis polynomial* $\bar{q}_j(Z)$ *to have minimum degree* $j$ *and maximum degree* $n - 1$ *for* $0 \le j < n$, *and for* $n \le j < N$, $\bar{q}_j(Z)$ *has minimum degree* $j$ *and maximum degree* $N - 1$.

*For all* $\mathbf{M}_N$ *with basis columns defined by* $(\bar{q}_j(Z))_{0 \le j < N}$ *as above, the operation*

$$\mathbf{u} \mapsto \left( \mathbf{M}_N^{-1}(\mathbf{M}_N \cdot (\mathbf{k}, \mathbf{0}_{N-n}) \circ \mathbf{M}_N \cdot (\mathbf{u}, \mathbf{0}_{N-n})) \right) [0 : n - 1] \qquad (34)$$

*gives a causal map.*

*Proof.* To prove this is causal means each entry, $\mathbf{f}[i]$ is dependent only on $\mathbf{u}[0], \mathbf{u}[1], \ldots \mathbf{u}[i]$, where $\mathbf{f} = \left( \mathbf{M}_N^{-1}(\mathbf{M}_N \cdot (\mathbf{k}, \mathbf{0}_{N-n}) \circ \mathbf{M}_N \cdot (\mathbf{u}, \mathbf{0}_{N-n})) \right)$. By Theorem 7 , we have

$$f(Z) = k(Z) \cdot u(Z) \mod (Z^N - 1),$$

where

$$k(Z) = \sum_{j=0}^{n-1} k_j \bar{q}_j(Z) \quad u(Z) = \sum_{j'=0}^{n-1} u_{j'} \bar{q}_{j'}(Z).$$

Since $\deg(k(Z) \cdot u(Z)) \le 2n - 2 \le N - 2$, this is equivalent to

$$\begin{aligned}
f(Z) &= k(Z) \cdot u(Z) \\
&= \sum_{j,j'=0}^{n-1} k_{j'} \cdot u_j \cdot \bar{q}_{j'}(Z) \cdot \bar{q}_j(Z).
\end{aligned}$$

By our choice of $\bar{q}_j$, we ensure that $\bar{q}_j \cdot \bar{q}_{j'}$ has minimum degree $j + j'$ and $\deg(\bar{q}_j \cdot \bar{q}_{j'}) \le 2n - 2 < N$ for any $0 \le j, j' < n$. Then by Lemma 2 (see below), there exists coefficients $\alpha_{j+j',i'}$ such that,

$$\begin{aligned}
f(Z) &= \sum_{j,j'=0}^{n-1} k_{j'} \cdot u_j \cdot \sum_{i'=j+j'}^{N-1} \alpha_{j+j',i'} \cdot \bar{q}_{i'}(Z) \\
&= \sum_{i=0}^{N-1} \left( \sum_{\substack{j,j'=0 \\ j+j' \le i}}^{n-1} \alpha_{j+j',i} \cdot k_{j'} \cdot u_j \right) \cdot \bar{q}_i(Z).
\end{aligned}$$

If we define

$$f(Z) = \sum_{i=0}^{N-1} f_i \cdot \bar{q}_i(Z),$$

then for $0 \leq i < n$, we get:

$$f_i = \sum_{\substack{j,j'=0 \\ j+j' \leq i}}^{n-1} \alpha_{j+j',i} \cdot k_{j'} \cdot u_j.$$

Note that $f_i$ only depends on $\mathbf{u}[0], \mathbf{u}[1], \ldots \mathbf{u}[i]$, as desired. $\qquad\square$

We used the following lemma in the proof of Theorem 8.

**Lemma 2.** *Let $\bar{q}_j(Z)$ be defined as in Theorem 8. Then for any $0 \leq j, j' < n$,*

$$\bar{q}_j(Z) \cdot \bar{q}_{j'}(Z) = \sum_{i=j+j'}^{N-1} \alpha_{j+j',i} \cdot \bar{q}_i(Z). \tag{35}$$

*for some set of coefficients $\alpha_{j+j',i}$.*

*Proof.* We first note that by our choice of $\bar{q}_j$, the minimum degree of $\bar{q}_j(Z) \cdot \bar{q}_{j'}(Z)$ is $j + j'$, and $\deg(\bar{q}_j(Z) \cdot \bar{q}_{j'}(Z)) \leq 2n - 2 \leq N - 1$. Our claim follows from that fact that any polynomial $p_d(Z)$ of minimum degree $d$ and $\deg(p_d) < N$ can be expressed as a linear combination of $\bar{q}_d(Z), \bar{q}_{d+1}(Z), \ldots, \bar{q}_{N-1}(Z)$. [4] $\qquad\square$

### D.3.3 Causal Monarch Convolutions

In this section we will prove Theorem 3. We will do so by showing that the basis polynomials $q_j(Z)$ as defined in Theorem 3 are a special case of the basis polynomials $\bar{q}_j(Z)$ as defined in Theorem 8.

We start with a couple of notation assumptions.

**Assumption 3.** *In this sub-section we will using block size $b = \sqrt{N}$ therefore, we are dropping the block size from index notation. For example, $(i_1, i_0)_{\sqrt{N}}$ becomes $(i_1, i_0)$ for this section.*

**Assumption 4.** *Permutation matrices in this subsection are all the same $\mathbf{P}_{(\sqrt{N}, N)}$ so we drop the subscript and just use $\mathbf{P}$.*

**Definition 5.** *Define*

$$q'_{(j_1, j_0)}(Z) \stackrel{def}{=} \ell_{j_1}(Z) \cdot r_{j_1, j_0}\left(Z^{\sqrt{N}}\right). \tag{36}$$

$\ell_{j_1}(Z)$ *has minimum degree $j_1$, and $r_{j_1, j_0}(Z)$ has minimum degree $j_0$. All polynomials $q'_{(j_1, j_0)}(Z)$ have maximum degree $\leq N - 1$.*

Next we argue that the above basis polynomials have a specific minimum degree.

**Lemma 3.** *Polynomial $q'_{(j_1, j_0)}(Z)$ as defined in equation (36) has minimum degree $j_0\sqrt{N} + j_1$.*

*Proof.* We need to show that $q'_{(j_1, j_0)}(Z)$ is a minimum degree $j_0\sqrt{N} + j_1$ polynomial. Note that

$$q'_{(j_1, j_0)}(Z) = \ell_{j_1}(Z) \cdot r_{j_1, j_0}\left(Z^{\sqrt{N}}\right)$$

$$= Z^{\sqrt{N}-1} \cdot \tilde{\ell}_{\sqrt{N}-1-j_1}\left(\frac{1}{Z}\right) \cdot Z^{(\sqrt{N}-1)\cdot\sqrt{N}} \cdot \tilde{r}_{\sqrt{N}-1-j_0}\left(\frac{1}{Z^{\sqrt{N}}}\right)$$

---

[4]This claim can be shown using downward induction on $d = N - 1, N - 2, \ldots$.

where $\tilde{\ell}_{\sqrt{N}-1-j_1}(\cdot)$ has degree $\sqrt{N}-1-j_1$ and $\tilde{r}_{\sqrt{N}-1-j_0}(\cdot)$ has degree $\sqrt{N}-1-j_0$. Simplifying we get

$$q'_{(j_1,j_0)}(Z) = Z^{N-1} \cdot \tilde{\ell}_{\sqrt{N}-1-j_1}\left(\frac{1}{Z}\right) \cdot \tilde{r}_{\sqrt{N}-1-j_0}\left(\frac{1}{Z^{\sqrt{N}}}\right).$$

This claim follows since $\tilde{\ell}_{\sqrt{N}-1-j_1}(Y) \cdot \tilde{r}_{j_1,j_0}\left(Y^{\sqrt{N}}\right)$ has degree $= (\sqrt{N}-1-j_1) + (\sqrt{N}-1-j_0) \cdot \sqrt{N} = (N-1) - (j_0\sqrt{N}+j_1)$. $\qquad\square$

Note that the polynomial $q'_{(j_1,j_0)}(Z)$ has minimum degree $j_0\sqrt{N}+j_1$. This is not $j_1\sqrt{N}+j_0$ as in defined in equation (1), we will talk more about this soon.

Next, we observe that the polynomials in (36) define a matrix that satisfies (20).

**Lemma 4.** *Let* $q'_{(j_1,j_0)}(Z)$ *for* $0 \le j_1\sqrt{N}+j_0 < n$ *be as in (36). Define*

$$\mathbf{M}'[(i_1,i_0),(j_1,j_0)] = q'_{(j_1,j_0)}(\omega_N^{i_1\sqrt{N}+i_0}).$$

*Then* $\mathbf{M}'$ *satisfies (20).*

*Proof.* If we evaluate the polynomials $q'_{(j_1,j_0)}(Z)$ at $\omega_N^i$ for $0 \le i < N$, we get

$$q_{(j_1,j_0)}(\omega_N^i) = \ell_{j_1}(\omega_N^i) \cdot r_{j_1,j_0}(\omega_N^{i\sqrt{N}}).$$

Since $\omega_N^{i\sqrt{N}} = \omega_N^{i_1N+i_0\sqrt{N}} = \omega_N^{i_0\sqrt{N}} = \omega_{\sqrt{N}}^{i_0}$, we get

$$\mathbf{M}'[(i_1,i_0),(j_1,j_0)] = \ell_{j_1}(\omega_N^{i_1\sqrt{N}+i_0}) \cdot r_{j_1,j_0}(\omega_{\sqrt{N}}^{i_0}).$$

The above corresponds to how we define (20) since we have

$$\mathbf{M}'[(i_1,i_0),(j_1,j_0)] = \overline{\mathbf{L}}_{i_1,j_1}[i_0,i_0] \cdot \mathbf{R}_{j_1,j_1}[i_0,j_0]$$

with

$$\overline{\mathbf{L}}_{i_1,j_1}[i_0,i_0] = \ell_{j_1}\left(\omega_N^{i_1\sqrt{N}+i_0}\right)$$

and

$$\mathbf{R}_{j_1,j_1}[i_0,j_0] = r_{j_1,j_0}\left(\omega_{\sqrt{N}}^{i_0}\right).$$

$\qquad\square$

Recall from Lemma 3 that $q'_{(j_1,j_0)}$ has minimum degree $j_0\sqrt{N}+j_1$. For causality we need $q_{(j_1,j_0)}(Z)$ to have degree minimum $j_1\sqrt{N}+j_0$ (then the polynomials will satisfy the minimum degree requirements from (1)). Therefore, we permute the columns of $\mathbf{M}'$,

$$\mathbf{M} = \mathbf{M}' \cdot \mathbf{P} \tag{37}$$

Note that the above $\mathbf{M} = \mathbf{PLPRP}$ and is indeed a Monarch matrix as defined in Section 3.

Note that the basis polynomials of $\mathbf{M}$ are defined as,

$$q_{(j_1,j_0)}(Z) = \ell_{j_0}(Z) \cdot \tilde{r}_{j_0,j_1}\left(Z^{\sqrt{N}}\right). \tag{38}$$

We note that the above is same as $q_j(Z)$ defined in (5) where $j = j_1\sqrt{N}+j_0$ with the correction in (5) that $q_j(Z) = \ell_{m(j)} \cdot r_j(Z^{\sqrt{N}})$ where $r_j(Y) = \tilde{r}_{j_0,j_1}(Y)$.

We are finally ready to prove Theorem 3.

**Corollary 5** (Theorem 3 restated). *Let* $N = \left\lceil\sqrt{2n}\right\rceil^2$. *Define* $\mathbf{M}_N$ *by* $q_{(j_1,j_0)}$ *as in (38). Then,*

$$\mathbf{u} \mapsto \left(\mathbf{M}_N^{-1}(\mathbf{M}_N \cdot (\mathbf{k},\mathbf{0}_{N-n}) \circ \mathbf{M}_N \cdot (\mathbf{u},\mathbf{0}_{N-n}))\right)[0:n-1]$$

*gives a causal map.*

*Proof.* Due to using $q$ polynomials as in (38), by Lemma 3 the degree of the $(j_1,j_0)^{\text{th}}$ column is a polynomial with minimum degree $j_1\sqrt{N}+j_0$.[5] This implies that these basis polynomials are a subset of the more general causal maps in Theorem 8, which proves the claim. $\qquad\square$

---

[5]It can also be verified that $q_{(j_1,j_0)}$ has maximum degree $\le N-1$.

## D.4 Block Algorithms for Complex Numbers and Block Size $\sqrt{N}$

In the following subsections we restate the results in Section D.3.3 in terms of block operations. As mentioned earlier, this is so that we can do computations on Monarch matrices using only GEMM operations (and simple data movement operations like permutations).

In Appendix D.4.1 we consider arbitrary Monarch matrices and in Appendix D.4.2 we consider the sub-class of Monarch matrices corresponding to Theorem 3.

### D.4.1 General Block Monarch Convolution

In this subsection we re-state general Monarch convolutions in terms of block operations.

Recall from equation (22) we defined the block diagonal matrix $\mathbf{R}$ as follows ($0 \leq i_0, j_1, j_0 < \sqrt{N}$):

$$\mathbf{R}_{j_1, j_1}[i_0, j_0] \leftarrow \tilde{r}_{j_1, j_0}\left(\omega_{\sqrt{N}}^{i_0}\right), \tag{39}$$

where $\deg(\tilde{r}_{j_1, j_0}) < \sqrt{N}$.

To do so, we will first work with $\mathbf{M}'_N$ such that $\mathbf{M}_N = \mathbf{M}'_N \cdot \mathbf{P}$. We want to express Monarch matrices, $\mathbf{M}_N$, as univariate polynomial evaluation over $\{1, \omega_N, \ldots, \omega_N^{N-1}\}$. Towards that end, define

$$r_{j_1, j_0}(Z) = \tilde{r}_{j_1, j_0}\left(Z^{\sqrt{N}}\right).$$

By simple observation that $\omega_N^{(i_1\sqrt{N}+i_0)\sqrt{N}} = \omega_N^{i_0\sqrt{N}} = \omega_{\sqrt{N}}^{i_0}$, we have

$$r_{j_1, j_0}(\omega_N^{i_1\sqrt{N}+i_0}) = \tilde{r}_{j_1, j_0}\left(\omega_{\sqrt{N}}^{i_0}\right).$$

In other words we have,

$$\mathbf{R}_{j_1, j_1}[i_0, j_0] = r_{j_1, j_0}\left(\omega_N^i\right).$$

Fix $0 \leq j_1, j_0 < \sqrt{N}$ so we're looking at the $j_0^{th}$ column of block $\mathbf{R}_{j_1, j_1}$, going down the column we evaluate the polynomial $\tilde{r}_{j_1, j_0}$ at points $\left(1, \omega_{\sqrt{N}}, \ldots, \omega_{\sqrt{N}}^{\sqrt{N}-1}\right)$. Which is equivalent to a matrix multiplication of Fourier matrix of size $\sqrt{N}$ and a matrix of the coefficients of the $\tilde{r}$ polynomials.

So we can think of the blocks of $\mathbf{R}$ as a Fourier matrix times a coefficient matrix. In other words, let us define a matrix $\widetilde{\mathbf{R}} \in \mathbb{R}^{N \times \sqrt{N}}$ which will hold $\sqrt{N}$ coefficient blocks $\widehat{\mathbf{R}}_0, \widehat{\mathbf{R}}_1, \ldots \widehat{\mathbf{R}}_{\sqrt{N}-1} \in \mathbb{R}^{\sqrt{N} \times \sqrt{N}}$ such that,

$$\widetilde{\mathbf{R}}_{j_1}[a, j_0] = \tilde{r}_{j_1, j_0}[a],$$

where

$$\tilde{r}_{j_1, j_0}(Y) = \sum_{a=0}^{\sqrt{N}-1} \tilde{r}_{j_1, j_0}[a] \cdot Y^a.$$

Then we define

$$\mathbf{R}_{j_1, j_1} = \mathbf{F}_{\sqrt{N}} \cdot \widetilde{\mathbf{R}}_{j_1}.$$

Next, we restate the above as product of two block diagonal matrices:

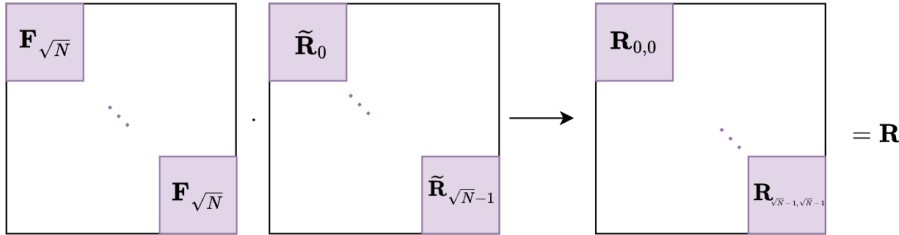

In other words, we have

$$\mathbf{R} = \text{diag}\underbrace{\left(\mathbf{F}_{\sqrt{N}}, \ldots, \mathbf{F}_{\sqrt{N}}\right)}_{\sqrt{N}\text{ times}} \cdot \text{diag}(\widetilde{\mathbf{R}}_0, \ldots, \widetilde{\mathbf{R}}_{\sqrt{N}-1}). \tag{40}$$

Equation (21) defined $\mathbf{L}$ as evaluation of bivariate polynomials. We now wish to do the same with univariate polynomials.

Recall that $\overline{\mathbf{L}} = \mathbf{PLP}$. We define the $i_0^{th}$ diagonal block ($0 \le i_1, i_0, j_1 < \sqrt{N}$) for $\mathbf{L}$ as:

$$\mathbf{L}_{i_0,i_0}[i_1, j_1] = \ell_{j_1}\left(\omega_N^{i_0\sqrt{N}+i_1}\right), \tag{41}$$

where $\deg(\ell_{j_1}) < N$.

Let us define a matrix $\widetilde{\mathbf{L}} \in \mathbb{R}^{N \times \sqrt{N}}$ that will hold the coefficients of polynomials $\ell_{j_1}(Z)$ i.e.

$$\widetilde{\mathbf{L}}[a, j_1] = \tilde{\ell}_{j_1}[a]$$

where

$$\ell_{j_1}(Z) = \sum_{a=0}^{N-1} \tilde{\ell}_{j_1}[a] \cdot Z^a.$$

We can multiply this matrix with the Fourier matrix of size $N \times N$ to get the blocks of $\mathbf{L}$ (which we will need to diagonalize). Specifically define

$$\mathbf{L}'' = \mathbf{F}_N \cdot \widetilde{\mathbf{L}}.$$

The rows of $\mathbf{L}''$ and the rows of $\mathbf{M}'$ are both indexed by $i$. Meaning they're ordered in lexicographic ordering $(i_1, i_0)$, which is a problem for the following reason. The block diagonal matrix $\overline{\mathbf{L}}$ made from $\mathbf{L}''$ needs to be in lexicographic ordering $(i_0, i_1)$ (see (41)) since it gets permuted on the left $\mathbf{M} = \mathbf{PLP}$ and right allowing $\mathbf{M}$ to be ordered by $(i_1, i_0)$. Therefore, when composing $\mathbf{L}$ from $\mathbf{L}''$ we must permute the rows by $\mathbf{P}$. So we get,

$$\mathbf{L}' = \mathbf{P} \cdot \mathbf{L}''.$$

Let

$$\mathbf{L}' = \begin{bmatrix} \mathbf{L}'_0 \\ \vdots \\ \mathbf{L}'_{\sqrt{N}-1} \end{bmatrix}.$$

Then

$$\mathbf{L} = \text{diag}(\mathbf{L}'_0, \ldots \mathbf{L}'_{\sqrt{N}-1}). \tag{42}$$

Pictorially:

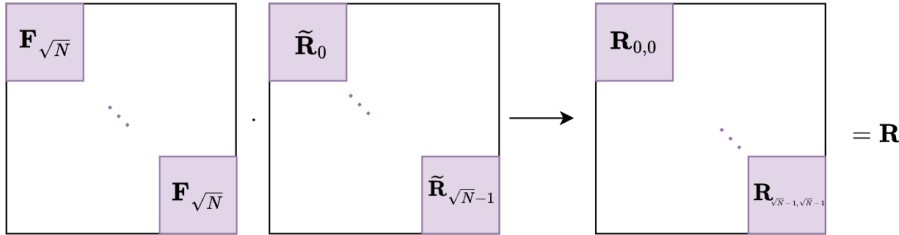

Let $f$ be the function that maps coefficient matrices $\widetilde{\mathbf{L}}, \widetilde{\mathbf{R}} \in \mathbb{R}^{N \times \sqrt{N}}$ to $(\mathbf{L}, \mathbf{R})$ where $\mathbf{L}$ and $\mathbf{R}$ are defined by (42) and (40).

**Theorem 9.** *Let $f$ be as defined above. Then $f$ is a bijection.*

*Proof.* To prove $f$ is a bijection we must show $f$ is one-to-one and $f^{-1}$ is one-to-one (and exists).

To show $f$ is one-to-one means both $\widetilde{\mathbf{L}}$ and $\widetilde{\mathbf{R}}$ given to $f$ will output a unique pair $(\mathbf{L}, \mathbf{R})$. This follows from (42) and (40), and the fact that polynomial evaluation is a function.

Now to show $f^{-1}$ exists and is one-to-one, we must show that there's a map for any $(\mathbf{L}, \mathbf{R})$ to unique $\widetilde{\mathbf{L}}, \widetilde{\mathbf{R}} \in \mathbb{R}^{N \times \sqrt{N}}$. Then for any pair $(\mathbf{L}, \mathbf{R})$ where both $\mathbf{L}$ and $\mathbf{R}$ are block diagonal matrices, there's a map to unique sets of polynomials, $\ell, \tilde{r}$ where each coefficient is an entry of $\widetilde{\mathbf{L}}, \widetilde{\mathbf{R}}$ (thus giving a unique mapping from $(\mathbf{L}, \mathbf{R})$'s to $(\widetilde{\mathbf{L}}, \widetilde{\mathbf{R}})$'s).

We need to show the existence of $\ell_{j_1}(Z)$ and $\tilde{r}_{j_1, j_0}(Y)$ such that:

$$\mathbf{L}_{i_0, i_0}[i_1, j_1] = \ell_{j_1}(\omega_N^{i_0 \sqrt{N} + i_1})$$

and

$$\mathbf{R}_{j_1, j_1}[i_0, j_0] = \tilde{r}_{j_1, j_0}(\omega_{\sqrt{N}}^{i_0}).$$

Fix $0 \leq j_1, j_0 < \sqrt{N}$. Then consider the values $0 \leq i_0 < \sqrt{N}$:

$$y_{i_0} \leftarrow \mathbf{R}_{j_1, j_1}[i_0, j_0].$$

Then by Theorem 4 there exists a unique polynomial of degree $< \sqrt{N}$ (call it $\tilde{r}_{j_1, j_0}$) such that for all $0 \leq i_0 < \sqrt{N}$:

$$\tilde{r}_{j_1, j_0}(\omega_{\sqrt{N}}^{i_0}) = y_{i_0}.$$

There will be $N$ polynomials with $\sqrt{N}$ coefficients each, meaning there's unique sets of coefficients to make up the indices of $\widetilde{\mathbf{R}}$.

Now to get the entries of $\widetilde{\mathbf{L}}$ from $\mathbf{L}$, fix $0 \leq j_1 < \sqrt{N}$. Then consider the values $0 \leq i_1, i_0 < \sqrt{N}$:

$$y_{i_1, i_0} \leftarrow \mathbf{L}_{i_0, i_0}[i_1, j_1].$$

Then by Theorem 4 there exists a unique polynomial of degree $< N$ (call it $\ell_{j_1}$) such that for all $0 \leq i_1, i_0 < \sqrt{N}$:

$$\ell_{j_1}(\omega_N^{i_0 \sqrt{N} + i_1}) = y_{i_1, i_0}.$$

There will be $\sqrt{N}$ polynomials with $N$ coefficients each, meaning there's unique sets of coefficients to make up the indices of $\widetilde{\mathbf{L}}$. $\square$

Algorithm 2 is pseudo code for the map from $\widetilde{\mathbf{L}}, \widetilde{\mathbf{R}} \in \mathbb{R}^{N \times \sqrt{N}}$ to block diagonal matrices $\mathbf{L}, \mathbf{R}$.

---

**Algorithm 2** BLOCKY MONARCH$(\widetilde{\mathbf{L}}, \widetilde{\mathbf{R}})$

---

**Input:** $\widetilde{\mathbf{L}}, \widetilde{\mathbf{R}} \in \mathbb{R}^{N \times \sqrt{N}}$
**Output:** Block diagonal matrices $\mathbf{L}, \mathbf{R} \in \mathbb{C}^{N \times N}$

                                                   $\triangleright$ First, compute $\mathbf{L}$ from $\widetilde{\mathbf{L}}$
1: Let $\mathbf{F}_N$ be the Fourier transform Monarch matrix $\mathbf{P}\mathbf{L}_F\mathbf{P}\mathbf{R}_F\mathbf{P}$            $\triangleright$ See Corollary 6
2: $\mathbf{L}' \leftarrow \mathbf{P} \cdot \mathbf{F}_N \cdot \widetilde{\mathbf{L}}$
3: **for** $a \leftarrow 0$ to $\sqrt{N} - 1$ **do**
4:      $\mathbf{L}'_a \leftarrow \mathbf{L}'[a\sqrt{N} : a\sqrt{N} + \sqrt{N} - 1, :]$
5: $\mathbf{L} \leftarrow \text{diag}(\mathbf{L}'_0, \dots \mathbf{L}'_{\sqrt{N}-1})$

                                                   $\triangleright$ Now compute $\mathbf{R}$ from $\widetilde{\mathbf{R}}$
6: **for** $a \leftarrow 0$ to $\sqrt{N} - 1$ **do**
7:      $\mathbf{R}_a \leftarrow \mathbf{F}_{\sqrt{N}} \cdot \widetilde{\mathbf{R}}[a\sqrt{N} : a\sqrt{N} + \sqrt{N} - 1, :]$
8: $\mathbf{R} \leftarrow \text{diag}(\mathbf{R}_0, \dots, \mathbf{R}_{\sqrt{N}-1})$
9: **return** $\overline{\mathbf{L}}, \mathbf{R}$

---

**Lemma 5.** *Algorithm 2 uses $O(N^{3/2})$ FLOPs and $3\sqrt{N}$ GEMMs of two $\sqrt{N} \times \sqrt{N}$ matrices.*

*Proof.* Lines 1 and 2 are multiplying a monarch matrix representing the Fourier transform times a $N \times \sqrt{N}$ matrix in block fashion again, giving $O(N^{3/2})$ FLOPs.

Similarly, lines 6 and 7 have two $\sqrt{N}$ matrices multiplied $\sqrt{N}$ times. Giving $O(N^{3/2})$ FLOPs.

Lines 3, 4, 5, 8, and 9 don't count towards FLOPs

Therefore we have $O(N^{3/2})$ FLOPS and $3\sqrt{N}$ GEMMs of two $\sqrt{N} \times \sqrt{N}$ matrices. $\qquad\square$

Now that we have the operations in terms of blocks, we will make them causal in the following sub-section.

### D.4.2 Causal Block Monarch Convolution

Recall that a Monarch matrix is defined as

$$\mathbf{M} = \mathbf{PLPRP},$$

then per equation (37) we have

$$\mathbf{M}' = \mathbf{PLPR}. \tag{43}$$

Then if $q_{(j_1,j_0)}(Z)$ is the basis polynomial corresponding to the $(j_1,j_0)^{th}$ column of $\mathbf{M}$, by (38) the basis polynomial corresponding to $\mathbf{M}$ is

$$q_{(j_1,j_0)}(Z) = \ell_{j_0}(Z) \cdot \tilde{r}_{j_0,j_1}(Z^{\sqrt{N}}) \tag{44}$$

with the minimum degree of $j_1\sqrt{N} + j_0$ (recall that we pick $\ell_{j_0}$ and $\tilde{r}_{j_0,j_1}$ to have minimum degree $j_0$ and $j_1$ respectively) as desired.

**Theorem 10.** *Let $\mathbf{L}, \mathbf{R} \leftarrow$ BLOCKY MONARCH($\widetilde{\mathbf{L}}, \widetilde{\mathbf{R}}$). Let $\mathbf{M}$ be as in (37). Then for every $0 \leq i$, $j < N$, we have:*

$$\mathbf{M}[(i_1,i_0),(j_1,j_0)] = q_{(j_1,j_0)}\left(\omega_N^{i_1\sqrt{N}+i_0}\right),$$

*where $q_{(j_1,j_0)}$ is as in (44).*

*Proof.* To prove this we need to show the $(j_1,j_0)^{th}$ column of $\mathbf{M}$ is the basis polynomial $q_{(j_1,j_0)}$ evaluated at the $N^{th}$ roots of unity, where $q_{(j_1,j_0)}$ is as defined in (44). Note we have

$$q_{(j_1,j_0)}(\omega_N^{i_1\sqrt{N}+i_0}) = \ell_{j_0}\left(\omega_N^{i_1\sqrt{N}+i_0}\right) \cdot \tilde{r}_{j_0,j_1}\left(\omega_N^{(i_1\sqrt{N}+i_0)\sqrt{N}}\right)$$

$$= \ell_{j_0}(\omega_N^{i_1\sqrt{N}+i_0}) \cdot \tilde{r}_{j_0,j_1}\left(\omega_{\sqrt{N}}^{i_0}\right). \tag{45}$$

By definition we have,

$$\mathbf{M}'[(i_1,i_0),(j_1,j_0)] = \overline{\mathbf{L}}_{i_1,j_1}[i_0,i_0] \cdot \mathbf{R}_{j_1,j_1}[i_0,j_0].$$

By (39) we have

$$\mathbf{R}_{j_1,j_1}[i_0,j_0] = \tilde{r}_{j_1,j_0}\left(\omega_{\sqrt{N}}^{i_0}\right)$$

and by (41) and the fact that

$$\overline{\mathbf{L}} = \mathbf{PLP},$$

we have

$$\overline{\mathbf{L}}_{i_1,j_1}[i_0,i_0] = \ell_{j_1}\left(\omega_N^{i_1\sqrt{N}+i_0}\right).$$

Thus, we have

$$\mathbf{M}'[(i_1,i_0),(j_1,j_0)] = \ell_{j_1}\left(\omega_N^{i_1\sqrt{N}+i_0}\right) \cdot \tilde{r}_{j_1,j_0}\left(\omega_{\sqrt{N}}^{i_0}\right).$$

Since

$$\mathbf{M} \cdot \mathbf{P} = \mathbf{M}',$$

we have

$$\mathbf{M}[(i_1, i_0), (j_1, j_0)] = \mathbf{M}'[(i_1, i_0), (j_0, j_1)]$$
$$= \ell_{j_0}\left(\omega_N^{i_1\sqrt{N}+i_0}\right) \cdot r_{j_0, j_1}\left(\omega_{\sqrt{N}}^{i_0}\right) = q_{(j_1, j_0)}(\omega_N^{i_1\sqrt{N}+i_0}),$$

where the last equality follows from (45).

$\square$

**Corollary 6.** *The DFT is $\mathbf{M}$ as in Theorem 10 when all blocks of $\widetilde{\mathbf{R}}$ and the top block of $\widetilde{\mathbf{L}}$ are the identity matrix of size $\sqrt{N} \times \sqrt{N}$ (the rest of $\widetilde{\mathbf{L}}$ is all 0's).*

*Proof.* Since only the diagonal of the top block of $\widetilde{\mathbf{L}}$ will contain any non-zero values we only index the first $0, \ldots, \sqrt{N} - 1$ rows. And since all blocks of $\widetilde{\mathbf{R}}$ are the identity matrix we get

$$\widetilde{\mathbf{L}}[j_1, j_1] = \tilde{\ell}_{j_1}[j_1] = 1$$

and

$$\widetilde{\mathbf{R}}_{j_1}[j_0, j_0] = \tilde{r}_{j_1, j_0}[j_0] = 1.$$

All other entries of $\widetilde{\mathbf{L}}$ and $\widetilde{\mathbf{R}}$ are 0. Thus, we have

$$\ell_{j_1}(Z) = Z^{j_1}$$

and

$$\tilde{r}_{j_1, j_0}(Z) = Z^{j_0}.$$

As per Theorem 10,

$$q_{(j_1, j_0)}(Z) = \ell_{j_0}(Z) \cdot \tilde{r}_{j_0, j_1}\left(Z^{\sqrt{N}}\right)$$
$$= Z^{j_0} \cdot Z^{j_1\sqrt{N}} = Z^{j_0 + j_1\sqrt{N}} = Z^j.$$

Then by Theorem 10 note

$$q_{(j_1, j_0)}\left(\omega_N^i\right) = \omega_N^{ij} = \mathbf{M}[(i_1, i_0), (j_1, j_0)],$$

which implies $\mathbf{M}$ is $\mathbf{F}_N$ as desired.

$\square$

Algorithm 3 is pseudo code for the Monarch convolution algorithm. It maps an input space of $\widetilde{\mathbf{L}}, \widetilde{\mathbf{R}}$ matrices, a kernel vector $\mathbf{k}$, and input vector $\mathbf{u}$ to a vector $\mathbf{f}$.

---

**Algorithm 3** BLOCKMONARCHCONV($\widetilde{\mathbf{L}}, \widetilde{\mathbf{R}}, \mathbf{k}, \mathbf{u}$)

---

**Input:** $\widetilde{\mathbf{L}}, \widetilde{\mathbf{R}} \in \mathbb{R}^{N \times \sqrt{N}}$, $\mathbf{k}, \mathbf{u} \in \mathbb{R}^N$
**Output:** $\mathbf{f} \in \mathbb{R}^N$
  1: $\mathbf{L}, \mathbf{R} \leftarrow$ BLOCKY MONARCH($\widetilde{\mathbf{L}}, \widetilde{\mathbf{R}}$)
  2: $\mathbf{M} \leftarrow \mathbf{PLPRP}$                                      ▷ Get $\mathbf{M}'$ from BLOCKY MONARCH
                                               ▷ Compute $\mathbf{k}_f, \mathbf{u}_f$ from $\mathbf{M}$
  3: $\mathbf{k}_f \leftarrow \mathbf{M} \cdot \mathbf{k}$
  4: $\mathbf{u}_f \leftarrow \mathbf{M} \cdot \mathbf{u}$

  5: $\mathbf{f} \leftarrow \mathbf{M}^{-1} \cdot (\mathbf{k}_f \odot \mathbf{u}_f)$                             ▷ Compute f
  6: **return f**

---

We next outline how to make the map in Algorithm 3 causal. Towards that end, we observe:

**Lemma 6.** *For any fixed $n \geq 1$, let $N = \left\lceil \sqrt{2n} \right\rceil^2$. Define $\widetilde{\mathbf{L}}, \widetilde{\mathbf{R}} \in \mathbb{R}^{N \times \sqrt{N}}$ such that for every $0 \leq j_1 < \sqrt{N}$,*

$$\widetilde{\mathbf{L}} = \begin{bmatrix} \widetilde{\mathbf{L}}_0 \\ \vdots \\ \widetilde{\mathbf{L}}_{\sqrt{N}-1} \end{bmatrix} \text{ and } \widetilde{\mathbf{R}} = \begin{bmatrix} \widetilde{\mathbf{R}}_0 \\ \vdots \\ \widetilde{\mathbf{R}}_{\sqrt{N}-1} \end{bmatrix}.$$

*Define $\widetilde{\mathbf{L}}', \widetilde{\mathbf{R}}' \in \mathbb{R}^{N \times \sqrt{N}}$ with corresponding coefficient blocks $\left( \widetilde{\mathbf{L}}'_k \right)_{0 \leq k < \sqrt{N}}$, $\left( \widetilde{\mathbf{R}}'_k \right)_{0 \leq k < \sqrt{N}}$ as*

$$\widetilde{\mathbf{L}}' = \begin{bmatrix} \widetilde{\mathbf{L}}'_0 \\ \mathbf{0}_{\sqrt{N} \times \sqrt{N}} \\ \vdots \\ \mathbf{0}_{\sqrt{N} \times \sqrt{N}} \end{bmatrix} \text{ and } \widetilde{\mathbf{R}}' = \begin{bmatrix} \widetilde{\mathbf{R}}'_0 \\ \widetilde{\mathbf{R}}'_1 \\ \vdots \\ \widetilde{\mathbf{R}}'_{\sqrt{N}-1} \end{bmatrix}$$

*where*

$$\widetilde{\mathbf{L}}'_0[(i_1, i_0), j_1] = \begin{cases} \widetilde{\mathbf{L}}_0[i_0, j_1] & \text{if } i_1 = 0 \text{ and } i_0 \geq j_1 \\ 0 & \text{otherwise} \end{cases} \quad , \quad \widetilde{\mathbf{L}}'_k = \mathbf{0}_{\sqrt{N} \times \sqrt{N}} \text{ for } k = 1, \cdots, \sqrt{N}-1$$

$$\widetilde{\mathbf{R}}'_{j_1}[i_0, j_0] = \begin{cases} 0 & \text{if } i_0 < j_0 \text{ or } ((i_0 \geq \left\lfloor \frac{\sqrt{N}}{2} \right\rfloor) \text{ and } (j_0 < \left\lfloor \frac{\sqrt{N}}{2} \right\rfloor)) \\ \widetilde{\mathbf{R}}_{j_1}[i_0, j_0] & \text{otherwise} \end{cases} \quad \text{for } j_1 = 0, \ldots, \sqrt{N} - 1.$$

*Further, we require that $\widetilde{\mathbf{R}}'_{j_1}[j_0, j_0]$, $\widetilde{\mathbf{L}}'_0[j_1, j_1]$ are all non-zero entries for all $0 \leq j_0, j_1 < \sqrt{N}$. Then for $j_0\sqrt{N} + j_1 < \sqrt{N} \left\lfloor \frac{\sqrt{N}}{2} \right\rfloor$, the basis polynomial $q'_{(j_1, j_0)}(Z) = \ell'_{j_1}(Z) r'_{j_1, j_0}\left( Z^{\sqrt{N}} \right)$ of $\mathbf{M}' = \mathbf{PLPR}$ has minimum degree $j_0\sqrt{N} + j_1$ and maximum degree $\leq \frac{N}{2} - 1$, and for $\sqrt{N} \left\lfloor \frac{\sqrt{N}}{2} \right\rfloor \leq j_0\sqrt{N} + j_1 < N$, the basis polynomial $q'_{(j_1, j_0)}(Z)$ has minimum degree $j_0\sqrt{N} + j_1$ and maximum degree $\leq N - 1$.*

Note that in $\mathbf{M} = \mathbf{M}'\mathbf{P}$ the $(j_1, j_0)$ basis polynomial $q_{j_1, j_0}(Z) = q'_{j_0, j_1}(Z)$ has the required degree bound.

*Proof.* Let $\mathbf{L}', \mathbf{R}' \leftarrow \text{BLOCKY MONARCH}(\widetilde{\mathbf{L}}', \widetilde{\mathbf{R}}')$, and denote their corresponding polynomials as $\ell'_{j_1}(Z), r'_{j_1, j_0}\left( Z^{\sqrt{N}} \right)$, respectively for every $0 \leq j_1, j_0 < \sqrt{N}$. By our definition, for all $0 \leq j_0, j_1 < \sqrt{N}$, $r'_{j_1, j_0}\left( Z^{\sqrt{N}} \right)$ has minimum degree $j_0\sqrt{N}$ and $\ell'_{j_1}(Z)$ has minimum degree $j_1$. Then it follows that the basis polynomial $q'_{(j_1, j_0)}(Z) = \ell'_{j_1}(Z) r'_{j_1, j_0}\left( Z^{\sqrt{N}} \right)$ has minimum degree $j_1 + j_0\sqrt{N}$.

We now look at the degrees of each basis polynomial. From our definition of $\mathbf{R}'_{j_1}$, all entries $\mathbf{R}'_{j_1}[i_0, j_0] = 0$ for $j_0 < \left\lfloor \frac{\sqrt{N}}{2} \right\rfloor$ and $i_0 \geq \left\lfloor \frac{\sqrt{N}}{2} \right\rfloor$. This implies that for $0 \leq j_0 < \left\lfloor \frac{\sqrt{N}}{2} \right\rfloor$, and $0 \leq j_1 < \sqrt{N}$, we have $\deg(r'_{j_1, j_0}) < \left\lfloor \frac{\sqrt{N}}{2} \right\rfloor$. Since we are only looking at $\mathbf{L}'_0(Z)$, note that degree

$\deg(\ell'_{j_1}) \le \sqrt{N} - 1$. Then it follows that

$$\deg\left(q'_{(j_1, j_0)}\right) \le \sqrt{N} - 1 + \left(\left\lfloor \frac{\sqrt{N}}{2} \right\rfloor - 1\right)\sqrt{N}$$

$$= \sqrt{N} - 1 + \left(\sqrt{N}\left\lfloor \frac{\sqrt{N}}{2} \right\rfloor - \sqrt{N}\right)$$

$$\le \sqrt{N}\frac{\sqrt{N}}{2} - 1$$

$$= \frac{N}{2} - 1.$$

(Note that in the above $j_0\sqrt{N} + j_1 \le \sqrt{N}\left\lfloor \frac{\sqrt{N}}{2} \right\rfloor - 1$ by same calculations above as needed.)

For $\left\lfloor \frac{\sqrt{N}}{2} \right\rfloor \le j_0 < \sqrt{N}$ and $0 \le j_1 < \sqrt{N}$, $\deg(r'_{j_1, j_0}) = \sqrt{N} - 1$, and $\ell'_{j_1}(Z)$ degree $\sqrt{N} - 1$. Then it follows that

$$\deg\left(q'_{(j_1, j_0)}\right) \le \sqrt{N} - 1 + \left(\sqrt{N} - 1\right)\sqrt{N}$$

$$= N - 1,$$

as desired. (Note that $j_0\sqrt{N} + j_1 \ge \left\lfloor \frac{\sqrt{N}}{2} \right\rfloor \sqrt{N}$ as needed.) $\qquad\square$

Finally, we use Lemma 6 and Theorem 8 to conclude the following:

**Theorem 11.** *For any fixed $n \ge 1$, let $N = \left\lceil \sqrt{2n} \right\rceil^2$. Let $\widetilde{\mathbf{L}}', \widetilde{\mathbf{R}}' \in \mathbb{R}^{N \times N}$ with corresponding coefficient blocks $\left(\widetilde{\mathbf{L}}'_k\right)_{0 \le k < \sqrt{N}}, \left(\widetilde{\mathbf{R}}'_k\right)_{0 \le k < \sqrt{N}}$ be defined as in Lemma 6. Then for any $\mathbf{k}, \mathbf{u} \in \mathbb{R}^n$, $\textsc{BlockMonarchConv}\left(\widetilde{\mathbf{L}}', \widetilde{\mathbf{R}}', \mathbf{k}', \mathbf{u}'\right)[0 : n-1]$, where $\mathbf{k}' = (\mathbf{k}, \mathbf{0}_{N-n})$, $\mathbf{u}' = (\mathbf{u}, \mathbf{0}_{N-n})$, is causal in $\mathbf{u}$.*

*Proof.* Note that this is the same setting as Theorem 8. If $N = \left\lceil \sqrt{2n} \right\rceil^2$, then $\left\lfloor \frac{N}{2} \right\rfloor \ge n$. Then by Lemma 6, the basis polynomials $q_{j_1, j_0}(Z)$ of $\mathbf{M} = \mathbf{PLPR}$ have $\deg(q_{(j_1, j_0)}) < \frac{N}{2}$ for $0 \le j_1\sqrt{N} + j_0 < \sqrt{N}\left\lfloor \frac{\sqrt{N}}{2} \right\rfloor \le \left\lfloor \frac{N}{2} \right\rfloor$. [6]

Then (34) computes $\textsc{BlockMonarchConv}\left(\widetilde{\mathbf{L}}', \widetilde{\mathbf{R}}', \mathbf{k}', \mathbf{u}'\right)$. Since Algorithm 3 performs the same operation as (34), the result follows from Lemma 6 and Theorem 8. $\qquad\square$

### D.5 Bivariate Polynomials with Kronecker Substitution Over Reals

Our earlier results pertain to complex evaluation points. In this section we define causal convolution over real evaluation points. To do this, we redefine our basis polynomials in terms of the Chebyshev polynomials of the first kind.

In Appendix D.5.1 we recover Theorem 8 for univariate polynomials defined over real evaluation points. This identifies a class of matrices that we use to define to define a structured subclass in Appendix D.5.2. We show that these matrices can be used to perform (31). Finally, in Appendix D.5.3 we give the analog of Theorem 11 over real evaluation points. However, the resulting matrices are not Monarch matrices but they are close. In Appendix D.5.4 and Appendix D.5.5 we discuss how we can exploit this closeness, and compute (31) more efficiently.

---

[6]Recall that Lemma 6 is stated for basis polynomials $q'_{j_1, j_0}(Z)$ for $\mathbf{M}' = \mathbf{PLPR}$ where $q_{j_1, j_0}(Z) = q'_{j_0, j_1}(Z)$.

### D.5.1 Univariate Evaluation over Real Numbers

For any integer $a \geq 0$, the Chebyshev polynomial of the first kind with degree $a$ is denoted as $T_a(Z)$ and is defined as

$$T_a(\cos \theta) \stackrel{\text{def}}{=} \cos(a\theta), \tag{46}$$

and has the property

$$T_a(-\cos \theta) = (-1)^a \cos(a\theta). \tag{47}$$

To parallel Appendix D.3.2, we consider the class of basis polynomials with the form

$$\bar{q}_j^N(Z) = \sum_{a=0}^{N-j-1} \bar{q}_j[a] \, T_a(Z), \tag{48}$$

evaluated over $(\omega_{N,i})_{0 \leq i < N}$ where

$$\omega_{N,i} \stackrel{\text{def}}{=} \cos \left( \frac{\pi(i + \frac{1}{2})}{N} \right). \tag{49}$$

Let us consider the class of matrices defined over (48) and (49).

**Lemma 7.** *For sequence of points $A = \{\omega_{N,0}, \ldots, \omega_{N,N-1}\}$, let $\mathbf{M}$ be defined as*

$$\mathbf{M}[i,j] = \bar{q}_j^N(\omega_{N,i}) \tag{50}$$

*where $\bar{q}_j^N(Z)$ and $\omega_{N,i}$ are defined as in (48) and (49), respectively. Then for any vector $\boldsymbol{u}$, $\mathbf{M} \cdot \boldsymbol{u}$ is equivalent to evaluating the polynomial*

$$u(Z) = \sum_{j=0}^{N-1} u_j \cdot \bar{q}_j^N(Z) \tag{51}$$

*at each point in $A$.*

*Proof.* By our definition of $\mathbf{M}$, the column $\mathbf{M}[:,j]$ is exactly the evaluation of the polynomial $\bar{q}_j^N(Z)$ at each point in $A$. The claimed result comes from the definition of matrix vector multiplication and (51). □

This leads to the following analog of Theorem 7.

**Theorem 12.** *For matrices $\mathbf{M}_0, \mathbf{M}_1, \mathbf{M}_2$, each of form as in Lemma 7, the operation*

$$\mathbf{f} = \mathbf{M}_0^{-1} \cdot ((\mathbf{M}_1 \cdot \mathbf{k}) \odot (\mathbf{M}_2 \cdot \mathbf{u})) . \tag{52}$$

*is equivalent to representing the polynomial*

$$f(Z) = k(Z) \cdot u(Z) \mod T_N(Z)$$

*in terms of the basis polynomials*

$$\hat{q}_j^N(Z) \text{ for } j = 0, \ldots, N-1$$

*where $k(Z), u(Z)$ are defined in terms of the respective basis polynomials corresponding to $\mathbf{M}_1$ and $\mathbf{M}_2$ as in Lemma 7, and $\left( \hat{q}_j^N(Z) \right)_{0 \leq j < N}$ corresponds to $\mathbf{M}_0$.*

*Proof.* For $A = \{\omega_{N,i}\}_{0 \leq i < N}$, define

$$q_A(Z) = \prod_{\alpha \in A} (Z - \alpha). \tag{53}$$

Then (52) follows since for

$$f(Z) = k(Z) \cdot u(Z) \mod q_A(Z),$$

we have the following for any $\alpha \in A$:

$$f(\alpha) = k(\alpha) \cdot u(\alpha).$$

The claim follows from Lemma 7, (52), the invertibility of $\mathbf{M}_0$, and the known fact that $T_N(Z) = q_A(Z)$. □

We also utilize the following result:

**Lemma 8.** *Let* $\bar{q}_j^{\lfloor \frac{N}{2} \rfloor}(Z)$ *be defined as in* (48). *Then for any* $0 \leq j, j' < N$,

$$\bar{q}_j^{\lfloor \frac{N}{2} \rfloor}(Z) \cdot \bar{q}_{j'}^{\lfloor \frac{N}{2} \rfloor}(Z) = \sum_{i=j+j'}^{N-1} \alpha_{j+j',i} \cdot \bar{q}_i^N(Z). \tag{54}$$

*for some set of coefficients* $\alpha_{j+j',i}$.

*Proof.* From (48), we have

$$\bar{q}_j^{\lfloor \frac{N}{2} \rfloor}(Z) \cdot \bar{q}_{j'}^{\lfloor \frac{N}{2} \rfloor}(Z) = \sum_{a=0}^{\lfloor \frac{N}{2} \rfloor - j - 1} q_j[a] T_a(Z) \cdot \sum_{a'=0}^{\lfloor \frac{N}{2} \rfloor - j' - 1} q'_{j'}[a'] T_{a'}(Z).$$

Recall that within a $m$ dimensional vector space of polynomials, we can define a basis by choosing any set of polynomials with degrees $0, \ldots, m-1$. Because $\deg \left( \bar{q}_j^{\lfloor \frac{N}{2} \rfloor} \cdot \bar{q}_{j'}^{\lfloor \frac{N}{2} \rfloor} \right) \leq 2 \lfloor \frac{N}{2} \rfloor - (j+j') - 2$, it can be written as a linear combination of any set of $2 \lfloor \frac{N}{2} \rfloor - (j+j') - 1$ polynomials where for $0 \leq a \leq 2 \lfloor \frac{N}{2} \rfloor - (j+j') - 2$, the $a$-th polynomial has degree $a$. In other words we can choose the $a$-th polynomial as $q_{N-a-1}^N(Z)$. Thus we have

$$\bar{q}_j^{\lfloor \frac{N}{2} \rfloor}(Z) \cdot \bar{q}_{j'}^{\lfloor \frac{N}{2} \rfloor}(Z) = \sum_{a=0}^{2\lfloor \frac{N}{2} \rfloor - (j+j') - 2} \bar{\alpha}_{j+j',a} \cdot \bar{q}_{N-a-1}^N(Z)$$

for some set of coefficients $\alpha_{j+j',a}$. Then after reindexing $i \leftarrow N - a - 1$ we have

$$\bar{q}_j^{\lfloor \frac{N}{2} \rfloor}(Z) \cdot \bar{q}_{j'}^{\lfloor \frac{N}{2} \rfloor}(Z) = \sum_{i=\left(N-2\lfloor \frac{N}{2} \rfloor\right)+j+j'+1}^{N-1} \bar{\alpha}_{j+j',N-i-1} \cdot \bar{q}_i^N(Z).$$

The claim follows by setting $\alpha_{j+j',j+j'+1} = 0$, and if $N$ is odd, $\alpha_{j+j'+1,j+j'+2} = 0$, and $\alpha_{j+j',i} = \bar{\alpha}_{j+j',N-i-1}$ for other $i$. $\qquad\square$

This allows us to prove the following causality result for convolutions over real evaluation points.

**Theorem 13.** *Fix a family of basis polynomials* $\bar{q}_0^N, \bar{q}_1^N \ldots, \bar{q}_{N-1}^N$ *as defined in* (48). *Let* $N \geq 1$ *be a perfect square,* $n \leq \lfloor \frac{N}{2} \rfloor$, $\mathbf{k}, \mathbf{u} \in \mathbb{R}^n$ *and* $\mathbf{M}_N$ *defined by basis* $\left( \bar{q}_j^N(Z) \right)_{0 \leq j < N}$. *Let* $\mathbf{k}'' = \left( \mathbf{k}, \mathbf{0}_{\lfloor \frac{N}{2} \rfloor - n} \right)$ *and* $\mathbf{u}'' = \left( \mathbf{u}, \mathbf{0}_{\lfloor \frac{N}{2} \rfloor - n} \right)$. *Then the operation*

$$\mathbf{u} \mapsto \left( \mathbf{M}_N^{-1} \left( \mathbf{M}_N \cdot \left( \mathbf{0}_{\lceil \frac{N}{2} \rceil}, \mathbf{k}'' \right) \circ \mathbf{M}_N \cdot \left( \mathbf{0}_{\lceil \frac{N}{2} \rceil}, \mathbf{u}'' \right) \right) \right) [0 : n-1] \tag{55}$$

*defines a causal map in* $\mathbf{u}$.

*Proof.* Let

$$\mathbf{f} = \left( \mathbf{M}_N^{-1} \left( \mathbf{M}_N \cdot \left( \mathbf{0}_{\lceil \frac{N}{2} \rceil}, \mathbf{k}'' \right) \circ \mathbf{M}_N \cdot \left( \mathbf{0}_{\lceil \frac{N}{2} \rceil}, \mathbf{u}'' \right) \right) \right) [0 : n-1]. \tag{56}$$

In order to prove that (55) is actually causal in the input $\mathbf{u} \in \mathbb{R}^n$, we must show that for all $0 \leq i < N$, $\mathbf{f}[i]$ is dependent only on $\mathbf{u}[0], \mathbf{u}[1], \ldots \mathbf{u}[i]$. Let $\mathbf{k}' = \left( \mathbf{0}_{\lceil \frac{N}{2} \rceil}, \mathbf{k}'' \right)$ and $\mathbf{u}' = \left( \mathbf{0}_{\lceil \frac{N}{2} \rceil}, \mathbf{u}'' \right)$. By Lemma 7, $\mathbf{M}_N \cdot \mathbf{k}'$ and $\mathbf{M}_N \cdot \mathbf{u}'$ correspond to the evaluations of the polynomials

$$k'(Z) = \sum_{j=0}^{N-1} k'_j \cdot \bar{q}_j^N(Z), \quad \text{and} \quad u'(Z) = \sum_{j'=0}^{N-1} u'_{j'} \cdot \bar{q}_{j'}^N(Z). \tag{57}$$

Let us define

$$k(Z) = \sum_{j=0}^{n-1} k_j \cdot \bar{q}_j^{\lfloor \frac{N}{2} \rfloor}(Z), \quad \text{and} \quad u(Z) = \sum_{j'=0}^{n-1} u_j \cdot \bar{q}_{j'}^{\lfloor \frac{N}{2} \rfloor}(Z). \tag{58}$$

Note that for $0 \le j < \lceil \frac{N}{2} \rceil$, the coefficients $k_j' = u_j' = 0$. Then (57) becomes

$$k'(Z) = \sum_{j=\lceil \frac{N}{2} \rceil}^{N-1} k_j' \cdot \bar{q}_j^N(Z), \quad \text{and} \quad u'(Z) = \sum_{j'=\lceil \frac{N}{2} \rceil}^{N-1} u_j' \cdot \bar{q}_{j'}^N(Z),$$

which is equivalent to

$$k'(Z) = \sum_{j=0}^{\lfloor \frac{N}{2} \rfloor-1} k'_{j+\lceil \frac{N}{2} \rceil} \cdot \bar{q}_{j+\lceil \frac{N}{2} \rceil}^N(Z), \quad \text{and} \quad u'(Z) = \sum_{j'=0}^{\lfloor \frac{N}{2} \rfloor-1} u'_{j'+\lceil \frac{N}{2} \rceil} \cdot \bar{q}_{j'+\lceil \frac{N}{2} \rceil}^N(Z).$$

For $0 \le j < \lfloor \frac{N}{2} \rfloor$, $\deg\left(\bar{q}_j^{\lfloor \frac{N}{2} \rfloor}\right) = \lfloor \frac{N}{2} \rfloor - j - 1$, and $\deg\left(\bar{q}_{j+\lceil \frac{N}{2} \rceil}^N\right) = N - \lceil \frac{N}{2} \rceil - j - 1 = \lfloor \frac{N}{2} \rfloor - j - 1$. This implies that $\bar{q}_j^{\lfloor \frac{N}{2} \rfloor}(Z)$ and $\bar{q}_{j+\lceil \frac{N}{2} \rceil}^N(Z)$ are both linear combinations of $\lfloor \frac{N}{2} \rfloor - j - 1$ Chebychev polynomials. Then we can set $\bar{q}_j^{\lfloor \frac{N}{2} \rfloor}(Z) = \bar{q}_{j+\lceil \frac{N}{2} \rceil}^N(Z)$. Similarly, note that for $0 \le j < n$, $k'_{j+\lceil \frac{N}{2} \rceil} = k_j$. Then it follows that $k(Z) = k'(Z)$, and by a similar argument, $u(Z) = u'(Z)$. Then by Theorem 12 we have

$$\begin{aligned} f(Z) &= k(Z) \cdot u(Z) \mod T_N(Z) \\ &= k(Z) \cdot u(Z) \end{aligned} \tag{59}$$

where the last statement follows since $\deg\left(k(Z)\right), \deg\left(u(Z)\right) \le n - 1 < \lfloor \frac{N}{2} \rfloor$, implying that their product has $\deg\left(\bar{k}(Z) \cdot \bar{u}(Z)\right) < \deg\left(T_N(Z)\right) = N$. We want to write $f(Z)$ in the form

$$f(Z) = \sum_{i=0}^{N-1} f_i \cdot \bar{q}_i^N(Z)$$

for some set of coefficients set of coefficients $f_i$. From (58), (59) becomes

$$f(Z) = \sum_{j=0}^{n-1} \sum_{j'=0}^{n-1} k_{j'} \cdot u_j \cdot \bar{q}_{j'}^{\lfloor \frac{N}{2} \rfloor}(Z) \cdot \bar{q}_j^{\lfloor \frac{N}{2} \rfloor}(Z).$$

Then by Lemma 8 we have

$$\sum_{i=0}^{N-1} f_i \cdot \bar{q}_i^N(Z) = \sum_{j,j'=0}^{n-1} k_{j'} \cdot u_j \cdot \sum_{i=j+j'}^{N-1} \alpha_{j+j',i} \bar{q}_i^N(Z).$$

We show that for all $0 \le i < N$, $f_i$ is a function of $(u_{i'})_{0 \le i' \le i}$. Then note from the above that each $k_j$ and $u_{j'}$ appears in terms of $\bar{q}_i^N$ where $i \ge j + j'$. Then we have

$$f_i = \sum_{\substack{j,j=0 \\ j+j' \le i}}^{N-1} \alpha_{j+j',i} \cdot k_{j'} \cdot u_j,$$

as desired.

$\square$

### D.5.2 Structured Causal Matrices

In this section we narrow our scope from the general class of matrices defined in Appendix D.5.1 to a particular structured subclass. Now let us define the class of structured causal matrices over the real numbers.

**Definition 6.** *Define*

$$\ell_{j_1}(Z) \stackrel{def}{=} \sum_{a=0}^{j_1} \ell_{j_1}[a] \, T_a(Z), \quad \tilde{r}_{j_1,j_0}(Z) \stackrel{def}{=} \sum_{a=0}^{j_0} \tilde{r}_{j_1,j_0}[a] \, T_a(Z) \tag{60}$$

*where*

$$\tilde{r}_{j_1,j_0}[a] = 0 \text{ if } (j_0 - a) \text{ is odd,} \tag{61}$$

*We define structured causal (SC) matrix polynomials as*

$$q^N_{(j_1,j_0)\sqrt{N}}(Z) = \ell_{\sqrt{N}-j_1-1}(Z) \cdot \tilde{r}_{\sqrt{N}-j_1-1,\sqrt{N}-j_0-1}\left(T_{\sqrt{N}}(Z)\right). \tag{62}$$

*A $N \times N$ SC matrix is defined over the set of real evaluation points as*

$$\mathbf{M}'[i,(j_0,j_1)] = q'^N_{(j_1,j_0)\sqrt{N}}(\omega_{N,i}) = q^N_{(j_0,j_1)\sqrt{N}}(\omega_{N,i}). \tag{63}$$

We note that in (63) we deviate from the usual ordering of indices $(j_1, j_0)$ to $(j_0, j_1)$. We show that the SC matrix falls under the category of matrices defined by (50).

**Lemma 9.** *Let $\mathcal{M}^C$ denote the set of all matrices defined by of (50), and $\mathcal{M}^{SC}$ denote the set of all matrices defined by Definition 6. Then $\mathcal{M}^{SC} \subset \mathcal{M}^C$.*

*Proof.* To show that $\mathcal{M}^{SC} \subset \mathcal{M}^C$, then it is sufficient to show that for $j = j_0\sqrt{N} + j_1$, any $q'^N_{(j_1,j_0)\sqrt{N}}(Z)$ is equivalent to some $\bar{q}^N_j(Z)$ as in (48). From (63) we have

$$q'^N_{(j_1,j_0)\sqrt{N}}(Z) = \ell_{\sqrt{N}-j_0-1}(Z) \cdot \tilde{r}_{\sqrt{N}-j_0-1,\sqrt{N}-j_1-1}\left(T_{\sqrt{N}}(Z)\right). \tag{64}$$

Note that

$$\begin{aligned}
\deg\left(q'^N_{(j_1,j_0)\sqrt{N}}\right) &= \sqrt{N} - j_0 - 1 + (\sqrt{N} - j_1 - 1)\sqrt{N} \\
&= N - \sqrt{N}j_1 - j_0 - 1 \\
&= N - j - 1.
\end{aligned}$$

From the fact that $\deg(T_a) = a$, we can use $T_a$ as a polynomial basis. Then $q'^N_{(j_1,j_0)\sqrt{N}}$ can be represented as a linear combination of $T_a(Z)$ like so:

$$q'^N_{(j_1,j_0)\sqrt{N}}(Z) = \sum_{a=0}^{N-j-1} q_{(j_1,j_0)}[a] T_a(Z),$$

which is exactly the form of (48).

$\square$

Lemma 9 allows us to apply the causality result from Appendix D.5.1.

**Corollary 7.** *Fix a family of basis polynomials $q^N_{0,0}, q^N_{0,1}, \ldots, q^N_{\sqrt{N}-1,\sqrt{N}-1}$ as defined in (62). For any perfect square $N \geq 1$, $n \leq \left\lfloor \frac{N}{2} \right\rfloor$, $\mathbf{k}, \mathbf{u} \in \mathbb{R}^n$ and $\mathbf{M}'_N$ defined by basis $\left(q'^N_{(j_1,j_0)\sqrt{N}}(Z)\right)_{0 \leq j_1,j_0 < \sqrt{N}}$ as in (63). Then the operation (55) with $\mathbf{M}_N \leftarrow \mathbf{M}'_N$ defines a causal map in $\mathbf{u}$.*

*Proof.* Follows from Theorem 13 and Lemma 9. $\square$

### D.5.3 Block Operations on Structured Causal Matrices

In this section we show how to build structured causal matrices through block operations.

**Constructing M**

Recall that in Appendix D.4.1, we defined $\mathbf{L}, \mathbf{R}$ in terms of coefficient matrices $\widetilde{\mathbf{R}}, \widetilde{\mathbf{L}} \in \mathbb{R}^{N \times \sqrt{N}}$ with blocks $\widetilde{\mathbf{R}}_k, \widetilde{\mathbf{L}}_k$ for $0 \leq k < \sqrt{N}$, noting that for each block $\mathbf{R}_{j_1, j_1}$,

$$\mathbf{R}_{j_1, j_1} = \mathbf{F}_{\sqrt{N}} \cdot \widetilde{\mathbf{R}}_{j_1, j_1},$$

and for $\widetilde{\mathbf{L}} \in \mathbb{R}^{N \times \sqrt{N}}$

$$\mathbf{L}' = \mathbf{P}\mathbf{F}_N \cdot \widetilde{\mathbf{L}}.$$

These matrices are then diagonalized into $\mathbf{L}, \mathbf{R}$. We use Definition 6 to similarly define $\mathbf{L}, \mathbf{R}, \in \mathbb{R}^{N \times N}$ with blocks $\{\mathbf{L}_{j_1, j_0}\}_{0 \leq j_1, j_0 < \sqrt{N}}$, $\{\mathbf{R}_{j_1, j_0}\}_{0 \leq j_1, j_0 < \sqrt{N}}$ where:

$$\mathbf{L}_{i_1, j_1}[i_0, i_0] = \ell_{\sqrt{N} - j_1 - 1}(\omega_{N, i}) \qquad \mathbf{R}_{j_1, j_1}[i_0, j_0] \leftarrow \tilde{r}_{\sqrt{N} - j_1 - 1, \sqrt{N} - j_0 - 1}(\omega_{N, i}), \qquad (65)$$

and all other entries are zero. Let the coefficient matrices $\widetilde{\mathbf{L}}, \widetilde{\mathbf{R}} \in \mathbb{R}^{N \times \sqrt{N}}$ be defined with respect to blocks as follows:

$$\widetilde{\mathbf{L}}[a, j_1] = \ell_{\sqrt{N} - j_1 - 1}[a] \qquad \widetilde{\mathbf{R}}_{j_1}[a, j_0] = \tilde{r}_{\sqrt{N} - j_1 - 1, \sqrt{N} - j_0 - 1}[a],$$

where the entries of $\ell_{\sqrt{N} - j_1 - 1}$ and $\tilde{r}_{\sqrt{N} - j_1 - 1, \sqrt{N} - j_0 - 1}$ are defined as in (60) and (61). Now let $\mathbf{C}_N \in \mathbb{R}^{N \times N}$ where

$$\mathbf{C}_N[i, j] = T_j(\omega_{N, i}) \qquad (66)$$

be the Chebyshev transform. Then analogous to Appendix D.4.1, we define

$$\mathbf{R}_{j_1, j_1} = \mathbf{C}_{\sqrt{N}} \cdot \widetilde{\mathbf{R}}_{j_1, j_1} \qquad \mathbf{L}' = \mathbf{P}\mathbf{C}_N \cdot \widetilde{\mathbf{L}}.$$

This allows us to give an algorithm the following construction algorithm for $\mathbf{L}$ and $\mathbf{R}$.

---

**Algorithm 4** BLOCKSC($\widetilde{\mathbf{L}}, \widetilde{\mathbf{R}}$)

---

**Input:** $\widetilde{\mathbf{L}}, \widetilde{\mathbf{R}} \in \mathbb{R}^{N \times \sqrt{N}}$
**Output:** Block diagonal matrices $\mathbf{L}, \mathbf{R} \in \mathbb{R}^{N \times N}$

▷ First, compute $\mathbf{L}$ from $\widetilde{\mathbf{L}}$

1: $\mathbf{L}' \leftarrow \mathbf{P} \cdot \mathbf{C}_N \cdot \widetilde{\mathbf{L}}$
2: **for** $a \leftarrow 0$ to $\sqrt{N} - 1$ **do**
3: $\quad \mathbf{L}'_a \leftarrow \overline{\mathbf{L}}'[a\sqrt{N} : a\sqrt{N} + \sqrt{N} - 1, :]$
4: $\mathbf{L} \leftarrow \text{diag}(\mathbf{L}'_0, \ldots \mathbf{L}'_{\sqrt{N} - 1})$

▷ Now compute $\mathbf{R}$ from $\widetilde{\mathbf{R}}$

5: **for** $a \leftarrow 0$ to $\sqrt{N} - 1$ **do**
6: $\quad \mathbf{R}_a \leftarrow \mathbf{C}_{\sqrt{N}} \cdot \widetilde{\mathbf{R}}[a\sqrt{N} : a\sqrt{N} + \sqrt{N} - 1, :]$
7: $\mathbf{R} \leftarrow \text{diag}\left(\mathbf{R}_0, \ldots, \mathbf{R}_{\sqrt{N} - 1}\right)$
8: **return** $\mathbf{L}, \mathbf{R}$

---

We use Algorithm 4 to specify another type of matrix $\mathbf{M} \in \mathbb{R}^{N \times N}$.

**Lemma 10.** *Let*

$$\mathbf{M} = \mathbf{P}\mathbf{L}\mathbf{P}\mathbf{R}\mathbf{P}$$

*where $\mathbf{L}$ and $\mathbf{R}$ are outputs from BLOCKSC. Then each entry in $\mathbf{M}$ is defined as*

$$\mathbf{M}[i, j] = \ell_{\sqrt{N} - j_0 - 1}(\omega_{N, i}) \cdot \tilde{r}_{\sqrt{N} - j_0 - 1, \sqrt{N} - j_1 - 1}\left(\omega_{\sqrt{N}, i_0}\right) \qquad (67)$$

*Proof.* Let $\mathbf{M}_0 = \mathbf{PLPR}$, then $\mathbf{M} = \mathbf{M}_0\mathbf{P}$. Then we have

$$\mathbf{M}_0[(i_1, i_0), (j_1, j_0)] = \ell_{\sqrt{N}-j_1-1}(\omega_{N,i}) \cdot \tilde{r}_{\sqrt{N}-j_1-1, \sqrt{N}-j_0-1}(\omega_{N,i_0}).$$

Then we get

$$\mathbf{M}[i, j] = \mathbf{M}_0\mathbf{P}[(i_1, i_0), (j_1, j_0)] = \mathbf{M}_0[(i_1, i_0), (j_0, j_1)].$$

This gives us

$$\mathbf{M}[(i_1, i_0), (j_1, j_0)] = \ell_{\sqrt{N}-j_0-1}(\omega_{N,i}) \cdot \tilde{r}_{\sqrt{N}-j_0-1, \sqrt{N}-j_1-1}(\omega_{N,i_0}).$$

as desired. □

Next we will discuss the relation between matrices that we get from Lemma 10 and SC matrices.

### D.5.4 Constructing $\mathbf{M}'$ from $\mathbf{M}$

Since $\mathbf{C}_N \notin \mathcal{M}^{SC}$ [7], Algorithm 4 is not a proper analog of Algorithm 2. In this section, we show how to convert $\mathbf{M}$ produced from Algorithm 4 into a matrix with a block decomposible form. Recall from (63) that

$$\mathbf{M}'[i, j] = q'^N_{(j_1, j_0)\sqrt{N}}(\omega_{N,i}) = \ell_{\sqrt{N}-j_0-1}(Z) \cdot \tilde{r}_{\sqrt{N}-j_0-1, \sqrt{N}-j_1-1}\left(T_{\sqrt{N}}(Z)\right). \tag{68}$$

We note the distinction between the above and (67). Specifically, we note that $\mathbf{M}$ is evaluated on two sets of evaluation points– the $\ell_{\sqrt{N}-j_1-1}(Z)$ and $\tilde{r}_{\sqrt{N}-j_1-1, \sqrt{N}-j_0-1}$ polynomials are evaluated over the $N$ roots and $\sqrt{N}$ roots of unity while $\mathbf{M}'$ is only evaluated the $N$-th roots of unity. However, they are close due to the following property:

**Lemma 11.** *Let $T_a$ be a Chebyshev polynomial of the first kind of degree $a$, and define $\omega_{N,i}$ as in (49). Then*

$$T_{\sqrt{N}}(\omega_{N,i}) = (-1)^{i_1} \cdot \omega_{\sqrt{N}, i_0}.$$

*Proof.*

$$
\begin{aligned}
T_{\sqrt{N}}(\omega_{N,i}) &= \cos\left(\frac{\sqrt{N}(i_1\sqrt{N} + i_0 + \frac{1}{2})\pi}{N}\right) \\
&= \cos\left(i_1\pi + \frac{\pi(i_0 + \frac{1}{2})}{\sqrt{N}}\right) \\
&= (-1)^{i_1}\cos\left(\frac{\pi(i_0 + \frac{1}{2})}{\sqrt{N}}\right) \\
&= (-1)^{i_1} \cdot \omega_{\sqrt{N}, i_0}.
\end{aligned}
$$

In the above the first equality follows from (46). □

Analogous to how we utilized roots of unity, Lemma 11 allows us to express our basis polynomials in terms of two related sets of evaluation points, $\omega_{\sqrt{N}, i_0}$ and $\omega_{N,i}$. The following lemma demonstrates how to translate between $\mathbf{M}'$ and $\mathbf{M}$.

**Lemma 12.** *For all $i, j$,*

$$\mathbf{M}'[i, j] = (-1)^{i_i(\sqrt{N}-1)}(-1)^{i_i j_1} \cdot \mathbf{M}[i, j]. \tag{69}$$

---

[7]We do not have a proof of this claim, but to us it seems unlikely that $\mathbf{C}_N \in \mathcal{M}^{SC}$

*Proof.* By (68) we have

$$\mathbf{M}'[i,j] = \ell_{\sqrt{N}-j_0-1}(\omega_{N,i}) \cdot \tilde{r}_{\sqrt{N}-j_0-1,\sqrt{N}-j_1-1}\left(T_{\sqrt{N}}(\omega_{N,i})\right)$$

$$= \ell_{\sqrt{N}-j_0-1}(\omega_{N,i}) \cdot \tilde{r}_{\sqrt{N}-j_0-1,\sqrt{N}-j_1-1}\left((-1)^{i_1}\omega_{\sqrt{N},i_0}\right)$$

where the second statement follows from Lemma 11. Then from (60) and (47) we get

$$\mathbf{M}'[i,j] = \ell_{\sqrt{N}-j_0-1}(\omega_{N,i}) \cdot \sum_{a=0}^{\sqrt{N}-j_1-1} \tilde{r}_{\sqrt{N}-j_0-1,\sqrt{N}-j_1-1}[a]\, T_a\left((-1)^{i_1}\omega_{\sqrt{N},i_0}\right)$$

$$= \ell_{\sqrt{N}-j_0-1}(\omega_{N,i}) \cdot \sum_{a=0}^{\sqrt{N}-j_1-1} \cdot\tilde{r}_{\sqrt{N}-j_0-1,\sqrt{N}-j_1-1}[a](-1)^{i_1 a}\, T_a\left(\omega_{\sqrt{N},i_0}\right).$$

Note from (61) that $\tilde{r}_{\sqrt{N}-j_0-1,\sqrt{N}-j_1-1}[a] = 0$ if $(\sqrt{N} - j_1 - 1 - a)$ is odd. Then equivalently we get

$$\mathbf{M}'[i,j] = \ell_{\sqrt{N}-j_0-1}(\omega_{N,i}) \cdot (-1)^{i_1\left(\sqrt{N}-j_1-1\right)}$$

$$\cdot \sum_{a=0}^{\sqrt{N}-j_1-1} \tilde{r}_{\sqrt{N}-j_0-1,\sqrt{N}-j_1-1}[a]\,(-1)^{i_1\left(\sqrt{N}-j_1-1-a\right)} T_a\left(\omega_{\sqrt{N},i_0}\right)$$

$$= \ell_{\sqrt{N}-j_0-1}(\omega_{N,i}) \cdot (-1)^{i_1\left(\sqrt{N}-j_1-1\right)} \left(\sum_{a=0}^{\sqrt{N}-j_1-1} \tilde{r}_{\sqrt{N}-j_0-1,\sqrt{N}-j_1-1}[a]\, T_a\left(\omega_{\sqrt{N},i_0}\right)\right)$$

$$= (-1)^{i_1\left(\sqrt{N}-j_1-1\right)} \ell_{\sqrt{N}-j_0-1}(\omega_{N,i}) \cdot \tilde{r}_{\sqrt{N}-j_0-1,\sqrt{N}-j_1-1}\left(\omega_{\sqrt{N},i_0}\right).$$

Then from (67) we have

$$\mathbf{M}'[i,j] = (-1)^{i_1\left(\sqrt{N}-j_1-1\right)}\mathbf{M}[i,j]$$

$$= (-1)^{i_1\left(\sqrt{N}-1\right)}(-1)^{i_1 j_1}\mathbf{M}[i,j].$$

$\square$

**Block Matrix Multiplication for Structured Causal Matrices** Lemma 12 shows us how to compute $\mathbf{M}'$ from $\mathbf{M}$. However, this does not mean that the matrix vector multiplication problem for $\mathbf{M}'$ can be implemented efficiently. In this section we show how to perform matrix vector multiplication of $\mathbf{M}'$ with any $\mathbf{u} \in \mathbb{R}^N$ from two matrix-vector multiplications of $\mathbf{M}$ (and so these operations are indeed efficient).

The key to our approach involves considering the parity of each block index $0 \le i_1 < \sqrt{N}$. Let us define a map $\text{MIX} : \mathbb{R}^{\lfloor\frac{N}{2}\rfloor} \times \mathbb{R}^{\lceil\frac{N}{2}\rceil} \mapsto \mathbb{R}^N$ such that $\text{MIX}(\mathbf{u}_0, \mathbf{u}_1) = \mathbf{u}$ where

$$\mathbf{u}[i] = \mathbf{u}_{i_1 \mod 2}[i/2].$$

We use this map to show that $\mathbf{M}'\mathbf{u}$ and $\mathbf{u}^\top\mathbf{M}'$ can be computed efficiently.

**Lemma 13.** *For any $\mathbf{u} \in \mathbb{R}^N$ and $\mathbf{M} \in \mathbb{R}^{N \times N}$, $\mathbf{M}'\mathbf{u}$ can be computed via two matrix-vector multiplications: $\mathbf{M} \cdot \text{MIX}(\mathbf{u}_0, \mathbf{0})$ and $\mathbf{M} \cdot \text{MIX}(\mathbf{0}, \mathbf{u}_1)$.*

*Proof.* For $0 \le i = i_1\sqrt{N} + i_0 < N$, $0 \le j = j_1\sqrt{N} + j_0 < N$, let $\mathbf{D} \in \mathbb{R}^{N \times N}$ be the diagonal matrix defined such that $\mathbf{D}[i,i] = (-1)^{i_1\left(\sqrt{N}-1\right)}$. Lemma 12 implies that

$$\mathbf{M}'[i,j] = \begin{cases} \mathbf{DM}[i,j] & \text{if } i_1 \text{ is even} \\ \mathbf{DM}[i,j] & \text{if } i_1 \text{ is odd, } j_1 \text{ is even} \\ -(\mathbf{DM}[i,j]) & \text{otherwise.} \end{cases}$$

We want to shift the $(-1)$ to $\mathbf{u}$. Compute $\mathbf{z}_0 = \mathbf{M} \cdot (\text{Mix}\,(\mathbf{u}_0, \mathbf{0}) + \text{Mix}\,(\mathbf{0}, \mathbf{u}_1))$, $\mathbf{z}_1 = \mathbf{M} \cdot (\text{Mix}\,(\mathbf{u}_0, \mathbf{0}) - \text{Mix}\,(\mathbf{0}, \mathbf{u}_1))$. Define $\mathbf{y}'$ such that

$$\mathbf{y}'[j] = \begin{cases} \mathbf{z}_0[j] & \text{if } j_1 \text{ is even} \\ \mathbf{z}_1[j] & \text{if } j_1 \text{ is odd} \end{cases}.$$

It can be verified that $\mathbf{D} \cdot \mathbf{y}' = \mathbf{M}'\mathbf{u}$, which completes the proof. $\qquad\square$

We now give the analogous result for $\mathbf{u}^\top \mathbf{M}'$.

**Lemma 14.** *For any $\mathbf{u} \in \mathbb{R}^N$ and $\mathbf{M} \in \mathbb{R}^{N \times N}$, $\mathbf{u}^\top \mathbf{M}'$ can be computed via two matrix-vector multiplications:* $\text{Mix}\,(\mathbf{u}_0, \mathbf{0})^\top \mathbf{M}$ *and* $\text{Mix}\,(\mathbf{0}, \mathbf{u}_1)^\top \mathbf{M}$.

*Proof.* For $0 \le i = i_1\sqrt{N} + i_0 < N$, $0 \le j = j_1\sqrt{N} + j_0 < N$, let $\mathbf{D} \in \mathbb{R}^{N \times N}$ be the diagonal matrix defined such that $\mathbf{D}[i, i] = (-1)^{i_1(\sqrt{N}-1)}$, and $\mathbf{u}' = \mathbf{u}^\top \mathbf{D} = \text{Mix}\,(\mathbf{u}'_0, \mathbf{u}'_1)$. Lemma 12 implies that

$$\mathbf{M}'[i, j] = \begin{cases} \mathbf{DM}[i, j] & \text{if } i_1 \text{ is even} \\ \mathbf{DM}[i, j] & \text{if } i_1 \text{ is odd, } j_1 \text{ is even} \\ -(\mathbf{DM}[i, j]) & \text{otherwise.} \end{cases}$$

We want to shift the $(-1)$ to $\mathbf{u}'$. Compute $\mathbf{z}_0 = \left( \text{Mix}\,(\mathbf{u}'_0, \mathbf{0})^\top + \text{Mix}\,(\mathbf{0}, \mathbf{u}'_1)^\top \right) \cdot \mathbf{M}$, $\mathbf{z}_1 = \left( \text{Mix}\,(\mathbf{u}'_0, \mathbf{0})^\top - \text{Mix}\,(\mathbf{0}, \mathbf{u}'_1)^\top \right) \cdot \mathbf{M}$. If $\mathbf{y} = \mathbf{M}'\mathbf{u}$, then one can check that

$$\mathbf{y}[j] = \begin{cases} \mathbf{z}_0[j] & \text{if } j_1 \text{ is even} \\ \mathbf{z}_1[j] & \text{if } j_1 \text{ is odd} \end{cases},$$

which completes the proof. $\qquad\square$

Lemma 13 implies that computing $\mathbf{M}_N \mathbf{k}$ and $\mathbf{M}_N \mathbf{u}$ in (55) can be done efficiently. However, we do not know how to compute $\mathbf{M}_N^{-1}\mathbf{y}$ for any arbitrary $\mathbf{y}$. We address this partially in the next section.

### D.5.5 Inverse of Chebyschev Transform

In this subsection we show how computing $\mathbf{C}_N^{-1}$ can be done efficiently.

**Lemma 15.** *Let $\mathbf{C}_N \in \mathbb{R}^{N \times N}$ be defined as* (66). *Then*

$$\mathbf{C}_N^{-1} = \mathbf{C}_N^\top \cdot diag(1/N, 2/N, \dots, 2/N).$$

*Proof.* Follows since $\mathbf{C}_N (\mathbf{C}_N)^\top = \text{diag}(N, N/2, N/2, \dots, N/2)$ [71]. $\qquad\square$

Given the above, the goal then is to compute $\mathbf{C}_N^\top \cdot \mathbf{u}$ or equivalently $\mathbf{u}^\top \mathbf{C}_N$ for any $\mathbf{u} \in \mathbb{R}^N$. Note that is sufficient to show that

$$\mathbf{C}_N = \overline{\mathbf{C}}_N + \overline{\mathbf{S}}_N$$

for $\overline{\mathbf{C}}_N, \overline{\mathbf{S}}_N \in \mathbb{R}^{N \times N}$ such that $\mathbf{u}^\top \overline{\mathbf{C}}_N$ and $\mathbf{u}^\top \overline{\mathbf{S}}_N$ can be computed with two invocations of Lemma 14. Indeed we have

$$\begin{aligned}
\mathbf{C}_N[i, j] &= \cos\left( \frac{(i + \frac{1}{2})(j_1\sqrt{N} + j_0)\pi}{N} \right) \\
&= \cos\left( \frac{\pi(i + \frac{1}{2})j_1}{\sqrt{N}} + \frac{\pi(i + \frac{1}{2})j_0}{N} \right), \\
&= \cos\left( \frac{\pi(i + \frac{1}{2})j_1}{\sqrt{N}} \right) \cos\left( \frac{\pi(i + \frac{1}{2})j_0}{N} \right) - \sin\left( \frac{\pi(i + \frac{1}{2})j_1}{\sqrt{N}} \right) \sin\left( \frac{\pi(i + \frac{1}{2})j_0}{N} \right).
\end{aligned}$$
$$(70)$$

We use this to define $\overline{\mathbf{C}}_N$ and $\overline{\mathbf{S}}_N$, along with two additional matrices $\widetilde{\mathbf{C}}_N, \widetilde{\mathbf{S}}_N \in \mathbb{R}^{N \times N}$ such that for $\overline{\mathbf{C}}_N$ and $\widetilde{\mathbf{C}}_N$:

$$
\begin{aligned}
\overline{\mathbf{C}}_N[i, j] &= \cos\left(\frac{\pi(i + \frac{1}{2})j_1}{\sqrt{N}}\right) \cos\left(\frac{\pi(i + \frac{1}{2})j_0}{N}\right) \\
&= \cos\left(\pi i_1 j_1 + \frac{\pi(i_0 + \frac{1}{2})j_1}{\sqrt{N}}\right) \cos\left(\frac{\pi(i + \frac{1}{2})j_0}{N}\right) \\
&= (-1)^{i_1 j_1} \cos\left(\frac{\pi(i_0 + \frac{1}{2})j_1}{\sqrt{N}}\right) \cos\left(\frac{\pi(i + \frac{1}{2})j_0}{N}\right) \overset{\text{def}}{=} (-1)^{i_1 j_1} \widetilde{\mathbf{C}}_N[i, j], \quad (71)
\end{aligned}
$$

and similarly for $\overline{\mathbf{S}}_N$ and $\widetilde{\mathbf{S}}_N$:

$$
\begin{aligned}
\overline{\mathbf{S}}_N[i, j] &= \sin\left(\frac{\pi(i + \frac{1}{2})j_1}{\sqrt{N}}\right) \sin\left(\frac{\pi(i + \frac{1}{2})j_0}{N}\right) \\
&= (-1)^{i_1 j_1} \sin\left(\frac{\pi(i_0 + \frac{1}{2})j_1}{\sqrt{N}}\right) \sin\left(\frac{\pi(i + \frac{1}{2})j_0}{N}\right) \overset{\text{def}}{=} (-1)^{i_1 j_1} \widetilde{\mathbf{S}}_N[i, j]. \quad (72)
\end{aligned}
$$

We summarize these results into the following lemma.

**Lemma 16.** *Define $\overline{\mathbf{C}}_N, \widetilde{\mathbf{C}}_N, \overline{\mathbf{S}}_N$ and $\widetilde{\mathbf{S}}_N$ as in (71) and (72), respectively. Then*

$$
\mathbf{C}_N = \overline{\mathbf{C}}_N - \overline{\mathbf{S}}_N
$$

*where $\widetilde{\mathbf{C}}_N$ and $\widetilde{\mathbf{S}}_N$ are of the form (67).*

Finally Lemma 14 and Lemma 16 imply the following:

**Theorem 14.** *For any $\mathbf{u} \in \mathbb{R}^N$,*

$$
\mathbf{y} = \mathbf{u}^\top \mathbf{C}_N
$$

*can be computed from four calls matrix vector multiplication with Monarch matrices.*

**Proof Sketch for Theorem 14**   Lemma 16 tells us that $\mathbf{u}^\top \overline{\mathbf{C}}_N$ and $\mathbf{u}^\top \overline{\mathbf{S}}_N$ are sufficient to compute $\mathbf{u}^\top \mathbf{C}_N$, and (72) shows us that $\widetilde{\mathbf{C}}_N, \widetilde{\mathbf{S}}_N$ are Monarch matrices that allows $\mathbf{u}^\top \overline{\mathbf{C}}_N$ and $\mathbf{u}^\top \overline{\mathbf{S}}_N$ to be computed from two matrix- vector multiplications with $\widetilde{\mathbf{C}}_N$ and $\widetilde{\mathbf{S}}_N$, respectively. Then the claim follows from applying Lemma 14 twice.

### D.6   Multivariate MONARCH MIXER with Block Size $= \sqrt[p]{N}$

In this section, we generalize Theorem 1, Theorem 2, and Theorem 3 to arbitrary $p \geq 2$. In Appendix D.6.1, we set up notation. We then generalize Theorem 1 and Theorem 2 to general $p$ in Appendix D.6.2 and then generalize Theorem 3 in Appendix D.6.3. In Appendix D.6.4 we connect definition of $p$-variate Monarch in Appendix D.6.2 to the one defined in Section 3. Finally in Appendix D.6.5 we discuss some extensions and generalizations.

#### D.6.1   Notation

We will use notation from Appendix D.5.1.

Fix an integer $p \geq 2$. We will be working with indices $\mathbf{j}, \mathbf{i}$ where $\mathbf{j} = (j_0, \ldots, j_{p-1})$ and $\mathbf{i} = (i_0, \ldots, i_{p-1})$ with $0 \leq i_a, j_a < \sqrt[p]{N}$ for every $0 \leq a < p$. We will denote the set of all sub-indices as $[0, \sqrt[p]{N})^p$, and the operator $\preceq$ to denote the lexicographical ordering of vectors in $[0, \sqrt[p]{N})^p$.

For any $0 \leq b' \leq M \leq N$ such that $b'$ divides $M$ and $M$ divides $N$, define the following permutation matrices:

$$
\mathbf{P}_{b', M, N} = \text{diag}\left(\underbrace{\mathbf{P}_{b', M}, \ldots, \mathbf{P}_{b', M}}_{\frac{N}{M} \text{ times}}\right).
$$

Note that $\mathbf{P}_{b',N,N}$ is exactly the same as $\mathbf{P}_{b',N}$ from earlier.

If we use $\sigma(b, M, N)$ to denote the corresponding permutation then it takes the input $(i_{p-1}, \ldots, i_0)_b$ and maps it to $(i_{p-1}, \ldots, i_{p'}, i_{p'-2}, \ldots, i_0, i_{p'})_b$ (where $p = \log_b N$ and $p' = \log_b M$). I.e. $\sigma(b, M, N)$ does not change the first $p - p'$ sub-indices and then does a left rotation on the remaining $p'$ sub-indices.

For the rest of the section, assume $b = \sqrt[p]{N}$ and then consider the following 'sub-index reversal' permutation matrix[8]:

$$\mathbf{P}^R_{b,b^p} = \prod_{a=0}^{p-2} \mathbf{P}_{b^{p-a-1},b^{p-a},b^p}.$$

If $\sigma^R(b, N)$ is the permutation corresponding to the permutation matrix above, then $\sigma^R(b, N)$ maps $(i_{p-1}, \ldots, i_0) \mapsto (i_0, \ldots, i_{p-1})$.

### D.6.2 Generalizing Theorem 1 and Theorem 2

For a specific $0 \le a < p$, let

$$\ell^{(a)}_{\mathbf{j},\mathbf{i}}(X_a) = \sum_{m=0}^{\sqrt[p]{N}-1} \ell^{(a)}_{(j_{a+1},\ldots,j_{p-1}),(i_0,\ldots,i_{a-1})}[m] \cdot T_m(X_a) \tag{73}$$

be an arbitrary polynomial of degree $< \sqrt[p]{N}$ in the Chebyshev polynomial basis (see (47)).

We will be interested in the evaluations of the above polynomials over the set

$$A \stackrel{\text{def}}{=} \left( \omega_{\sqrt[p]{N},0}, \ldots, \omega_{\sqrt[p]{N},\sqrt[p]{N}-1} \right), \tag{74}$$

where $\omega_{\sqrt{N},i}$ is defined as in (??).

The $p$-variate version of Monarch matrices $\mathbf{M}' \in \mathbb{R}^{N \times N}$ as follows. (See Appendix D.6.4 to see how these are exactly related to the definition in (1)). For every row index $\mathbf{i} \in [0, \sqrt[p]{N})^p$ and column index $\mathbf{j} \in [0, \sqrt[p]{N})^p$, we have

$$\mathbf{M}'[\mathbf{i}, \mathbf{j}] = \prod_{a=0}^{p-1} \ell^{(a)}_{\mathbf{j},\mathbf{i}} \left( \omega_{\sqrt[p]{N},i_a} \right). \tag{75}$$

To express the above in terms of a polynomial basis we will need the following definition. For any $0 \le a < p$ and $\mathbf{i}$ define the Lagrange basis polynomial $\Delta^{(a)}_{\mathbf{i}}(X_0, \ldots, X_{a-1})$ such that for any $0 \le m_0, \ldots, m_{a-1} < \sqrt[p]{N}$, we have

$$\Delta^{(a)}_{\mathbf{i}} \left( \omega_{\sqrt[p]{N},m_0}, \ldots, \omega_{\sqrt[p]{N},m_{a-1}} \right) = \begin{cases} 1 & \text{if } i_0 = m_0, i_1 = m_1, \ldots, i_{a-1} = m_{a-1} \\ 0 & \text{otherwise.} \end{cases}$$

We use the above to convert the polynomials in (73) to not depend on the sub-indices in $\mathbf{i}$ (at least for the definition in (75)). For every $\mathbf{j}$ and $0 \le a < p$, define:

$$\ell^{(a)}_{\mathbf{j}}(X_0, \ldots, X_a) = \sum_{\mathbf{i}=(i_0,\ldots,i_{a-1},\mathbf{0}_{p-a}),i_0,\ldots,i_{a-1}\in[0,\sqrt[p]{N})} \Delta^{(a)}_{\mathbf{i}}(X_0, \ldots, X_{a-1}) \cdot \ell^{(a)}_{\mathbf{j},\mathbf{i}}(X_a).$$

Note that the summation fixes the last $p - a$ sub-indices in $\mathbf{i}$ since the definition of $\ell^{(a)}_{\mathbf{j},\mathbf{i}}(X_a)$ only depends on $(i_0, \ldots, i_{a-1})$. This implies that for any $0 \le i_0, \ldots, i_a < \sqrt[p]{N}$, we have

$$\ell^{(a)}_{\mathbf{j}} \left( \omega_{\sqrt[p]{N},i_0}, \ldots, \omega_{\sqrt[p]{N},i_a} \right) = \ell^{(a)}_{\mathbf{j},\mathbf{i}} \left( \omega_{\sqrt[p]{N},i_a} \right). \tag{76}$$

---

[8] $b = 2$ gives the well known bit reversal permutation.

We are now ready to define our basis $p$-variate polynomials. For any index $\mathbf{j} \in [0, \sqrt[p]{N})^p$, define

$$q_\mathbf{j}^N(X_0, \ldots, X_{p-1}) = \prod_{a=0}^{p-1} \ell_\mathbf{j}^{(a)}(X_0, \ldots, X_a). \tag{77}$$

Then (75) can be re-written as

$$\mathbf{M}[\mathbf{i}, \mathbf{j}] = q_\mathbf{j}^N\left(\omega_{\sqrt[p]{N}, i_0}, \ldots, \omega_{\sqrt[p]{N}, i_{p-1}}\right). \tag{78}$$

The above leads to the following result, which generalizes Theorem 1 to general $p$:

**Theorem 15.** *Let* $\mathbf{M}'$, $A$ *and* $q_\mathbf{j}^N(X_0, \ldots, X_{p-1})$ *be as defined in* (75), (74) *and* (77)*. Then for any vector* $\mathbf{u}$, $\mathbf{M} \cdot \mathbf{u}$ *is equivalent to evaluating the polynomial*

$$u(X_0, \ldots, X_{p-1}) = \sum_\mathbf{j} u_\mathbf{j} \cdot q_\mathbf{j}^N(X_0, \ldots, X_{p-1}) \tag{79}$$

*at each point in* $A^p$.

*Proof.* By (78), the column $\mathbf{M}'[:, j]$ is exactly the evaluation of the polynomial (77) at each point in $A^p$. Then the claim follows from the definition of matrix vector multiplication and (79). □

This then leads to the following result, (which generalizes Theorem 2):

**Theorem 16.** *For matrices* $\mathbf{M}_0, \mathbf{M}_1, \mathbf{M}_2$, *each of form as in* Theorem 15, *the operation*

$$\mathbf{f} = \mathbf{M}_0^{-1} \cdot ((\mathbf{M}_1 \cdot \mathbf{k}) \odot (\mathbf{M}_2 \cdot \mathbf{u})). \tag{80}$$

*is equivalent to representing the polynomial*

$$f(\mathbf{X}) = k(\mathbf{X}) \cdot u(\mathbf{X}) \mod \left(T_{\sqrt[p]{N}}(X_0), \ldots, T_{\sqrt[p]{N}}(X_{p-1})\right)$$

*in terms of the basis polynomials*

$$\hat{q}_\mathbf{j}^N(\mathbf{X})$$

*where* $k(\mathbf{X}), u(\mathbf{X})$ *are defined in terms of the respective basis polynomials corresponding to* $\mathbf{M}_1$ *and* $\mathbf{M}_2$ *as in* Theorem 15, *and* $\hat{q}_\mathbf{j}^N(\mathbf{X})$s *corresponds to* $\mathbf{M}_0$.

*Proof.* Define

$$q_A(Z) = \prod_{\alpha \in A}(Z - \alpha). \tag{81}$$

Then (80) follows since for

$$f(\mathbf{X}) = k(\mathbf{X}) \cdot u(\mathbf{X}) \mod (q_A(X_0), \ldots q_A(X_{p-1})),$$

we have the following for any $\mathbf{a} \in A^p$ we have:

$$f(\mathbf{a}) = k(\mathbf{a}) \cdot u(\mathbf{a}).$$

Using the known fact that $T_{\sqrt[p]{N}}(Z) = \prod_{c=0}^{\sqrt[p]{N}-1}\left(Z - \omega_{\sqrt[p]{N}, c}\right) = q_A(Z)$, the claim follows from Theorem 15, (80), and the invertibility of $\mathbf{M}_0$. □

### D.6.3 Generalizing Theorem 3 for $p \geq 2$

To convert Theorem 16 into a causal map we basically have to blow up $n \to 2^p \cdot n$. Further, paralleling (48) we need to change the definition in (73) to have degree $\sqrt[p]{N} - j_a - 1$ instead of the earlier $\sqrt[p]{N} - 1$:

$$\widetilde{\ell}_{\mathbf{j}, \mathbf{i}}^{(a)}(X_a) = \sum_{m=0}^{\sqrt[p]{N}-j_a-1} \widetilde{\ell}_{(j_a)}^{(a)}[m] \cdot T_m(X_a). \tag{82}$$

Note that now the RHS only depends on $a$ and $j_a$ (let us call the RHS $\widetilde{\ell}_{j_a}^{\left(a,\,\sqrt[p]{N}\right)}(X_a)$), so the next definition becomes easier:

$$\widetilde{\ell}_{\mathbf{j}}^{(a)}(X_0,\ldots,X_a) = \widetilde{\ell}_{j_a}^{\left(a,\,\sqrt[p]{N}\right)}(X_a).$$

We are now ready to define our causal basis polynomials. For any index $\mathbf{j}$, define

$$\widetilde{q}_{\mathbf{j}}^N(X_0,\ldots,X_{p-1}) = \prod_{a=0}^{p-1} \widetilde{\ell}_{\mathbf{j}}^{(a)}(X_0,\ldots,X_a). \tag{83}$$

These polynomials form a structured subclass of the polynomials defined in (77).

**Lemma 17.** *The class of polynomials $\widetilde{q}_{\mathbf{j}}^N(X_0,\ldots,X_{p-1})$ defined in (83) are a special case of (77).*

*Proof.* This follows from the fact that (82) is a special case of (73) for every $\mathbf{i},\mathbf{j}, 0 \leq a < p$. $\qquad\square$

We show that the product of two $\widetilde{q}_{\mathbf{j}}^{\left\lfloor \frac{\sqrt[p]{N}}{2} \right\rfloor^p}(X_0,\ldots,X_{p-1})$ type polynomials can be written has a linear combination of $\widetilde{q}_{\mathbf{m}}^N(X_0,\ldots,X_{p-1})$ with the indices of $\mathbf{m}$ being lexicographically larger than the original indices.

**Lemma 18.** *Let $\widetilde{q}_{\mathbf{j}}^{\left\lfloor \frac{\sqrt[p]{N}}{2} \right\rfloor^p}(X_0,\ldots,X_{p-1})$ be defined as in (83). Then for any $\mathbf{j},\mathbf{j}' \in \left[0,\frac{\sqrt[p]{N}}{2}\right)^p$,*

$$\widetilde{q}_{\mathbf{j}}^{\left\lfloor \frac{\sqrt[p]{N}}{2} \right\rfloor^p}(X_0,\ldots,X_{p-1})\cdot\widetilde{q}_{\mathbf{j}'}^{\left\lfloor \frac{\sqrt[p]{N}}{2} \right\rfloor^p}(X_0,\ldots,X_{p-1}) = \sum_{\mathbf{j}+\mathbf{j}'\preceq\mathbf{m}\in[0,\,\sqrt[p]{N})^p} \alpha_{\mathbf{j}+\mathbf{j}',\mathbf{m}}\,\widetilde{q}_{\mathbf{m}}^N(X_0,\ldots,X_{p-1}) \tag{84}$$

*for some set of coefficients $\alpha_{\mathbf{j}+\mathbf{j}',\mathbf{m}}$.*

*Proof.* From (83) we have,

$$\widetilde{q}_{\mathbf{j}}^{\left\lfloor \frac{\sqrt[p]{N}}{2} \right\rfloor^p}(X_0,\ldots,X_{p-1}) \cdot \widetilde{q}_{\mathbf{j}'}^{\left\lfloor \frac{\sqrt[p]{N}}{2} \right\rfloor^p}(X_0,\ldots,X_{p-1}) = \prod_{a=0}^{p-1} \widetilde{\ell}_{j_a}^{\left(a,\left\lfloor \frac{\sqrt[p]{N}}{2} \right\rfloor\right)}(X_a) \cdot \widetilde{\ell}_{j_a'}^{\left(a,\left\lfloor \frac{\sqrt[p]{N}}{2} \right\rfloor\right)}(X_a) \tag{85}$$

Let us fix $0 \leq a < p$.

Because (82) is of the same form as in (48), we can apply Lemma 8 to each product $\widetilde{\ell}_{j_a}^{\left(a,\left\lfloor \frac{\sqrt[p]{N}}{2} \right\rfloor\right)}(X_a) \cdot \widetilde{\ell}_{j_a'}^{\left(a,\left\lfloor \frac{\sqrt[p]{N}}{2} \right\rfloor\right)}(X_a)$, which gives us

$$\widetilde{\ell}_{j_a}^{\left(a,\left\lfloor \frac{\sqrt[p]{N}}{2} \right\rfloor\right)}(X_a) \cdot \widetilde{\ell}_{j_a'}^{\left(a,\left\lfloor \frac{\sqrt[p]{N}}{2} \right\rfloor\right)}(X_a) = \sum_{m_a=j_a+j_a'}^{\sqrt[p]{N}-1} \alpha_{j_a+j_a',m_a}^{(a)} \cdot \widetilde{\ell}_{m_a}^{\left(a,\,\sqrt[p]{N}\right)}(X_a).$$

Going back to (85), we get

$$\widetilde{q}_{\mathbf{j}}^{\left\lfloor \frac{\sqrt[p]{N}}{2} \right\rfloor^p}(X_0,\ldots,X_{p-1}) \cdot \widetilde{q}_{\mathbf{j}'}^{\left\lfloor \frac{\sqrt[p]{N}}{2} \right\rfloor^p}(X_0,\ldots,X_{p-1}) = \prod_{a=0}^{p-1} \sum_{m_a=j_a+j_a'}^{N-1} \alpha_{j_a+j_a',m_a}^{(a)} \cdot \widetilde{\ell}_{m_a}^{\left(a,\,\sqrt[p]{N}\right)}(X_a).$$

Let $\alpha_{\mathbf{j}+\mathbf{j}',\mathbf{m}} = \prod_{a=0}^{p-1} \alpha_{j_a+j_a',m_a}^{(a)}$. Then we get

$$\widetilde{q}_{\mathbf{j}}^{\left\lfloor \frac{\sqrt[p]{N}}{2} \right\rfloor^p}(X_0,\ldots,X_{p-1}) \cdot \widetilde{q}_{\mathbf{j}'}^{\left\lfloor \frac{\sqrt[p]{N}}{2} \right\rfloor^p}(X_0,\ldots,X_{p-1}) = \sum_{\mathbf{j}+\mathbf{j}'\preceq\mathbf{m}\in[0,\,\sqrt[p]{N})^p} \alpha_{\mathbf{j}+\mathbf{j}',\mathbf{m}}\,\widetilde{q}_{\mathbf{m}}^N(X_0,\ldots,X_{p-1}),$$

as desired. $\qquad\square$

We now define the following padding scheme, PAD $(\mathbf{k})$.

---
**Algorithm 5** PAD$(\mathbf{k})$
---
**Input:** $\mathbf{k} \in \left( \mathbb{R}^{\left\lfloor \frac{\sqrt[p]{N}}{2} \right\rfloor} \right)^{p}$, indexed as $\mathbf{k_j}$ for $\mathbf{j} \in \left[ 0, \frac{\sqrt[p]{N}}{2} \right)^{p}$

**Output:** $\mathbf{k}' \in \left( \mathbb{R}^{\sqrt[p]{N}} \right)^{p}$

1: **for** $\mathbf{j} \in \left[ 0, \frac{\sqrt[p]{N}}{2} \right)^{p}$ **do**

2:      **if** $j_a \geq \left\lfloor \frac{\sqrt[p]{N}}{2} \right\rfloor$ for $0 \leq a < p$ **then**

3:          $\mathbf{k}'_{(j_0,\ldots,j_{p-1})} \leftarrow \mathbf{k}_{\left( j_0 + \left\lceil \frac{\sqrt[p]{N}}{2} \right\rceil, \ldots, j_{p-1} + \left\lceil \frac{\sqrt[p]{N}}{2} \right\rceil \right)}$

4:      **else**

5:          $\mathbf{k}'_\mathbf{j} = 0$

6: **return** $\mathbf{k}'$

---

The above basis and padding scheme allows us to extend Theorem 3 to general $p$.

**Theorem 17.** *Fix a family of basis polynomials $\widetilde{q}^N_\mathbf{j}(\mathbf{X})$ as defined in (83). Let $N \geq 1$ be a perfect power of $p$, $n \leq \left\lfloor \frac{\sqrt[p]{N}}{2} \right\rfloor^p$, $\mathbf{k}, \mathbf{u} \in \mathbb{R}^n$ and $\mathbf{M}'_N$ defined by basis $\widetilde{q}^N_\mathbf{j}(\mathbf{X})$. Then the operation*

$$\mathbf{u} \mapsto \left( \mathbf{M}'^{-1}_N \left( \mathbf{M}'_N \cdot \mathrm{PAD}(\mathbf{k}) \circ \mathbf{M}'_N \cdot \mathrm{PAD}(\mathbf{u}) \right) \right) [0:n-1] \tag{86}$$

*defines a causal map in $\mathbf{u}$.*

*Proof.* Let $\mathbf{k}' = \mathrm{PAD}(\mathbf{k})$, $\mathbf{u}' = \mathrm{PAD}(\mathbf{u})$, and

$$\mathbf{f} = \left( \mathbf{M}'^{-1}_N \left( \mathbf{M}'_N \cdot \mathbf{k}' \circ \mathbf{M}'_N \cdot \mathbf{u}' \right) \right) [0:n-1]. \tag{87}$$

In order to prove that (86) is causal in the input $\mathbf{u} \in \mathbb{R}^n$, we must show that for all $\mathbf{i} \in \left[ 0, \sqrt[p]{N} \right)^p$, $\mathbf{f_i}$ is dependent only on $\mathbf{u_{i'}}$ for $\mathbf{i}' \preceq \mathbf{i}$.

By Theorem 15, $\mathbf{M}'_N \cdot \mathbf{k}'$ and $\mathbf{M}'_N \cdot \mathbf{u}'$ correspond to the evaluations of the polynomials

$$k'(X_0, \ldots, X_{p-1}) = \sum_{\mathbf{j}' \in \left[ 0, \sqrt[p]{N} \right)^p} k'_{\mathbf{j}'} \cdot \widetilde{q}^N_{\mathbf{j}'}(X_0, \ldots, X_{p-1}), \quad \text{and}$$

$$u'(X_0, \ldots, X_{p-1}) = \sum_{\mathbf{j}' \in \left[ 0, \sqrt[p]{N} \right)^p} u'_{\mathbf{j}'} \cdot \widetilde{q}^N_{\mathbf{j}'}(X_0, \ldots, X_{p-1}), \tag{88}$$

respectively. Let us define

$$k(X_0, \ldots, X_{p-1}) = \sum_{\mathbf{j} \in \left[ 0, \left\lfloor \frac{\sqrt[p]{N}}{2} \right\rfloor \right)^p} k_\mathbf{j} \cdot \widetilde{q}^{\left\lfloor \frac{\sqrt[p]{N}}{2} \right\rfloor^p}_\mathbf{j}(X_0, \ldots, X_{p-1}), \quad \text{and}$$

$$u(X_0, \ldots, X_{p-1}) = \sum_{\mathbf{j} \in \left[ 0, \left\lfloor \frac{\sqrt[p]{N}}{2} \right\rfloor \right)^p} u_\mathbf{j} \cdot \widetilde{q}^{\left\lfloor \frac{\sqrt[p]{N}}{2} \right\rfloor^p}_\mathbf{j}(X_0, \ldots, X_{p-1}). \tag{89}$$

Let us define $\deg\left( \widetilde{q}^N_\mathbf{j} \right) = \left( \sqrt[p]{N} - j_a - 1 \right)^{p-1}_{a=0}$ as the vector of length $p$ consisting of the degrees of the component univariate polynomials $\widetilde{\ell}^{(a)}_{\mathbf{j},\mathbf{i}}(X_a)$ defined as in (82). Then we have $\deg\left( \widetilde{q}^{\left\lfloor \frac{\sqrt[p]{N}}{2} \right\rfloor^p}_\mathbf{j} \right) =$

$\left(\left\lfloor\frac{\sqrt[p]{N}}{2}\right\rfloor - j_0 - 1, \ldots, \left\lfloor\frac{\sqrt[p]{N}}{2}\right\rfloor - j_{p-1} - 1\right)$ for $\mathbf{j} = (j_0, \cdots, j_{p-1})$ such that $0 \le j_a < \left\lceil\frac{\sqrt[p]{N}}{2}\right\rceil$ for $0 \le a < p$. Further, for $\mathbf{j}' = \left(j_0 + \left\lceil\frac{\sqrt[p]{N}}{2}\right\rceil, \cdots, j_{p-1} + \left\lceil\frac{\sqrt[p]{N}}{2}\right\rceil\right)$ we have

$$
\begin{aligned}
\deg\left(\widetilde{q}_{\mathbf{j}'}^N\right) &= \left(\sqrt[p]{N} - j_0' - 1, \ldots, \sqrt[p]{N} - j_{p-1}' - 1\right) \\
&= \left(\sqrt[p]{N} - \left\lceil\frac{\sqrt[p]{N}}{2}\right\rceil - j_0 - 1, \ldots, \sqrt[p]{N} - \left\lceil\frac{\sqrt[p]{N}}{2}\right\rceil - j_{p-1} - 1\right) \\
&= \left(\left\lfloor\frac{\sqrt[p]{N}}{2}\right\rfloor^p - j_0 - 1, \ldots, \left\lfloor\frac{\sqrt[p]{N}}{2}\right\rfloor^p - j_{p-1} - 1\right).
\end{aligned}
$$

Since $\deg\left(\widetilde{q}_{\mathbf{j}}^{\left\lfloor\frac{\sqrt[p]{N}}{2}\right\rfloor^p}\right) = \deg\left(\widetilde{q}_{\mathbf{j}'}^N\right)$, we can set $\widetilde{\ell}_{\mathbf{j},\mathbf{i}}^{(a)}(X_a) = \widetilde{\ell}_{\mathbf{j}',\mathbf{i}}^{(a)}(X_a)$. Similarly, note that for $\mathbf{j}, \mathbf{j}'$ as above, $k_{\mathbf{j}'}' = k_{\mathbf{j}}$ and $u_{\mathbf{j}'}' = u_{\mathbf{j}}$. Then it follows that $k(X_0, \ldots, X_{p-1}) = k'(X_0, \ldots, X_{p-1})$, and by a similar argument, $u(X_0, \ldots, X_{p-1}) = u'(X_0, \ldots, X_{p-1})$. Then by Theorem 16 we have

$$
\begin{aligned}
f(X_0, \ldots, X_{p-1}) &= k(X_0, \ldots, X_{p-1}) \cdot u(X_0, \ldots, X_{p-1}) \quad \mathrm{mod}\ \left(T_{\sqrt[p]{N}}(X_0), \ldots, T_{\sqrt[p]{N}}(X_{p-1})\right) \\
&= k(X_0, \ldots, X_{p-1}) \cdot u(X_0, \ldots, X_{p-1}) \quad\quad\quad\quad\quad\quad\quad\quad\quad\quad (90)
\end{aligned}
$$

where the second line follows by observing that each $0 \le a < p$, we have $\deg_{X_a}(k(\mathbf{X}) \cdot u(\mathbf{X})) < 2\left\lfloor\frac{\sqrt[p]{N}}{2}\right\rfloor$ and observing that $2\left\lfloor\frac{\sqrt[p]{N}}{2}\right\rfloor \le \sqrt[p]{N}$. We want to write $f(X_0, \ldots, X_{p-1})$ in the form

$$
f(X_0, \ldots, X_{p-1}) = \sum_{\mathbf{m} \in \left[0, \sqrt[p]{N}\right)^p} f_{\mathbf{m}} \cdot \widetilde{q}_{\mathbf{m}}^N(X_0, \ldots, X_{p-1}),
$$

for a set of coefficients $f_{\mathbf{m}}$. From (89) and (90) we get

$$
f(X_0, \ldots, X_{p-1}) = \sum_{\mathbf{j}, \mathbf{j}' \in \left[0, \left\lfloor\frac{\sqrt[p]{N}}{2}\right\rfloor\right)^p} k_{\mathbf{j}'} u_{\mathbf{j}} \cdot \widetilde{q}_{\mathbf{j}'}^{\left\lfloor\frac{\sqrt[p]{N}}{2}\right\rfloor^p}(X_0, \ldots, X_{p-1}) \cdot \widetilde{q}_{\mathbf{j}}^{\left\lfloor\frac{\sqrt[p]{N}}{2}\right\rfloor^p}(X_0, \ldots, X_{p-1}).
$$

Then by Lemma 18 we have

$$
f(X_0, \ldots, X_{p-1}) = \sum_{\mathbf{j}, \mathbf{j}' \in \left[0, \left\lfloor\frac{\sqrt[p]{N}}{2}\right\rfloor\right)^p} k_{\mathbf{j}'} u_{\mathbf{j}} \cdot \sum_{\mathbf{j}+\mathbf{j}' \preceq \mathbf{m} \in [0, \sqrt[p]{N})^p} \alpha_{\mathbf{j}+\mathbf{j}',\mathbf{m}}\, \widetilde{q}_{\mathbf{m}}^N(X_0, \ldots, X_{p-1}).
$$

Thus, for any $\mathbf{m} \in \left[0, \sqrt[p]{N}\right)^p$, we have

$$
f_{\mathbf{m}} = \sum_{\mathbf{j}+\mathbf{j}' \preceq \mathbf{m}} \alpha_{\mathbf{j}+\mathbf{j}',\mathbf{m}} \cdot k_{\mathbf{j}'} u_{\mathbf{j}}
$$

implying that $f_{\mathbf{m}}$ depends only on $u_{\mathbf{j}}$ for $\mathbf{j} \preceq \mathbf{m}$, as desired.

$\qquad\qquad\qquad\qquad\qquad\qquad\qquad\qquad\qquad\qquad\qquad\qquad\qquad\qquad\qquad\qquad\qquad\qquad\qquad\qquad\square$

### D.6.4   $p$-variate Monarch

Recall that we have fixed $b = \sqrt[p]{N}$ (and hence $N = b^p$).

Define the $p$-variate Monarch matrix as follows:

$$
\mathbf{M}' = \mathbf{P}_{b,N}^R \left(\prod_{a=0}^{p-2} \mathbf{B}_{p-1-a} \cdot \left(\mathbf{P}_{b^{a+1},b^{a+2},N}\right)^\top\right) \mathbf{B}_0, \qquad\qquad (91)
$$

Where each $\mathbf{B}_a$ is block diagonal with $b \times b$ blocks for every $0 \le a < p$. Recall that Equation (1) has a permutation $\mathbf{P}_0$ at the end while the above definition does not have any permutation at the end. One trivial way to show the equivalence of above to Equation (1) is to define $\mathbf{P}_0 = \mathbf{I}$. In Appendix D.6.5,

we show that exists other non-trivial choices for $\mathbf{P}_0$. Further for $1 \leq i \leq p$, the $\mathbf{P}_i$ in Equation (1) connect to the above definition as follows:

$$\mathbf{P}_i = \begin{cases} \left(\mathbf{P}_{b^{p-i},b^{p-i+1},N}\right)^\top & \text{for } 1 \leq i \leq p-1 \\ \mathbf{P}_{b,N}^R & \text{for } i = p \end{cases}.$$

Finally we connect the above definition of $p$-variate Monarch to the earlier definition based on polynomial evaluation:

**Lemma 19.** *Equation (75) can be written as Equation (91).*

*Proof.* In (91), $\mathbf{B}_a$ would correspond to the evaluations $\ell_{\mathbf{j},\mathbf{i}}^{(a)}\left(\omega_{\sqrt[p]{N},i_a}\right)$ from earlier. Specifically the following holds for any $0 \leq a < p$. We index the $q^{p-1}$ blocks of $\mathbf{B}_a$ by $(i_0,\ldots,i_{a-1}),(j_{p-1},\ldots,j_{a+1})$ and the row and column 'offsets' within each such block are indexed by $i_a$ and $j_a$ respectively. Connecting back to $\ell_{\mathbf{j},\mathbf{i}}^{(a)}(\cdot)$ we set

$$\mathbf{B}_a^{((i_0,\ldots,i_{a-1}),(j_{p-1},\ldots,j_{a+1}))}[i_a,j_a] = \ell_{\mathbf{j},\mathbf{i}}^{(a)}\left(\omega_{\sqrt[p]{N},i_a}\right). \tag{92}$$

We will prove the claim as follows. Let $\mathbf{e_j} \in \mathbb{R}^N$ have a 1 in the $\mathbf{j}^{\text{th}}$ location and 0's elsewhere. Our goal is to multiply this vector on the right of Equation (91) and show that we get the $\mathbf{j}^{\text{th}}$ column of $\mathbf{M}'$ as defined in Equation (75). We will do so by induction.

Define $\mathbf{y}_0 = \mathbf{e_j}$ and $\mathbf{y}_1 = \mathbf{B}_0 \cdot \mathbf{y}_0$. Then for every $1 \leq a < p$, define

$$\mathbf{y}_{a+1} = \mathbf{B}_a \left(\mathbf{P}_{b^{p-a},b^{p-a+1},N}\right)^\top \mathbf{y}_a.$$

Note that we have

$$\mathbf{M}' \cdot \mathbf{e_j} = \mathbf{P}_{b,N}^R \cdot \mathbf{y}_p. \tag{93}$$

Next, we claim that for every $1 \leq a \leq p$, we have for every $(i_0,\ldots,i_{a-1}) \in [0,\sqrt[p]{N})^{a-1}$,

$$\mathbf{y}_a\left[((i_0,\ldots,i_{a-2}),(j_{p-1},\ldots,j_a),i_{a-1})\right] = \prod_{b=0}^{a-1} \ell_{\mathbf{j},\mathbf{i}}^{(b)}\left(\omega_{\sqrt[p]{N},b}\right), \tag{94}$$

where for $a = p$, we think of $((i_0,\ldots,i_{a-2}),(j_{p-1},\ldots,j_a),i_{a-1}) = (i_0,\ldots,i_{p-1})$.

We first note that Equation (94) is enough to prove the claim. Indeed we have that

$$\mathbf{y}_p\left[((i_0,\ldots,i_{p-1}))\right] = \prod_{b=0}^{p-1} \ell_{\mathbf{j},\mathbf{i}}^{(b)}\left(\omega_{\sqrt[p]{N},b}\right),$$

where the RHS in the above is the same as RHS in Equation (75). To see the claim note that by definition of $\mathbf{P}_{b,N}^R$, we have $\left(\mathbf{P}_{b,N}^R \cdot \mathbf{y}_p\right)[(i_{p-1},\ldots,i_0)] = \mathbf{y}_p[(i_0,\ldots,i_{p-1})]$. Equation (93) then established the claim.

To complete the proof we prove Equation (94) by induction on $a$.

We next consider the base case of $a = 1$. Note that $\mathbf{y}_1$ is just the $\mathbf{j}$th column of $\mathbf{B}_0$ (as $\mathbf{y}_0 = \mathbf{e_j}$) and in that case Equation (94) follows from Equation (92).

For the inductive hypothesis, assume that Equation (94) is true for $a$ for some $a \geq 1$. We now want to argue Equation (94) for $\mathbf{y}_{a+1}$. Towards that end define

$$\mathbf{z}_a = \left(\mathbf{P}_{b^{p-a},b^{p-a+1},N}\right)^\top \mathbf{y}_a = \mathbf{P}_{b,b^{p-a+1},N}\mathbf{y}_a.$$

Note that

$$\mathbf{y}_{a+1} = \mathbf{B}_a \cdot \mathbf{z}_a. \tag{95}$$

Now by definition of $\mathbf{P}_{b,b^{p-a+1},N}$, we have

$$\mathbf{z}_a\left[((i_0,\ldots,i_{a-1}),(j_{p-1},\ldots,j_a))\right] = \mathbf{y}_a\left[((i_0,\ldots,i_{a-2}),(j_{p-1},\ldots,j_a),i_{a-1})\right].$$

We claim that Equation (94) is true for $a + 1$ from the above along with Equation (95) and Equation (92). Indeed, fix any $(i_0, \ldots, i_{a-1})$. Then in Equation (95) the entry $\mathbf{z}_a\left[((i_0, \ldots, i_{a-1}), (j_{p-1}, \ldots, j_a))\right]$ gets multiplied by the entries $\mathbf{B}_a^{((i_0, \ldots, i_{a-1}), (j_{p-1}, \ldots, j_{a+1}))}[i_a, j_a]$ for all values of $i_a \in [0, \sqrt[p]{N})$. The inductive hypothesis and Equation (92) then proves the inductive step, as desired. $\qquad\square$

Finally, we note that we can generalize Algorithm 4 for constructing $\mathbf{B}_a$ for $0 \leq a < p$ (since this is the multivariate case some of the steps are a bit simplified):

---

**Algorithm 6** BLOCKY MULTIVAR MONARCH($\widetilde{\mathbf{B}}_{(0)}, \ldots \widetilde{\mathbf{B}}_{(p-1)}, N, p$)

---

**Input:** $\widetilde{\mathbf{B}}_{(0)}, \ldots, \widetilde{\mathbf{B}}_{(p-1)} \in \mathbb{R}^{N \times b}$ where $b = \sqrt[p]{N}$
**Output:** Block diagonal matrices $\mathbf{B}_0, \ldots, \mathbf{B}_{p-1} \in \mathbb{R}^{N \times N}$
$\hspace{5cm} \triangleright$ Compose each output matrix from corresponding input matrix
1: **for** $y \leftarrow 0$ to $p - 1$ **do**
2: $\quad$ **for** $a \leftarrow 0$ to $\frac{N}{b} - 1$ **do**
3: $\quad\quad$ $\mathbf{B}_y^{(a)} \leftarrow \mathbf{C}_b \cdot \widetilde{\mathbf{B}}_{(y)}^{(a)}[ab : ab + b - 1, :]$
4: $\quad$ $\mathbf{B}_y \leftarrow \text{diag}(\mathbf{B}_y^{(0)}, \ldots \mathbf{B}_y^{(\frac{N}{b}-1)})$
5: **return** $\mathbf{B}_0, \ldots, \mathbf{B}_{p-1}$

---

### D.6.5 Extensions and Open Questions

In this sub-section we outline certain (fairly) straightforward extension of our theoretical results and conclude with some open questions.

**Comparing Equation (91) to Equation (1)** We note that we can post multiply $\mathbf{M}$ in Equation (91) with a large class of permutations for which Theorem 17 still holds. We outline the technical reason why this is true. At the heart of the argument for why Equation (86) gives a causal map is Lemma 18. Specifically note that the sum in RHS in Equation (84), is over all $\mathbf{j} + \mathbf{j}' \preceq \mathbf{m}$. The main observation is that this partial order still holds if we permute the $b$-variate representation of $\mathbf{j}, \mathbf{j}'$ and $\mathbf{m}$ in the same way. In other words, for any permutation $\sigma : [0, p) \to [0, p)$ if we define $\sigma(\mathbf{j}) = \left(j_{\sigma(0)}, \ldots, j_{\sigma(p-1)}\right)$ and similarly $\sigma(\mathbf{j}'), \sigma(\mathbf{m})$. Then we still have $\sigma(\mathbf{j}) + \sigma(\mathbf{j}') \preceq \sigma(\mathbf{m})$. This in turn implies the following. Let $\mathbf{P}_\sigma$ be a permutation that maps $\mathbf{j} \in [0, \sqrt[p]{N})^p$ to $\sigma(\mathbf{j}) \in [0, \sqrt[p]{N})^p$. Then Theorem 17 holds if we replace $\mathbf{M}'$ by $\mathbf{M} \cdot \mathbf{P}_\sigma$ with $\mathbf{M}$ as in Equation (91).

**Evaluation points** Our results as presented are for specific classes of evaluation points. A natural question to ask is if our results can be extended to more general set of evaluation points. It turns out that our results for $p$-variate Monarch matrices can be extended to a wider class of evaluation points. Specifically, for each $0 \leq a < p$, let $S_a \subset \mathbb{C}$ with $|S_a| = \sqrt[p]{N}$. Then our results in this sub-section hold if we replace the evaluation points from $A^p$ to $\times_{a=0}^{p-1} S_a$. The only thing that changes in our proofs is that in Theorem 16, we replace $\mod \left(T_{\sqrt[p]{N}}(X_0), \ldots, T_{\sqrt[p]{N}}(X_{p-1})\right)$ by $\mod \left(q_{S_0}(X_0), \ldots, q_{S_{p-1}}(X_{p-1})\right)$, where $q_A(Z)$ is as defined in Equation (81). This result can then be propagated throughout the rest of our proofs.

On the other hand, our results in Appendix D.3 and Appendix D.5 do exploit specific properties of the evaluation points (specifically $\left(\omega_N^i\right)^{\sqrt{N}} = \omega_{\sqrt{N}}^{i_0}$ for Appendix D.3 and $T_{\sqrt{N}}(\omega_{N,i}) = (-1)^{i_1}\omega_{\sqrt{N}, i_0}$ for Appendix D.5). To generalize these results to other sets of evaluation points, we need the existence of degree $\sqrt{N}$ polynomial that maps (in a $\sqrt{N}$-to-1 fashion) $A$ to a set of $\sqrt{N}$ elements. Another interesting open question is to avoid the blowup $n \to 2^p \cdot n$ in Theorem 17 and ideally only pay a blowup $n \to 2n$ for every $p \geq 2$ as we were able to do in Appendix D.3 and Appendix D.5 (with $p = 2$).

## E  Implementation

```
1  from einops import rearrange
2  import torch
3  from torch import nn
4
5  def blockdiag_matmul(x, w):
6      return torch.einsum(
7          "bnm,...bm->...bn", w, x.view(*x.shape[:-1], w.shape[0],  w.
   shape[-1])
8      ).reshape(*x.shape)
9
10 class MonarchMatrix(nn.Module):
11
12     def __init__(self, sqrt_n: int):
13         super().__init__()
14         self.sqrt_n = sqrt_n
15         self.L = nn.Parameter(torch.randn((sqrt_n, sqrt_n, sqrt_n)))
16         self.R = nn.Parameter(torch.randn((sqrt_n, sqrt_n, sqrt_n)))
17
18     def forward(self, x):
19         x = rearrange(x, "... (m n) -> ... (n m)", n=self.sqrt_n)
20         x = blockdiag_matmul(x, self.L)
21         x = rearrange(x, "... (m n) -> ... (n m)", n=self.sqrt_n)
22         x = blockdiag_matmul(x, self.R)
23         return rearrange(x, "... (m n) -> ... (n m)", n=self.sqrt_n)
24
25 class MonarchMixerLayer(nn.Module):
26     def __init__(self, sqrt_n: int, sqrt_d: int):
27         super().__init__()
28         self.m1 = MonarchMatrix(sqrt_n)
29         self.m2 = MonarchMatrix(sqrt_n)
30         self.m3 = MonarchMatrix(sqrt_d)
31         self.m4 = MonarchMatrix(sqrt_d)
32
33         self.n_kernel = nn.Parameter(torch.randn(sqrt_d ** 2, sqrt_n
   ** 2))
34         self.d_kernel = nn.Parameter(torch.randn(1, sqrt_d ** 2))
35         self.layer_norm = nn.LayerNorm(sqrt_d ** 2)
36
37     def forward(self, x: torch.Tensor):  # x.shape = (b, n, d)
38         x_tilde = self.m2(self.n_kernel * self.m1(x.transpose(-1, -2))
   ).transpose(-1, -2)  # mix sequence
39         y = self.m4(torch.relu(self.d_kernel * self.m3(x_tilde)))  #
   mix features
40         return self.layer_norm(y + x_tilde)  # skip connection
```
Listing 1: A basic implementation of the M2 layer.

