# OpenReview forum: "Monarch Mixer: A Simple Sub-Quadratic GEMM-Based Architecture"
_NeurIPS.cc/2023/Conference — NeurIPS 2023 oral_

### Official Review · Reviewer_zN2M · 2023-07-04

**Soundness:** 4 excellent
**Presentation:** 3 good
**Contribution:** 4 excellent
**Rating:** 8
**Confidence:** 3

**Summary:**

The paper introduces a new neural network layer which runs efficiently on modern GPUs, and exhibits strong performance against state-of-the art on several benchmarks. The layer is based on Monarch matrices, introduced in [7]. Monarch matrices use permutation matrices, and block diagonal matrices to represent dependancies across features and temporal dimensions. Inspired by butterfly matrices in FFT, this matrix parameterization can represent anything from convolutions to fully connected matrices, depending on their order. The Monarch Mixer layer introduced in this paper uses two monarch matrices as well as a matrix (K) that is used in point-wise multiplication. Performance is compared on language and image classification, replacing transformer and fully connected layers with the M2 layer. Finally, the paper presents a theoretical derivation of how this layer can be modified for use in causal language tasks, while maintaining its computational efficiency.

**Strengths:**

Technically, this paper makes a strong contribution by proposing a computationally efficient layer. The computational performance is benchmarked on modern GPUs. I really liked the discussion on factors affecting runtime performance on modern GPUs -- this is a really valuable introduction to this topic. The proposed layer achieves superior performance, with fewer trainable parameters, in less time. Like MLPMixer, it also does away with the attention mechanism of transformers -- which suggests M2 is also a good architectural inductive bias -- perhaps competitive with attention? Comparing against [7], I would say the largest technical contribution of this paper is on how to modify the M2 layer perform in the causal setting.

**Weaknesses:**

The main weakness of the paper is that it is somewhat hard to read/understand without having to read [7]. Section 4 in particular is quite dense. It seems that the authors struggled to fit the paper within the NeurIPS page limit.

**Questions:**

-Does the expressivity of M increase with the order p? Perhaps this is obvious, but it should be stated explicitly in the paper. Can M be used to express any dense matrix?

-Why do you think your model exceeds the performance of other state-of-the art models without using attention? I always thought one of the most important aspects of attention was that it afforded permutation invariance, does M2 serve a similar purpose?

**Limitations:**

Limitations were addressed in second paragraph of Section 6.

---

> ### Author Rebuttal · Authors · 2023-08-09
>
> Thank you for your positive feedback on the technical contributions of our paper. We are glad that you found the GPU performance discussion insightful, and we appreciate your constructive comments on how to improve the paper.
>
> **W1. More intuitive explanation of Monarch matrices.** Thank you for your suggestions on how to improve the clarity of the paper. We plan to use the extra space in the camera ready to include both a more complete description of Monarch matrices and the motivation behind their definition. We plan to motivate the Monarch matrices via the FFT algorithm, as follows:
>
> The motivation behind the Monarch matrix is to adapt and generalize the FFT algorithm. The FFT algorithm splits an FFT of size N into smaller FFT’s over portions of the input, interleaved with permutations. More precisely, let $F_N$ denote the Fourier transform of size $N$, and assume N is a perfect square for simplicity. The FFT breaks down $F_N$ as follows:
>
> $F_N = P F_L P D P F_R P,$
>
> where $F_L$ and $F_R$ are block-diagonal matrices whose blocks are made of $F_\sqrt{N}$, $D$ is a diagonal matrix, and $P$ is a permutation that reshapes a 1D input into $\sqrt{N} \times \sqrt{N}$, and takes the transpose.
>
> A Monarch matrix generalizes this computation pattern by “rolling in” the diagonal matrix, and letting the blocks in the block-diagonal matrices be arbitrary instead of fixed to an FFT:
>
> $M = PLPRP$
>
> This additional flexibility allows Monarch matrices to express a wider class of structured matrices than the FFT (but they are not as completely expressive as a dense matrix). In our paper, we also generalize Monarch matrices past the order 2 phase, so there can be more than two block-diagonal matrices interleaved with permutations (Figure 1, left in our original submission).
>
> **Q1. Expressivity of $M$ with the order $p$.** This is a little subtle - the expressivity actually goes _down_ with increasing $p$, since we decrease the sizes of the blocks (block size $b$ is $\sqrt{N}$ for order 2, $\sqrt[3]{N}$ for order 3: for general $p$ the block size is $b=\sqrt[p]{N}$). This in turn implies that a $p$-variate Monarch has $O\left(pN^{1+1/p}\right)$ many parameters i.e. the number of parameters decreases as $p$ increases and hence its expressivity goes down (this follows e.g. from a counting argument).
>
> $p=1$ gives arbitrary dense matrix, but that is because in that case the block size is now $b=N$ and hence is a trivial case. For $p>1$, a single $p$-variate Monarch matrix cannot express an arbitrary matrix.
>
> However, one can show that one can express an arbitrary $N\times N$ matrix as a product of $m=O(N)$  matrices $M_1,\dots,M_m$ where each $M_j$ is a $p$-variate Monarch matrix (this follows from a known result that an arbitrary matrix can be represented a product of O(N) Toeplitz matrices).
>
> We will add a discussion of these properties to the main body when we introduce $p$-order Monarch matrices.
>
> **Q2. Why does M2 outperform attention-based models?** There are two pieces – first, we build on prior work to build sub-quadratic replacements for attention. Second, we replace the MLPs with sub-quadratic alternatives, which achieves the same performance with the same height/width but fewer overall parameters.
>
> M2 builds on prior work studying how to replace attention with a sub-quadratic alternative while maintaining high quality. Many of these models use a combination of long and short convolutions with elementwise multiplication, e.g. [1, 2, 3, 4]. These convolutions are often computed with an FFT, which means that they can be expressed using Monarch matrices and elementwise multiplication. We build on the insights from these architectures when using Monarch for sequence mixing in our M2 models with an alternative that scales sub-quadratically in sequence length while maintaining high quality.
>
> In addition, M2 uses Monarch matrices to scale sub-quadratically in the model dimension by replacing MLPs. Our results – that the dense layers in MLPs can be replaced by sparse(r) matrices without losing quality – may suggest that the current generation of models is overparameterized, and that there exist much more efficient architectures to develop. We are excited by these possibilities, and we look forward to building on these ideas in the future.
>
> [1] Long Range Language Modeling via Gated State Spaces. Harsh Mehta, Ankit Gupta, Ashok Cutkosky, Behnam Neyshabur. ICLR 2022.
>
> [2] Pretraining Without Attention. Junxiong Wang, Jing Nathan Yan, Albert Gu, Alexander M. Rush. ACL 2023.
>
> [3] Hungry Hungry Hippos: Towards Language Modeling with State Space Models. Daniel Y. Fu, Tri Dao, Khaled K. Saab, Armin W. Thomas, Atri Rudra, Christopher Ré. ICLR 2023.
>
> [4] Hyena Hierarchy: Towards Larger Convolutional Language Models. Michael Poli, Stefano Massaroli, Eric Nguyen, Daniel Y. Fu, Tri Dao, Stephen Baccus, Yoshua Bengio, Stefano Ermon, Christopher Ré. ICML 2023.

---

> > ### Comment · Reviewer_zN2M · 2023-08-14
> > **Reply to Rebuttal**
> >
> > Thank you for your clarifying comments. I suggest you incorporate them in your paper.

---

> > > ### Author Response · Authors · 2023-08-14
> > >
> > > Thank you, we have added it to our updated manuscript!

---

### Official Review · Reviewer_qwKK · 2023-07-05

**Soundness:** 4 excellent
**Presentation:** 4 excellent
**Contribution:** 4 excellent
**Rating:** 8
**Confidence:** 5

**Summary:**

This paper takes a fresh approach by addressing the issue of high complexity in current neural networks. It points out that the computational complexity of Transformers is quadratic with respect to both the sequence length and the feature dimension. Previous papers primarily focused on reducing the complexity related to sequence length, but this paper is the first to propose a method that reduces complexity for both sequence length and feature dimension. The specific approach used is the utilization of second-order Monarch Matrices, with the model structure referred to as the Monarch Mixer. Additionally, the authors introduce a novel initialization method for the Monarch Matrices, enabling them to handle causal l language modeling. The effectiveness of this approach is validated in the areas of non-causal language modeling, causal language modeling, and image classification.

**Strengths:**

Indeed, it is novel to consider the optimization of complexity from both the sequence dimension and the feature dimension. Furthermore, initializing the model for the causal scenario poses a definite challenge, and the authors have successfully accomplished this task.

**Weaknesses:**

There aren't many weaknesses, for specific questions, please refer to the **Questions** section.

**Questions:**

1. I'm not very familiar with the Monarch Matrix, but is its core idea to use the product of block-diagonal matrices and permutation matrices as a replacement for the dense matrix?

2. What is the motivation behind the Monarch Matrix? Despite having significantly fewer parameters, it seems to perform comparably to dense matrices in small-scale models. Can this conclusion be extended to models larger than 10 billion parameters?

3. In Line 257, "We set the Monarch matrices to DFT and inverse DFT matrices, to simulate long convolutions [15, 37], and do not learn them." Does this refer to setting the block-diagonal matrices of the Monarch Matrix as DFT matrices?

4. The experiments for Non-Causal Language Modeling and Image Classification should be more comprehensive, such as considering the Monarch matrices as learnable.

5. Could you provide a more intuitive explanation of the initialization for the causal scenario? From examining the code, it seems like the Monarch Matrix is initialized as a lower triangular matrix?

**Limitations:**

Yes.

---

> ### Author Rebuttal · Authors · 2023-08-09
>
> Thank you for your positive feedback and questions. These questions have helped us improve the presentation of our paper. We provide a more detailed explanation of Monarch matrices below, which we plan to add to the paper.
>
> **Q1. Monarch Motivation.** The motivation behind the Monarch matrix is to adapt and generalize the FFT algorithm. The FFT algorithm splits an FFT of size N into smaller FFT’s over portions of the input, interleaved with permutations. More precisely, let $F_N$ denote the Fourier transform of size $N$, and assume N is a perfect square for simplicity. The FFT breaks down $F_N$ as follows:
>
> $F_N = P F_L P D P F_R P,$
>
> where $F_L$ and $F_R$ are block-diagonal matrices whose blocks are made of $F_\sqrt{N}$, $D$ is a diagonal matrix, and $P$ is a permutation that reshapes a 1D input into $\sqrt{N} \times \sqrt{N}$, and takes the transpose.
>
> A Monarch matrix generalizes this computation pattern by “rolling in” the diagonal matrix, and letting the blocks in the block-diagonal matrices be arbitrary instead of fixed to an FFT:
>
> $M = PLPRP$
>
> This additional flexibility allows Monarch matrices to express a wider class of structured matrices than the FFT (but they are not as completely expressive as a dense matrix). In our paper, we also generalize Monarch matrices past the order 2 phase, so there can be more than two block-diagonal matrices interleaved with permutations (Figure 1, left in our original submission).
>
> **Q2. Scaling Results.** We have seen promising initial scaling results – in the common response, we have reported results for both M2-BERT-Base and M2-BERT-Large. In our original submission, we also saw similar scaling performance on causal language modeling with GPT2-s and GPT2-m equivalent models (Table 9 in the main paper). We look forward to continuing to scale these models and seeing how well the trends hold.
>
> **Q3. Monarch DFT.** A Monarch matrix can exactly express both a DFT and an inverse DFT (see Appendix F, corollary 6 in the original submission for the exact parameterization – they are similar to DFT matrices, but with slight modifications). For these experiments, we set the Monarch matrices to express the DFT and inverse DFT, respectively.
>
> **Q4. Learnable Monarch.** Thank you for the suggestion to extend the experiments for learnable Monarch matrices. Please see the common response for additional experiments along these lines. Making the Monarch matrices learnable yields small benefits in quality.
>
> **Q5. Causal Monarch Interpretation.** One way to interpret a Monarch matrix is to view it as evaluating a polynomial at a set of evaluation points $(a_i)$. When the Monarch matrix is used in a convolution, it is equivalent to multiplying two polynomials, by multiplying their evaluations $h(a_i) = f(a_i)g(a_i)$. The causal parameterization ensures that the resulting polynomial $h(a_i)$ is causal in $f$ – i.e., its coefficients do not depend on coefficients of $f$ that come later in the sequence.

---

### Official Review · Reviewer_xUjR · 2023-07-06

**Soundness:** 3 good
**Presentation:** 4 excellent
**Contribution:** 3 good
**Rating:** 8
**Confidence:** 3

**Summary:**

The Monarch Mixer (M2) combines MLP mixer and Conv mixer and yields a new family of mixers that is formalized in terms of Monarch matrices. the approach is novel and reminds me of an extension of [15] in their reference.

The main advantage of M2 is in its sub-quadratic computation capability. The authors evaluate their method on a set of large language models and prove their framework functions properly on meaningful and challenging tasks. The paper is well-written and easy to follow.

**Strengths:**

The paper stands out in its clear and proper presentation. The idea is well motivated from a hardware perspective, explained intuitively, and studied theoretically. Evaluation of the method on causal and non-causal large language models is an asset and demonstrates a clear potential of the method.

**Weaknesses:**

- The new approach is only benchmarked on Transformer tasks, while speech applications look relevant but are ignored.
- It would be fair to see a comparison of inference latency and metric performance of the M2-based BERT with a configuration of BERT that has a smaller number of parameters, but the same number of FLOPs.

**Questions:**

- Table 3 mentions a "Transformer" model. What is the configuration of the model?
- Tables 4 to 7 mention BERT-s. What is BERT-s? What are the details of its architecture?
- Table 2 seems to be incomplete. The FLOP utilization for dense matmul is missing from Table 2.
- Caption of Table 4 mentions GPU, however the body of the Table contains no inference results on GPU. Did the authors perform inference on CPU only? Why GPU inference is not reported in Table 4?
- A comparison with SwinMLP-B [1] and other SWIN-v2 models is more informative than the relatively older ViT model.
- What is BERT-s? What is ViT-s? please provide the model details.

[1]: Zheng, Hao, Guohui Wang, and Xuchen Li. "Swin-MLP: a strawberry appearance quality identification method by Swin Transformer and multi-layer perceptron." Journal of Food Measurement and Characterization 16.4 (2022): 2789-2800.

**Limitations:**

While the method clearly shows that the method works well on Transformer models, convolutional models, and speech applications are ignored.

---

> ### Author Rebuttal · Authors · 2023-08-09
>
> Thank you for your positive feedback and insightful suggestions. We hope that the additional experiments reported in the common response have improved the paper, and we look forward to further discussion. Here, we answer the specific weaknesses and questions raised in your review.
>
> **W1. Evaluation on speech applications.** Thank you for the suggestion to evaluate on speech applications. Please see the experiment on speech commands in the common response. We hope this result helps provide further evidence for the generality of our method.
>
> **W2. Comparison against parameter-matched BERT models**. Thank you for your suggestion to evaluate M2-BERT against parameter-matched BERT models. We have reported the results of these experiments in our common response. The 80M BERT model underperforms M2-BERT in quality on GLUE. A highly-optimized 80M BERT achieves higher throughput than M2-BERT on short sequences, but underperforms on inputs longer than 1K.
>
> **Q1. Architecture in synthetics.** For these synthetics, all models, including the Transformer, are small two-layer models with hidden dimension 64.
>
> **Q2 & Q6. BERT-s, ViT-s.** “BERT-s” and “ViT-s” in the submission are typos, we mean BERT-base and ViT-B (the -s terminology is the equivalent model for GPT2-s). These are now fixed in the draft.
>
> **Q3. Dense Matmul FLOP Util.** We report FLOP utilization of dense matmul here, and we have added it to our updated manuscript:
>
> |      |   **4K**  |  **16K**  | **65K**  |
> |-----:|:-----:|:-----:|:-----:|
> | **A100** | 63.0% | 78.0% | 80.0% |
> | **4090** | 74.6% | 96.7% | 97.9% |
>
> **Q4. GPU Inference.** We have reported the GPU inference results in the common response, and we will add them to Table 4 in the updated manuscript.
>
> **Q5. Swin Comparisons.** Thank you for the suggestion to compare to Swin-v2 and Swin-MLP. We have run these experiments and reported them in the common response (Table **4** in the accompanying PDF).

---

> > ### Comment · Reviewer_xUjR · 2023-08-18
> > **Thanks**
> >
> > I would like to thank the authors for their rebuttal. I keep my rating.

---

### Author Rebuttal · Authors · 2023-08-09

# Common Response
We thank all reviewers for their time and valuable comments, which have helped us improve our paper. In this paper, we introduce Monarch Mixer, a new architecture that is hardware-efficient and sub-quadratic in both sequence length and model dimension. We demonstrate that Monarch Mixer can be used as a drop-in replacement for attention and MLP in Transformers in BERT-style, GPT-style, and ViT-style modeling, matching quality with up to 27% parameter reduction and up to 9x faster inference for long sequences. We are excited to take a first step with this work in developing new architectures that are fundamentally more efficient than Transformers while maintaining quality.

We are glad to have received positive feedback on the motivation (reviewers **xUjR**, **zN2M**), clear technical contribution (reviewers **qwKK**, **zN2M**), execution and experiments (reviewers **xUjR**, **zN2M**), and overall clarity of presentation (reviewers **xUjR**, **qwKK**).

In our general response, we are excited to report the results of some additional experiments in pretraining quality, as well as experiments requested by the reviewers:
* Stronger BERT results:
  * M2 achieves GLUE performance that matches BERT-Base from (Devlin et al 2018), with 27% fewer parameters and up to 9X faster GPU inference time for long sequences (requested by reviewer **xUjR**)
  * Scaling up to BERT-Large equivalent – M2-BERT-Large matches BERT-Large in quality with 24% fewer parameters and achieves up to 4X faster GPU inference time (demonstrating scaling performance, as requested by reviewer **qwKK**)
  * Benchmarks against smaller BERT models (requested by reviewer **xUjR**)
* Swin-M2: matching ImageNet accuracy with 32% fewer parameters when replacing attention and MLPs in Swin-V2 with Monarch Mixer, as a drop-in replacement (requested by reviewer **xUjR**)
* Speech: M2 matches state-of-the-art in classification accuracy when classifying raw 16 kHz speech signals on the SpeechCommands task (requested by reviewer **xUjR**)
* Experiments with learnable Monarch matrices in the sequence mixer for CIFAR, achieving 1.5 points in lift from learning the matrices (requested by reviewer **qwKK**)

The results tables for these experiments are in the accompanying PDF for the rebuttal. We plan to include these experiments in our updated manuscript and welcome further discussion on how to improve the paper during the discussion period.

## BERT
Since our initial submission, we have improved our BERT pretraining formula via improvements to pretraining hyperparameters and data. We are happy to report stronger downstream GLUE results, competitive with those from the official BERT-Base model trained by (Devlin et al 2018). We have also included an 80M BERT trained using our formula as a baseline, following the suggestion from reviewer **xUjR**.

Table **1** in the accompanying PDF shows the results. M2-BERT-base is competitive with the Devlin et al BERT-base, with 27% fewer parameters. When scaled up to parameter-match BERT-base, M2-BERT outperforms BERT-base by 1.3 GLUE points on average.

We have also scaled up to a BERT-Large equivalent, which is competitive with the BERT-Large trained by (Devlin et al 2018). Table **2** in the accompanying PDF shows that M2-BERT-large is competitive with BERT-large with 24% fewer parameters. These results suggest that M2 can scale well.  We are excited to scale up to even larger models in future work (as suggested by reviewer **qwKK**).

**BERT GPU Inference**

Next, we report additional results on GPU inference times for different sequence lengths (addressing a question by reviewer **xUjR**). Inference times are reported as throughput, in tokens/ms on a single A100-40GB.

Table **3** (top) in the accompanying PDF shows that M2-BERT-base achieves higher throughput than even highly-optimized BERT models, and achieves 9.0X higher throughput than a standard BERT-base model for long (4K) input sequences.

Table **3** (bottom) in the accompanying PDF reports throughput against the 80M BERT as well. When parameter-matched, M2-BERT is slower than the most highly-optimized attention kernels for short sequences, but still faster for long sequences.

Benchmarks for BERT-large and M2-BERT-large are omitted for space in the rebuttal, but the trends are similar (but the HF model OOM’s earlier, leading to a maximum speedup over HF of 4.34X at sequence length 2K).

## Swin-M2 Experiments
Reviewer **xUjR** suggests comparing ImageNet performance against Swin-V2 and Swin-MLP. Table **4** in the accompanying PDF reports the results of replacing attention and the MLP in Swin-V2, using M2 as a drop-in replacement. Surprisingly, Swin-M2 outperforms Swin-MLP-B, is competitive with Swin-V1-B, and comes within 1 point of Swin-V2-B – even without any hyperparameter tuning or architecture adjustment from our ViT formula.

We expect that performance may improve further with hyperparameter tuning specific to M2. These results provide additional evidence that M2 is a strong drop-in replacement for attention and MLPs in various architectures.

## Speech Applications
Reviewer **xUjR** suggests evaluating M2 on speech applications. Table **5** in the accompanying PDF presents the performance of M2 on Speech Commands-10, a speech classification task over raw 1-second clips sampled at 16 kHz. M2 is competitive with state-of-the-art architectures on this task.

## Learnable Monarch Matrices in Sequence Mixer
Reviewer **qwKK** suggests extending the experiments to include the case where the Monarch matrices are learnable. In our submission, the monarch matrices in the state mixer (i.e. MLP) are already learnable. Table **6** in the accompanying PDF presents an experiment evaluating sequence mixers (i.e. attention) with learnable monarch matrices on sequential CIFAR. Learning the Monarch matrices in the sequence mixer yields 1.5 points of lift. We look forward to further exploring this regime in future work.

---

### Decision · Program_Chairs · 2023-09-21

**Decision:**

Accept (oral)

**Comment:**

This paper proposed a new architecture called Monarch Mixer that is able to achieve sub-quadratic cost in many GEMM-based models including ViT, non-causal and causal language models.

The reviewers generally agree that this work has made nontrivial contribution to the field, and this paper made a timely contribution to the increased computation need for large models.

We also appreciate the additional experiments made by the authors, which we would also encourage the authors to include in their camera ready version.